# Integrator dynamics in the cortico-basal ganglia loop for flexible motor timing

Zidan Yang[1,2,3], Miho Inagaki[1], Charles R. Gerfen[4], Lorenzo Fontolan[5] & Hidehiko K. Inagaki[1✉]

Flexible control of motor timing is crucial for behaviour[1–4]. Before volitional movement begins, the frontal cortex and striatum exhibit ramping spiking activity, with variable ramp slopes anticipating movement onsets[5–12]. This activity in the cortico-basal ganglia loop may function as an adjustable 'timer,' triggering actions at the desired timing. However, because the frontal cortex and striatum share similar ramping dynamics and are both necessary for timing behaviours, distinguishing their individual roles in this timer function remains challenging. Here, to address this, we conducted perturbation experiments combined with multi-regional electrophysiology in mice performing a flexible lick-timing task. Following transient silencing of the frontal cortex, cortical and striatal activity swiftly returned to pre-silencing levels and resumed ramping, leading to a shift in lick timing close to the silencing duration. Conversely, briefly inhibiting the striatum caused a gradual decrease in ramping activity in both regions, with ramping resuming from post-inhibition levels, shifting lick timing beyond the inhibition duration. Thus, inhibiting the frontal cortex and striatum effectively paused and rewound the timer, respectively. These findings are consistent with a model in which the striatum is part of a network that temporally integrates input from the frontal cortex and generates ramping activity that regulates motor timing.

Flexible and precise control of motor timing is essential for most behaviours, including vocal communication and driving, as well as foraging and avoiding threats in animals[1–4]. Without this ability, behaviour would be limited to immediate reactions.

To execute timed actions, the brain tracks time over seconds and triggers actions at the desired moment, much like a timer beeping after a preset duration[2–5]. Neurons in the frontal cortex and basal ganglia, especially the striatum, exhibit neural correlates of such a 'timer': before voluntary movement begins, many neurons demonstrate a gradual change in spiking activity, such as ramping activity that peaks at movement onset[2,3,5–12]. When actions occur at various timings, the slope of this ramp changes, reaching a hypothetical threshold that triggers action at different timings[2,3,5–12]. Thus, alternating the speed of dynamics in these areas may serve as an adjustable timer (Fig. 1a).

Because isolated neurons can sustain activity only for tens of milliseconds, the seconds-long 'timer' dynamics probably arise from network interactions[12,13]. From a dynamical systems perspective, population activity traces trajectories in a high-dimensional state space, with each dimension corresponding to the activity of individual neurons. Network interactions can stabilize certain activity patterns, known as attractors[12–14]. Slow dynamics emerging from continuous or shallow point attractors enable temporal integration of network inputs[14]. Such integrator[14–20] networks can generate ramping activity by temporally integrating non-ramping (for example, step) inputs[9,16], with the input strength adjusting ramp speed (Fig. 1a). This integrator mechanism has therefore been proposed to underlie ramping activity for flexible motor timing[9,16].

Manipulations of the frontal cortex and striatum affect timing behaviour, supporting their causal roles[5,11,21–29]. Furthermore, neural correlates and perturbation effects during evidence accumulation[30,31] suggest the striatum as a key area for integration. However, most previous studies have examined neural correlates and manipulations separately, limiting insights into the computational roles of each area and their interactions[32,33]. First, neural correlates may be internally generated or externally driven. Second, a behaviourally 'causal' region might (1) house the integrator, (2) supply essential inputs to an integrator elsewhere, or (3) affect behaviour independently of the integrator (for example, movement execution). In the context of motor timing, the integrator can be: distributed across the frontal cortex and striatum; redundantly present in both areas; present in one area (specialized); or located in upstream areas (Fig. 1b).

We addressed these gaps with a series of transient perturbations and simultaneous multi-regional electrophysiology. Depending on the role of the manipulated brain area, multi-regional dynamics are expected to respond and recover differently[34,35] (Fig. 1c and Extended Data Fig. 1). For instance, silencing an area with externally driven dynamics will result in a rapid return of ramping dynamics to the original trajectory after the silencing without affecting subsequent dynamics and actions[35] (Fig. 1c (1)). By contrast, silencing an area supplying essential input for an integrator will pause integration in the recipient area, delaying action by the silencing duration (Fig. 1c (2)). Silencing an area serving as an integrator may reset the ramping dynamics, delaying action beyond the silencing duration (Fig. 1c (3)). When inhibition is aligned with the direction of integration, this 'on-manifold' inhibition[33] will be integrated

[1]Max Planck Florida Institute for Neuroscience, Jupiter, FL, USA. [2]Florida Atlantic University, Boca Raton, FL, USA. [3]IMPRS for Synapses and Circuits, Jupiter, FL, USA. [4]National Institute of Mental Health, Bethesda, MD, USA. [5]Turing Centre for Living Systems, Aix-Marseille University, INSERM, INMED U1249, Marseille, France. ✉e-mail: Hidehiko.inagaki@mpfi.org

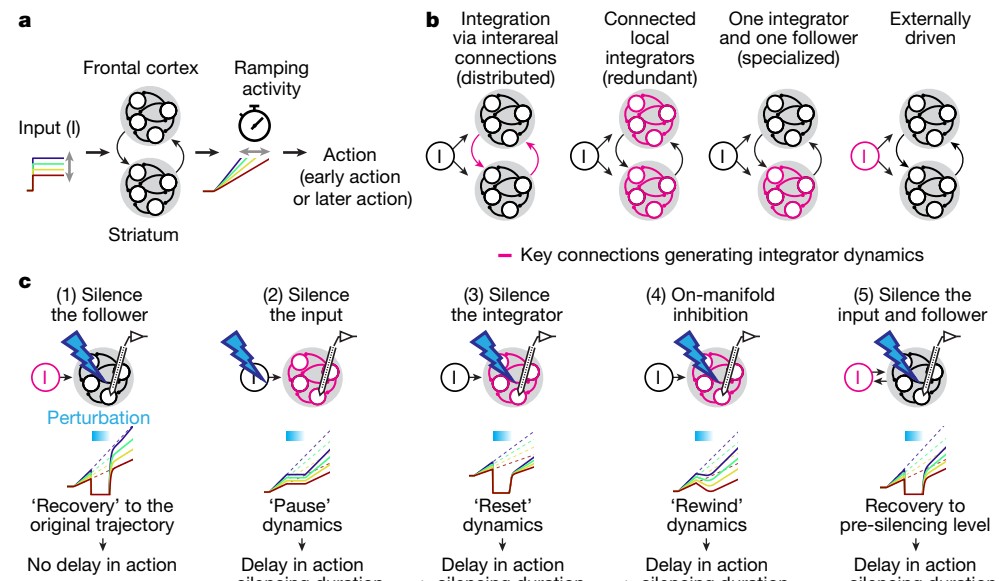

**Fig. 1 | Multi-regional models of flexible motor timing. a**, Schema of the cortico-basal ganglia (striatum) loop computation for motor timing. The network may integrate non-ramping (for example, step) inputs to generate ramping dynamics with variable slopes, producing different lick timings.

Other areas are omitted for simplicity. **b**, Possible configurations of the cortico-striatal network implementing integrator dynamics. Key connections that generate integrator dynamics are in pink. **c**, Schema of perturbation experiments and expected results.

into the ramping dynamics, rewinding the representation of time and delaying action beyond the silencing duration (Fig. 1c (4)). In addition, a brain area may serve multiple functions, such as functioning as both the input and the follower of an integrator (Fig. 1c (5)).

To systematically dissect multi-regional dynamics following this model-driven approach, we developed a flexible lick-timing task, in which mice explored various lick times over 600 trials per session (652 ± 9 trials; mean ± s.e.m.; 48 mice; Fig. 2a), enabling numerous perturbations within single sessions. Large-scale electrophysiology in the frontal cortex and striatum allowed decoding of planned lick time in individual trials, providing an ideal testbed to quantify perturbation effects. Leveraging this system, we identified specialized roles of the frontal cortex and striatum in implementing integrator dynamics, generating ramping activity that functions as an adjustable timer.

## Premotor cortex controls lick timing

In the flexible lick-timing task, an auditory cue (3 kHz, 0.6 s) signals trial onset, followed by a delay epoch of unsignalled duration (1–3 s; Fig. 2a; see Methods). Licks after the delay were rewarded, whereas premature licks during the delay terminated the trial without reward. Delay duration varied across trial blocks (with block length randomly selected from 30 to 70 trials; Fig. 2b). Despite no cue instructing delay duration or block transitions, mice dynamically adjusted their lick-time distribution within approximately 10 trials after the delay switch (Fig. 2b,c; see Supplementary Figs. 3 and 4 for detailed quantifications of behaviour).

The only information available for mice to guide lick time was previous lick times and outcomes. To investigate whether such 'trial history' shapes trial-by-trial lick timing, we exploited a linear regression model to predict lick times[21,36] (*n* = 30 mice; Methods). This analysis revealed upcoming lick times positively correlated with previous lick times, whereas negatively correlated with previous reward outcomes: mice tended to lick earlier after a reward and later after no reward (Fig. 2d). As unrewarded trials reflected premature licks in this task, licking later after an unrewarded attempt is an adaptive strategy. By contrast, when delay duration was constant across trials and sessions ('constant delay condition'; *n* = 13 mice), former trials had no significant influence on lick timing (Fig. 2d). Thus, mice use trial history to strategically adjust lick

timings[22] only when delay duration is variable. We used this behaviour as a model system to examine how the brain flexibly adjusts action timing.

To identify dorsal cortical areas controlling lick timing, we performed optogenetic loss-of-function screening using transgenic mice expressing ChR2 in GABAergic neurons (Vgat-ChR2-eYFP mice) with clear skull preparations[37] (Methods). Dorsal cortical areas were bilaterally silenced during the delay epoch by scanning a blue laser in randomly interleaved trials (488 nm, 1.5 mW; 3 mice; Fig. 2e; see Methods). Lick initiation was blocked when a frontal cortical area, the anterolateral motor cortex (ALM), was silenced, consistent with its role as a premotor cortex for licking[12]. Of note, following transient ALM silencing, licking did not recover immediately; instead, the lick-time distribution shifted significantly later, suggesting a potential role of the ALM in controlling lick timing.

## Similar dynamics in the ALM and striatum

To investigate neural activity underlying lick-time control in the ALM, we conducted high-density silicon probe recordings (4,467 putative pyramidal neurons, 45 mice; Supplementary Table 1). Many ALM neurons displayed ramping activity (Fig. 3a; polynomial fitting showed 59% of neurons peaked firing within 100 ms of cue or lick onset, and 37% of these showed monotonic ramping; Extended Data Fig. 2a–d; see Methods). The ramping speed varied across trials and predicted lick timing (Fig. 3a). Of note, temporal warping[5,9,24,29,38–40], which normalizes the temporal axis between cue and lick, significantly reduced across-trial variability in spike rate in 68.1% of neurons (Fig. 3a and Extended Data Fig. 2e–h). Thus, two-thirds of ALM neurons exhibited temporal scaling (stretching or shrinking) of activity patterns, with the speed of their dynamics anticipating lick time.

At the population level, ALM activity also scaled with lick time (Fig. 3b). Pearson's correlations of population activity across trials with different lick times revealed similar activity patterns unfolded at different speeds depending on lick time (Fig. 3c,d). We also applied targeted dimensionality reduction to define three task-related modes (directions in population activity space; see Methods). Each mode captures activity during a specific task epoch: cue mode reflects the transient cue response (0–300 ms after cue); middle mode captures activity bridging cue and movement preparation (500–800 ms before lick);

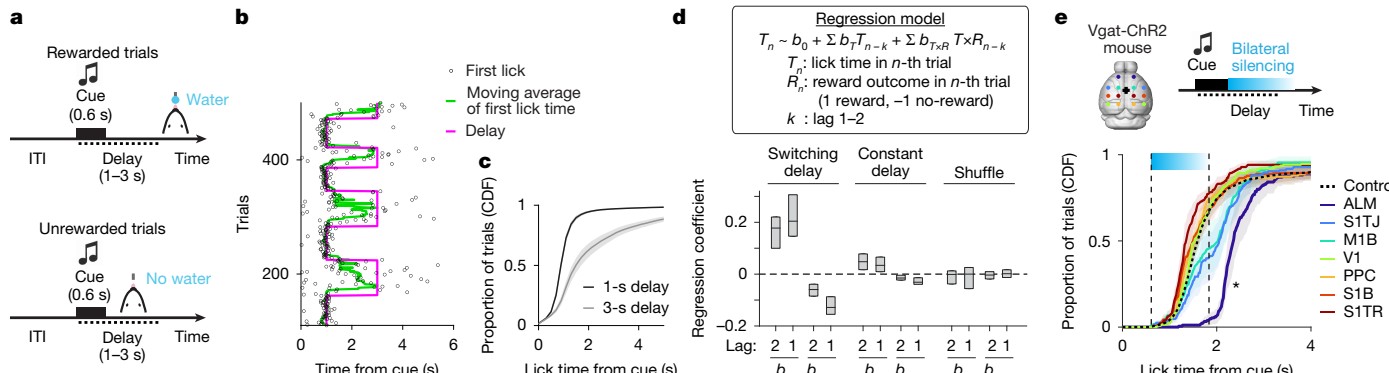

**Fig. 2 | The ALM is required for lick-timing control. a**, Flexible lick-timing task. The delay epoch started at trial onset signalled by the cue. The first lick after the delay was rewarded; premature licks during the delay aborted the trial. **b**, Example session. Only a portion of the session is shown. **c**, Cumulative lick-time distribution in 1-s and 3-s delay blocks (56 sessions, 10 mice). The shading denotes s.e.m. (hierarchical bootstrap). CDF, cumulative distribution function. **d**, Regression coefficients of the regression model based on previous lick time ($T$) and its interaction with reward outcome ($T \times R$), with two-trial lags: switching delay ($n = 30$ mice), and constant delay ($n = 13$ mice). For the boxplot, the central line indicates the median, the box edges denote the 25th–75th percentiles, and the whiskers represent the lowest and highest values within 1.5-times the interquartile range of the lower and upper quartiles. **e**, Optogenetic loss-of-function screening of dorsal cortical areas during delay (top). Cumulative

lick-time distribution in trials with different silenced areas (bottom; 3,879 control trials, 172 ± 5 silencing trials per region; mean ± s.e.m.; 9 sessions, 3 mice). The cyan bar indicates the silencing window (1.2 s). The shading denotes s.e.m. (hierarchical bootstrap). *$P < 0.007$ (hierarchical bootstrap with Bonferroni correction, null hypothesis: control ≥ silencing trials). Regions adjacent to the ALM (M1B and S1TJ) exhibited weaker effects, attributed to the limited spatial resolution of the manipulation[37]. With ALM silencing, median lick time was delayed by 0.79 s (0.59–0.97 s; mean, 95% CI). M1B, body region of primary motor cortex; PPC, posterior parietal cortex; S1B, body region of primary somatosensory cortex; S1TJ, tongue and jaw region of primary somatosensory cortex; S1TR, trunk region of primary somatosensory cortex; V1, primary visual cortex. The brain atlas in panel **e** was adapted from the Allen Institute for Brain Science (https://atlas.brain-map.org).

and ramp mode captures pre-lick activity (200–500 ms before lick), which exhibits a ramping profile. Together, these three modes explained most of the task-modulated activity (74%; Extended Data Fig. 3). Of note, population activity along the middle mode (middle mode activity) and ramp mode (ramp mode activity) displayed temporal scaling (Fig. 3e and Extended Data Fig. 3). Thus, a large proportion of ALM activity between cue and lick, which we refer to as 'timing dynamics', exhibited temporal scaling[5,9,24,29,38–40].

Because major excitatory neurons in the ALM project to the striatum, including the ventrolateral striatum (VLS)[41–43], and the striatum is implicated in timing behaviours[5,7,11,23,24,26–28,38,39], it probably cooperates with the ALM to control lick timing. We recorded striatal activity using Neuropixels probes (1,972 neurons, 16 mice; Fig. 3f). Most neurons were classified as putative striatal projection neurons (SPNs; 64%) or fast-spiking interneurons (30%) based on spike features[44]. Because both showed similar activity patterns (consistent with a previous study[44]), we have pooled them for analysis. Overall, striatal activity resembled that in the ALM[5,11,38,44]. First, many striatal neurons exhibited temporal scaling (56% of cells; Fig. 3f,g and Extended Data Fig. 2g) and ramping activity (67% peak firing within 100 ms at cue or lick onset, and 37% of these showed monotonic ramping; Extended Data Fig. 2a–d; see Methods). Second, striatal population activity (correlation in population activity, middle mode and ramp mode activity) showed similar temporal profiles to that in the ALM with temporal scaling (Fig. 3h–j and Extended Data Fig. 3).

In 19 sessions (14 mice), we recorded the ALM and striatum simultaneously. We applied a $k$-nearest neighbour (kNN) decoder to estimate the remaining time to lick from the ALM or striatum population activity ('$T_{to\,lick}$'; see Methods). Decoded $T_{to\,lick}$ in these two brain areas was significantly correlated across trials, as was ramping activity (Extended Data Fig. 4). Together, the ALM and striatum show similar scalable timing dynamics coupled at the single-trial level.

## Neural correlates of trial history

It remains unclear what determines the speed of timing dynamics after the cue and how these dynamic guide lick timing. Because trial history influences lick time (Fig. 2d), we hypothesized that some neurons

encode trial history before the cue, thereby establishing the initial conditions of the network[4,36] and/or provide inputs[21] to guide timing dynamics and action timing.

Supporting this hypothesis, some ALM neurons exhibited tonic activity during the inter-trial interval (ITI), predicting upcoming lick time even 2 s before cue onset: 22.8% (19.4–26.5%; mean, 95% CI) of neurons exhibited significant rank correlation of spiking activity during the ITI versus upcoming lick time (Extended Data Fig. 2i–k; see Methods). This tonic activity also correlated with previous lick time and reward outcome (Fig. 3k and Extended Data Fig. 2i–m). Because upcoming lick time and trial history are correlated (Fig. 2d), we calculated partial correlation between ALM activity and previous lick time while removing the effect of upcoming lick time (Methods). These values were significantly higher than for trial shuffle and session permutation controls[45] (Extended Data Fig. 2l). Together, ALM neurons encode trial history and anticipate upcoming lick time even before the cue.

If ALM trial-history information guides lick time, such neural correlates may be absent in contexts where trial history is not used. Consistently, ALM activity showed no significant correlation with trial history under the constant delay condition (Extended Data Fig. 2l).

To characterize the evolution of ALM population activity encoding trial history, we defined a 'trial-history mode' by constructing a population vector with the contribution of each neuron weighted by the strength of its correlation with trial history during the ITI (Fig. 3l; see Methods). ALM activity along this mode was modulated by reward outcome (Extended Data Fig. 5a), exhibited a graded persistent activity during the ITI and showed a step-like increase at cue onset (Fig. 3l). Thus, activity along this mode carries trial-history information throughout the trial and predicts upcoming lick time.

Temporal integration of graded activity with varying amplitudes produces a ramp with different slopes (with the cue at trial onset acting as a gate to initiate integration). Indeed, integration of the activity profile of the trial-history mode after the cue generates ramping with different slopes (Fig. 3m), providing a potential mechanism for shaping timing dynamics and lick timing according to trial history ('integrator hypothesis').

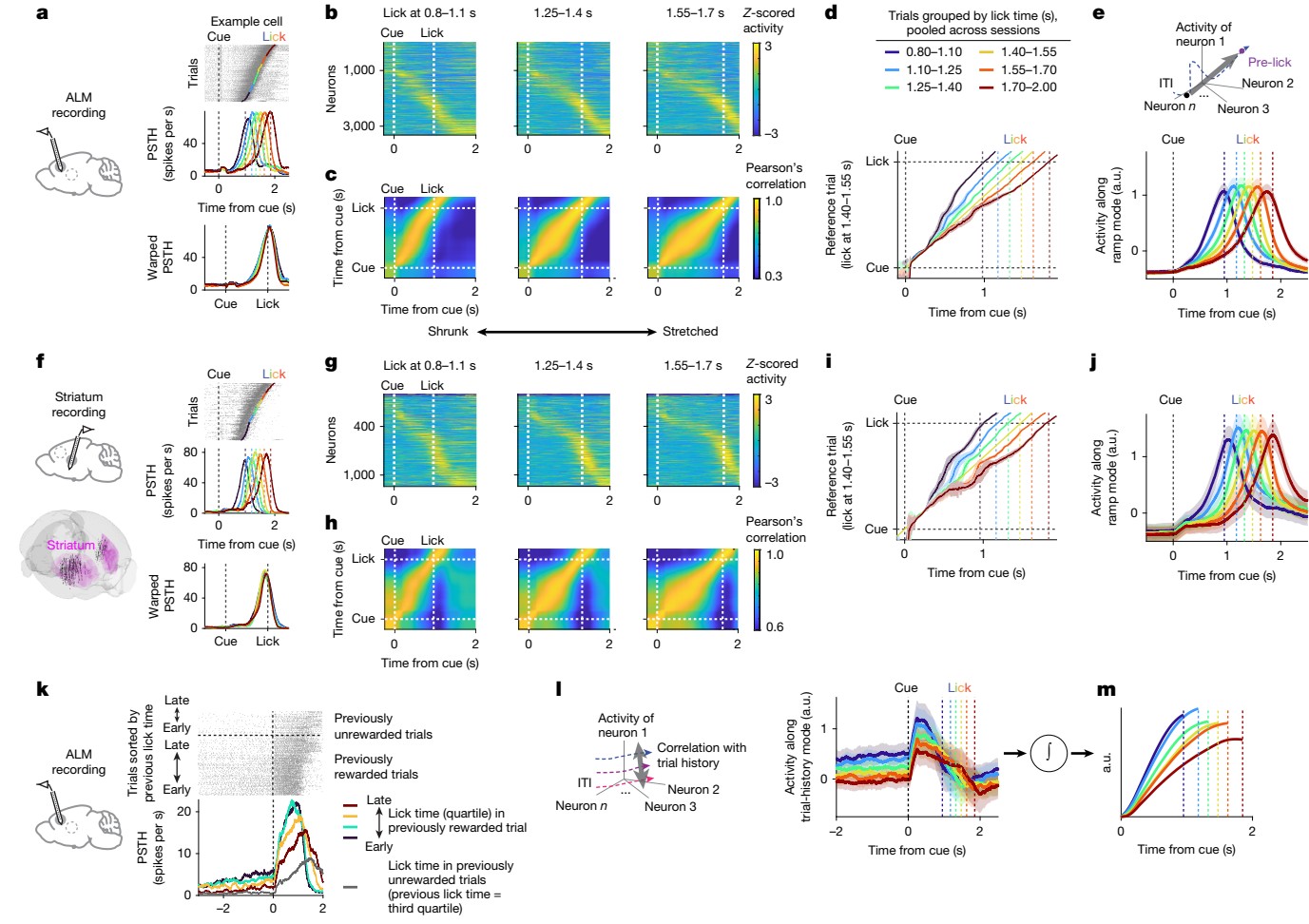

**Fig. 3 | Neural dynamics in the ALM and striatum. a**, An example ALM cell. Spike raster (top right); trials are sorted by lick time. The coloured dots denote first lick (six lick-time ranges shown in panel **d**). Peri-stimulus time histogram (PSTH) of six lick-time groups (middle right). The vertical dotted lines mark lick times. PSTH after temporal warping (bottom right). **b**, ALM Z-scored activity across trials with different lick-time ranges (neurons sorted by peak firing time in 1.40–1.55-s trials; n = 3,261 neurons, 45 mice). Only neurons with 10 or more trials in all ranges included. **c**, Pearson's correlation matrix comparing ALM population activity between reference trials (lick at 1.40–1.55 s) with other trials shown in panel **b**. **d**, Time points with the peak correlation in the correlation matrix (**c**). The lines denote mean, and the shading represents s.e.m. (hierarchical bootstrap). **e**, Schema (top), and ALM population activity projected along the ramp mode, grouped by lick time (bottom; same colour scheme as panel **d**). n = 3,261 neurons, 45 mice. The lines denote grand mean, and the shading indicates s.e.m. (hierarchical bootstrap). a.u., arbitrary units.

**f**–**j**, The same as panels **a**–**e** but for the striatum (n = 1,073 neurons, 16 mice). The unit locations are registered to the Allen Common Coordinate Framework (**f**, bottom). **k**, Example ALM cell modulated by previous lick time and reward. Spike raster grouped by previous reward outcome and sorted by previous lick time (top right). PSTH: previous rewarded trials divided into quartiles (colours; bottom right). Grey denotes unrewarded trials from the third quartile, and orange indicates rewarded trials with the same previous lick times. This comparison isolates reward effects while controlling for previous lick time; the cell shows reduced ITI firing after unrewarded trials. The brain atlas in panels **a**,**f**,**k** was adapted from the Allen Institute for Brain Science (https://atlas.brain-map.org). **l**, Schema of the trial-history mode (left), and ALM population activity projected along the trial-history mode (right). n = 3,261 neurons, 45 mice. The lines denote grand mean, and shading indicates s.e.m. (hierarchical bootstrap). **m**, Temporal integration of activity along the trial-history mode after the cue (based on panel **l**) produces ramps with different slopes.

Consistent with the integrator hypothesis, the amplitude of ALM activity along the trial-history mode highly correlated with the slope of ramping activity in both the ALM and striatum on a trial-by-trial basis (Extended Data Fig. 4f–k). The hypothesis further predicts that modulation of the step-like increase in trial-history mode at cue onset would be integrated into persistent changes in ramping dynamics, thereby influencing lick timing. Indeed, varying cue intensity in randomly interleaved trials altered step amplitude and produced lasting changes in ramping dynamics and lick timing: a fainter cue produced a shallower ramp and delayed licking, and vice versa with a stronger cue (Extended Data Fig. 5e–k; see Methods).

Some striatal neurons also exhibited tonic activity during the ITI, anticipating upcoming lick time, although weaker than ALM (Extended Data Fig. 5a–d). Together, there are robust neural correlates of trial history in the ALM, possibly serving as inputs to the integrator that govern the speed of timing dynamics and lick timing.

## ALM silencing pauses the timer

Neural correlates of temporal integration (that is, trial-history mode and ramp mode) in the ALM are insufficient to conclude that the ALM functions as the integrator. Likewise, although ALM silencing shifted lick time (Fig. 2e), this behavioural effect alone does not attribute a specific computational role (Extended Data Fig. 1). Therefore, to examine whether the ALM serves as an integrator, we recorded ALM activity using silicon probes during calibrated silencing.

Strong cortical silencing caused post-silencing rebound activity that triggers actions[46,47], complicating the interpretation of subsequent

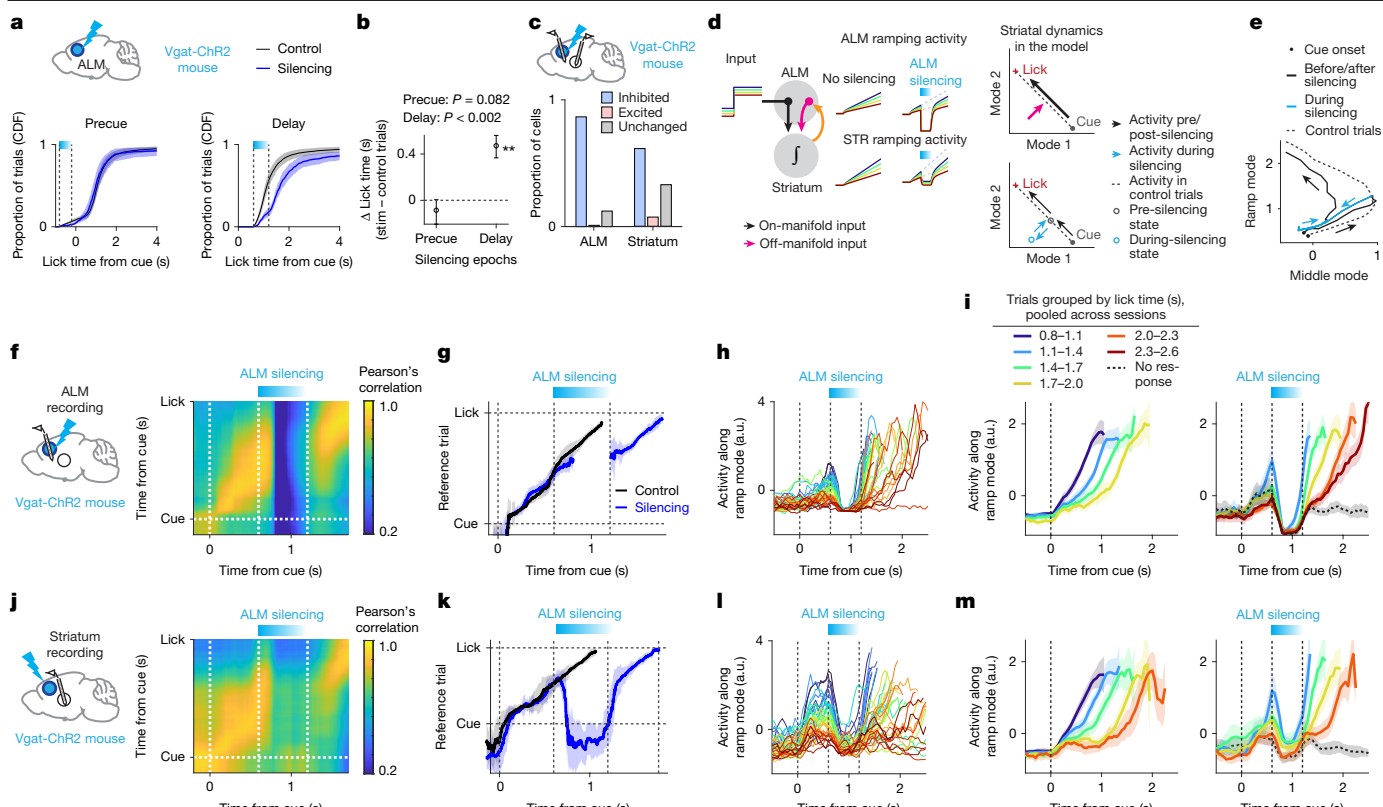

**Fig. 4 | Transient optogenetic perturbation of the ALM. a,** Bilateral ALM silencing in Vgat-ChR2 mouse (top). Cumulative lick-time distribution with precue (bottom left) or delay (bottom right) silencing. The shading denotes 95% CI (hierarchical bootstrap). $n = 28$ sessions, 14 mice. **b,** Change in median lick time in panel **a**. Mean ± 95% CI (hierarchical bootstrap) is shown. $P$ value, hierarchical bootstrap (null hypothesis: no change in lick time from control): **$P < 0.005$. **c,** Proportion of ALM pyramidal neurons (bottom left; $n = 391$ neurons, 14 mice) and striatal neurons (bottom right; $n = 238$ neurons, 7 mice) significantly inhibited or excited ($P < 0.05$; two-sided rank-sum test) during ALM silencing in the delay period. Neurons >1 Hz in controls were analysed. **d,** Schema of a model explaining the data (left); how on-manifold and off-manifold ALM inputs modulate striatal activity (top right); and striatal activity during ALM silencing (bottom right). The orange arrow represents that the ALM receives striatal input to follow ramping dynamics generated there. **e,** Striatal activity in the ramp–middle mode space during ALM delay silencing. The mean trajectory

for trials with lick was 1.7–2.0 s; the arrows show trajectory direction. **f,** ALM recording during ALM silencing (left), and correlation matrix of ALM population activity between ALM silencing trials (lick at 1.7–2.0 s) versus reference trials (unperturbed trials with lick at 1.4–1.7 s; right). **g,** Points with the peak correlation in panel **f** (Methods). The lines indicate mean, and the shading denotes s.e.m. (hierarchical bootstrap). **h,** Example session. The lines indicate individual trials colour-coded by activity level before the silencing onset (blue–red: high–low). **i,** ALM population activity along the ramp mode ($n = 17$ sessions, 14 mice); trials were grouped by lick time across sessions; only ranges present in 2/3 or more of sessions are shown. Activity up to lick is shown. The lines indicate grand mean, and the shading denotes s.e.m. (hierarchical bootstrap). Control (left) and ALM silencing (right) are shown. **j–m,** Same as panels **f–i** but for striatal recording. $n = 12$ sessions, 7 mice. The brain atlas in panels **a,c,f,j** was adapted from the Allen Institute for Brain Science (https://atlas.brain-map.org).

dynamics and behaviour. To minimize this, we calibrated the silencing protocol. Silicon probe recordings confirmed near-complete ALM silencing with 1.5 mW (488-nm laser) bilateral photostimulation in Vgat-ChR2-eYFP mice. Limiting the silencing duration to 0.6 s (including a 0.3-s ramp down) minimized rebound activity and licking, yet rebound increased over sessions, probably due to adaptation[48] (Extended Data Fig. 6ab). Therefore, we restricted analysis to the initial 2 days of ALM silencing (and to the first day for striatal manipulation in Fig. 5; Extended Data Fig. 6).

Silencing the ALM during the delay epoch (0.6 s after cue) with this protocol shifted lick time by 0.47 s (0.37–0.56 s; mean, 95% CI; $n = 14$ mice; Fig. 4a,b; see Supplementary Fig. 7 for comparisons of all manipulation conditions). This shift was close to the silencing duration (0.6 s), as if the 'timer' was paused during ALM silencing. By contrast, silencing the ALM before the cue did not alter lick-time distribution and ALM activity recovered rapidly (Fig. 4a and Extended Data Fig. 7q,r), suggesting no lasting effect of the manipulation, and trial-history information is redundantly maintained across brain areas[49–51].

Recordings of ALM during delay silencing (590 neurons, 14 mice) confirmed near-complete silencing during photostimulation. Once silencing ceased, population activity resembling pre-silencing patterns

rapidly reemerged and ALM dynamics unfolded in parallel with unperturbed conditions (quantified by population correlation; Fig. 4f,g). Consistently, at a single-trial level, ramp mode activity collapsed but rapidly recovered close to the pre-silencing level at the end of ALM silencing, rather than the original trajectory (unlike in Fig. 1c (1)), and resumed ramping at rates similar to those before silencing (Fig. 4h,i and Extended Data Fig. 8a–e).

To assess high-dimensional timing dynamics, we applied a kNN decoder to estimate $T_{to lick}$. ALM population activity before and after silencing significantly predicted lick time on single trials, whereas this predictability was lost during silencing but recovered afterwards (Extended Data Fig. 7c). Comparing silencing and unperturbed trials with matched decoded $T_{to lick}$ at the silencing onset revealed that decoded $T_{to lick}$ diverged during silencing and this offset persisted in parallel afterwards (Extended Data Fig. 7d). Together, following ALM silencing, ALM activity rapidly recovered to the pre-silencing levels, with dynamics unfolded in parallel with unperturbed conditions, explaining the shift in lick time close to the silencing duration (consistent with Fig. 1c (5)).

These results challenge models in which ALM is the sole integrator or purely driven by external inputs (Extended Data Fig. 1). Instead, our data

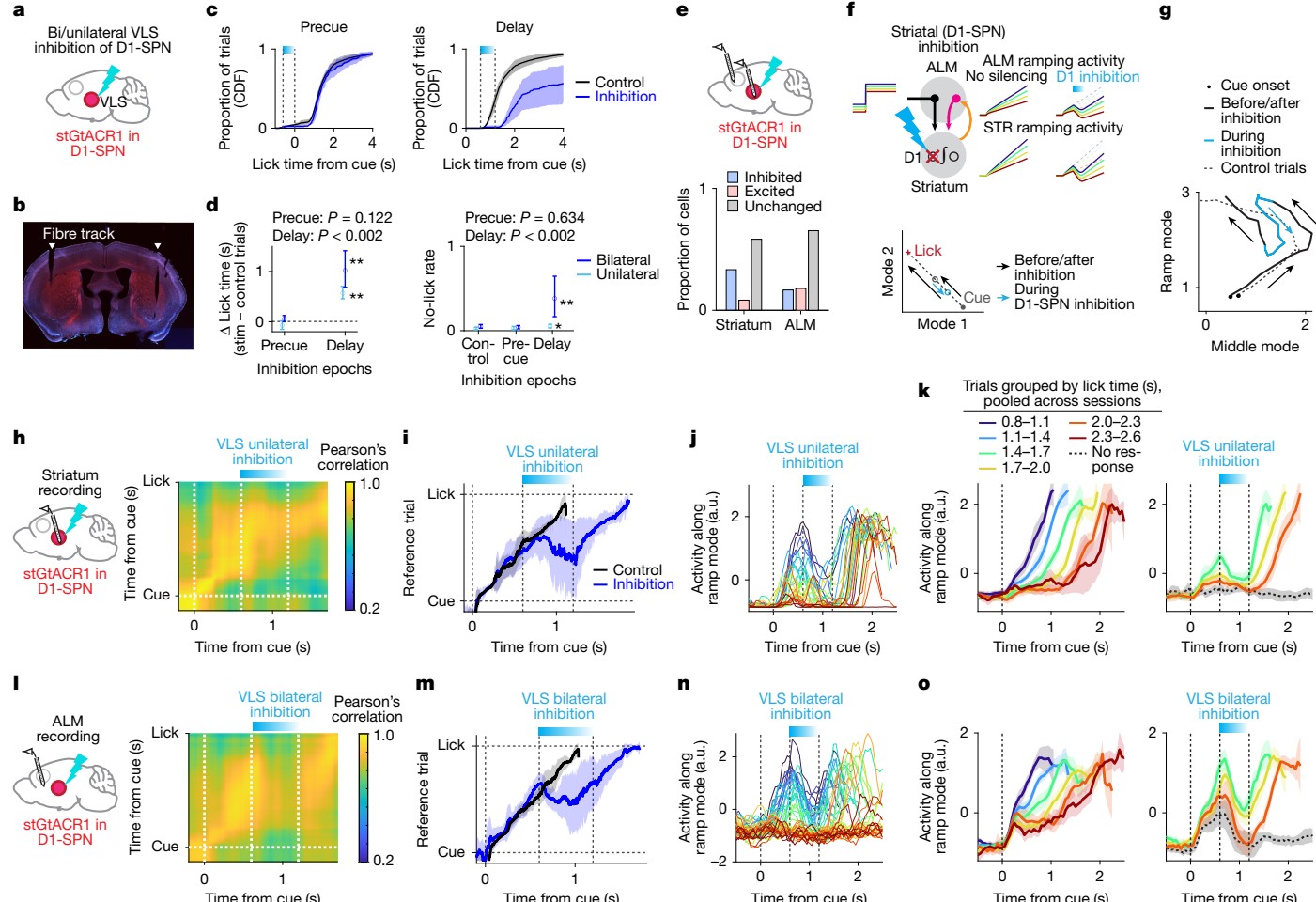

**Fig. 5 | Transient optogenetic perturbation of the striatum. a**, Schema.
**b**, Coronal section of a *Drd1*–Cre;Cre-dependent stGtACR1-fusion-red mouse
with optic fibres in the VLS. Fusion red is in red, and DAPI is in blue. **c**, Cumulative
lick-time distributions with precue (left) or delay (right) inhibition. The shading
denotes 95% CI (hierarchical bootstrap). *n* = 6 sessions, 6 mice. **d**, Change in
median lick time in panel **c** (left), and no-lick rate (right). Data are mean ± 95% CI
(hierarchical bootstrap). *P* value, hierarchical bootstrap (null hypothesis: no
change from control): \**P* < 0.05, \*\**P* < 0.005; *P* values for bilateral perturbations
are shown; *P* = 0.052 (precue) and *P* < 0.002 (delay; left), and *P* = 0.23 (precue)
and *P* = 0.018 (delay; right) for unilateral perturbations. *n* = 6 sessions, 6 mice
(bilateral) and 5 sessions, 5 mice (unilateral). **e**, Proportion of striatal projection
neurons (*n* = 25 neurons, 5 mice) and ALM pyramidal neurons (*n* = 156 neurons,
6 mice) significantly inhibited or excited (*P* < 0.05; two-sided rank-sum test)
during VLS D1-SPN inhibition during the delay. Neurons > 1 Hz in controls were
analysed. **f**, Schema of the model (top) and its dynamics (bottom). **g**, ALM

activity in the ramp–middle mode space during D1-SPN inhibition. The mean
trajectory of silenced trials (lick at 1.7–2.0 s) is shown; the arrows denote
trajectory direction. **h**, Striatal recording during unilateral VLS D1-SPN
inhibition (left), and correlation matrix of inhibition versus reference trials
(lick at 1.4–1.7 s; right). **i**, Points with the peak correlation in panel **h**. The lines
denote mean, and the shading indicates s.e.m. (hierarchical bootstrap).
**j**, Example session. The lines denote individual trials colour-coded by activity
level before the silencing (blue–red: high–low). **k**, Striatal population activity
along the ramp mode (*n* = 10 sessions, 5 mice), with trials grouped by lick time;
only ranges present in 2/3 or more of sessions are shown. The lines denote
grand mean, and the shading indicates s.e.m. (hierarchical bootstrap). Control
(left) and inhibition (right) are shown. **l**–**o**, Same as in panels **h**–**k** but for ALM
recording during bilateral VLS D1-SPN inhibition. *n* = 6 sessions, 6 mice. The
brain atlas in panels **a**,**e**,**h**,**l** was adapted from the Allen Institute for Brain
Science (https://atlas.brain-map.org).

suggest that ALM silencing momentarily pauses temporal integration
because the ALM provides input to an integrator ('timer') elsewhere.
In addition, the rapid recovery of ALM activity to pre-perturbation
levels implies that ALM dynamics follow the external integrator, which
was paused during ALM silencing. Therefore, the ALM may act as both
an input to and a follower of the integrator (Fig. 1c (5) and Extended
Data Fig. 1f).

## Striatal dynamics during ALM silencing

During ALM silencing, other brain areas must retain the timing informa-
tion to restore ALM dynamics. Given the prominent timing dynamics in
the striatum (Fig. 3), we tested whether the striatum has this role. We
recorded striatal activity during ALM silencing (372 neurons, 7 mice).

A majority of striatal neurons (60%) decreased spike rates during ALM
silencing, consistent with the ALM providing major excitatory drive
(Fig. 4c). Consistently, striatal ramp mode activity decayed rapidly dur-
ing ALM silencing (Fig. 4j–m). After silencing, striatal activity recovered
to near pre-silencing levels and unfolded in parallel with unperturbed
conditions (Fig. 4j–m), similar to the ALM. However, striatal activity
was not entirely abolished during ALM silencing (Fig. 4c,l,m). This
residual activity preserved rank order and predicted lick time at the
single-trial level (Extended Data Figs. 7 and 8). Thus, despite reduced
mean activity, the striatum retained timing information during ALM
silencing.

Striatal activity remained low and did not ramp during ALM silencing
(Fig. 4l,m), suggesting that the ALM provides essential input for ramp-
ing activity. The observed multi-regional dynamics support a model

in which the striatum (and/or subcortical areas situated between the striatum and the ALM, such as the substantia nigra reticulata and the thalamus) functions as an integrator, whereas the ALM acts as both input and follower of this 'subcortical integrator' (Fig. 4d (left) and Extended Data Fig. 1f). In this model, the ALM inputs to the striatum consist of two components:

(1) On-manifold input aligned with the direction of integration in striatal state space. This input is temporally integrated by the subcortical integrator to generate scalable timing dynamics (Fig. 4d (left and top right), black arrows). Trial-history mode activity may have this role (Fig. 3l,m). During ALM silencing, loss of this input pauses integration and the representation of time.

(2) Off-manifold input orthogonal to the integration direction provides excitatory drive that amplifies striatal activity without affecting time representation (Fig. 4d (left and top right), pink arrows). ALM silencing removes this drive, reducing mean striatal activity.

Together, loss of both inputs during ALM silencing pauses time representation at a reduced activity level in the striatum (Fig. 4d (bottom right), cyan circle). Once ALM silencing ends, excitatory drive returns and the striatal activity recovers to pre-silencing levels; meanwhile, the restoration of on-manifold input enables timing dynamics to resume along a normal trajectory, producing the parallel shift in dynamics (Fig. 4d,e and Extended Data Fig. 1f).

Temporal integration can be achieved through feedforward networks[52,53], as well as positive-feedback loops (Extended Data Fig. 1). Recurrent network models with feedforward connections produce both sequential and ramping activity, as observed in the data (Extended Data Fig. 9a–c). We modelled the ALM as the input and follower to a subcortical feedforward network; in this configuration, ALM perturbation reproduced the pause in time representation (but not in alternative models; Extended Data Fig. 9d–i). Therefore, regardless of the implementation, our data support models in which the ALM provides essential input to a subcortical integrator representing time. The key assumption of these models is that the striatum (and/or intermediate subcortical areas) serves as the integrator. To test this, we next perturbed striatal activity.

## Striatal inhibition rewinds the timer

The striatum contains two major projection cell types: D1 receptor-expressing direct pathway SPN (D1-SPN) and D2 receptor-expressing indirect pathway SPN (D2-SPN)[54]. Consistent with the anti-kinetic function of D2-SPN[54,55], inhibiting D2-SPN or both SPN types with the soma-targeted light-dependent chloride channel, stGtACR1, triggered licking during photostimulation (Supplementary Fig. 7), making these methods unsuitable for testing lick timing. We therefore focused on inhibiting D1-SPNs.

To inhibit D1-SPN, we crossed *Drd1*–Cre FK150 mice with Cre-dependent stGtACR1 mice, and bilaterally implanted tapered fibre optics in the striatum (Fig. 5a,b). Optrode recordings (488 nm, 0.25 mW) confirmed inhibition: 9 out of 25 SPNs (36%) significantly reduced spike rates without axonal excitation[47] or post-silencing rebound (Fig. 5e and Extended Data Fig. 6d,e). SPN types cannot be distinguished from spike features[43], but as approximately 50% are D1-SPN[54], we estimated that approximately 70% of D1-SPN near the fibres were inhibited.

Transient bilateral inhibition of D1-SPN in the VLS during the delay epoch (0.6-s duration, starting 0.6 s after the cue; 488 nm, 0.25–0.5 mW) increased the no-lick rate by 38% (5.7–41%; mean, 95% CI; $n = 6$ mice; Fig. 5c,d). In trials with licks, lick time was shifted later by 1.0 s (0.69–1.4 s; mean, 95% CI), significantly longer than the photostimulation duration and the effect of ALM silencing.

Unilateral inhibition produced approximately half the effect of bilateral inhibition, suggesting additive effects (Fig. 5c,d). Inhibition before the cue had no effect on lick-time distribution, implying no long-lasting

effect and that the contribution of the striatum is specifically after the cue (Fig. 5c,d). The behavioural effect of inhibition was stronger in the VLS than in the dorsomedial striatum (Supplementary Fig. 7), consistent with strong ALM–VLS connectivity[41–43].

To measure the effect of D1-SPN inhibition on striatal dynamics, we performed optrode recordings (103 neurons, 5 mice). During unilateral inhibition, population activity patterns stopped unfolding, showed a slight recession in the peak correlation points, without deviating from normal activity patterns – that is, remained 'on-manifold'[33] (Fig. 5h,i). After inhibition, activity developed from the post-inhibition state in parallel with the unperturbed condition (Fig. 5i).

Unlike ALM silencing, which rapidly decreased striatal activity at the stimulation onset (Fig. 4l,m), striatal ramp mode activity and decoded $T_{to\,lick}$ gradually decayed during photoinhibition (Fig. 5j,k and Extended Data Fig. 7l). Because significantly photoinhibited cells (putative D1-SPN expressing stGtACR1) were silenced within 50 ms (Supplementary Fig. 8l), the gradual decay is unlikely due to slow photoinhibition but reflects network effects (for example, D1-SPN modulates the thalamus via the substantia nigra reticulata, which projects back to the striatum). Unlike ALM silencing, after inhibition, the ramp restarted from the post-inhibition level without a rapid recovery (Fig. 5j,k and Extended Data Fig. 7i). Hence, striatal timing dynamics slowly rewind during unilateral VLS D1-SPN inhibition, as if the D1-SPN inhibition is integrated into the timing dynamics. This effect is consistent with D1-SPN being part of, or providing on-manifold input to, the integrator (Fig. 1c (4)).

## The striatum supports ALM timing dynamics

In our network model (Fig. 5f and Extended Data Figs. 1f and 9i), ALM timing dynamics follow those generated by the subcortical integrator. If so, VLS D1-SPN inhibition should also decay the ALM timing dynamics. To test this, we recorded ALM activity during bilateral VLS D1-SPN inhibition (255 neurons, 6 mice). Inhibiting VLS D1-SPN during the ITI reduced ALM activity just by 0.17 ± 0.14 spikes per second (mean ± s.e.m.). During delay inhibition, only 16.7% and 18.0% of ALM neurons were significantly inhibited and excited, respectively (Fig. 5e). Thus, VLS D1-SPN is not the major excitatory drive of the ALM.

However, inhibiting VLS D1-SPN during the delay exerted specific effects on ALM timing dynamics. During VLS D1-SPN inhibition, ALM population activity patterns paused their progression and slightly receded without deviating from normal activity patterns (Fig. 5l,m), suggesting an on-manifold perturbation.

ALM ramp mode activity and decoded $T_{to\,lick}$ gradually decayed during VLS D1-SPN inhibition but resumed ramping in near-parallel with controls after inhibition ended, without rapid recovery (Fig. 5n,o and Extended Data Figs. 7 and 8). Consistently, the recovery of $T_{to\,lick}$ to the pre-perturbation level was significantly slower than that after ALM silencing (Extended Data Fig. 7u). This decay in timing dynamics explains both the extended lick-time shifts and the increased no-lick trials (Extended Data Fig. 10 and Supplementary Discussion). Consistent with 'rewinding', ALM dynamics in the two-dimensional space defined by ramp mode and middle mode tended to evolve opposing to the normal trajectory during VLS D1-SPN inhibition (Fig. 5g), unlike ALM silencing, which drove activity towards zero (Fig. 4e and Extended Data Fig. 11). Thus, although D1-SPN is not a major excitatory drive of ALM activity, it strongly influences ALM timing dynamics.

The effect of ALM and VLS inhibition differed qualitatively. Even weaker ALM inhibition (0.3 mW) caused a weak yet rapid decay in ramp mode activity at photostimulation onset, followed by a recovery of ramping during photostimulation (Extended Data Fig. 12c,d) and a mild behavioural effect (Extended Data Fig. 12a,b). Thus, the gradual decay in ALM timing dynamics during D1-SPN inhibition cannot be explained by weak inhibition. Moreover, although longer perturbations in D1-SPNs and the ALM both produced larger lick-time shifts, only prolonged D1-SPN inhibition increased the no-lick rate (Extended Data Fig. 10).

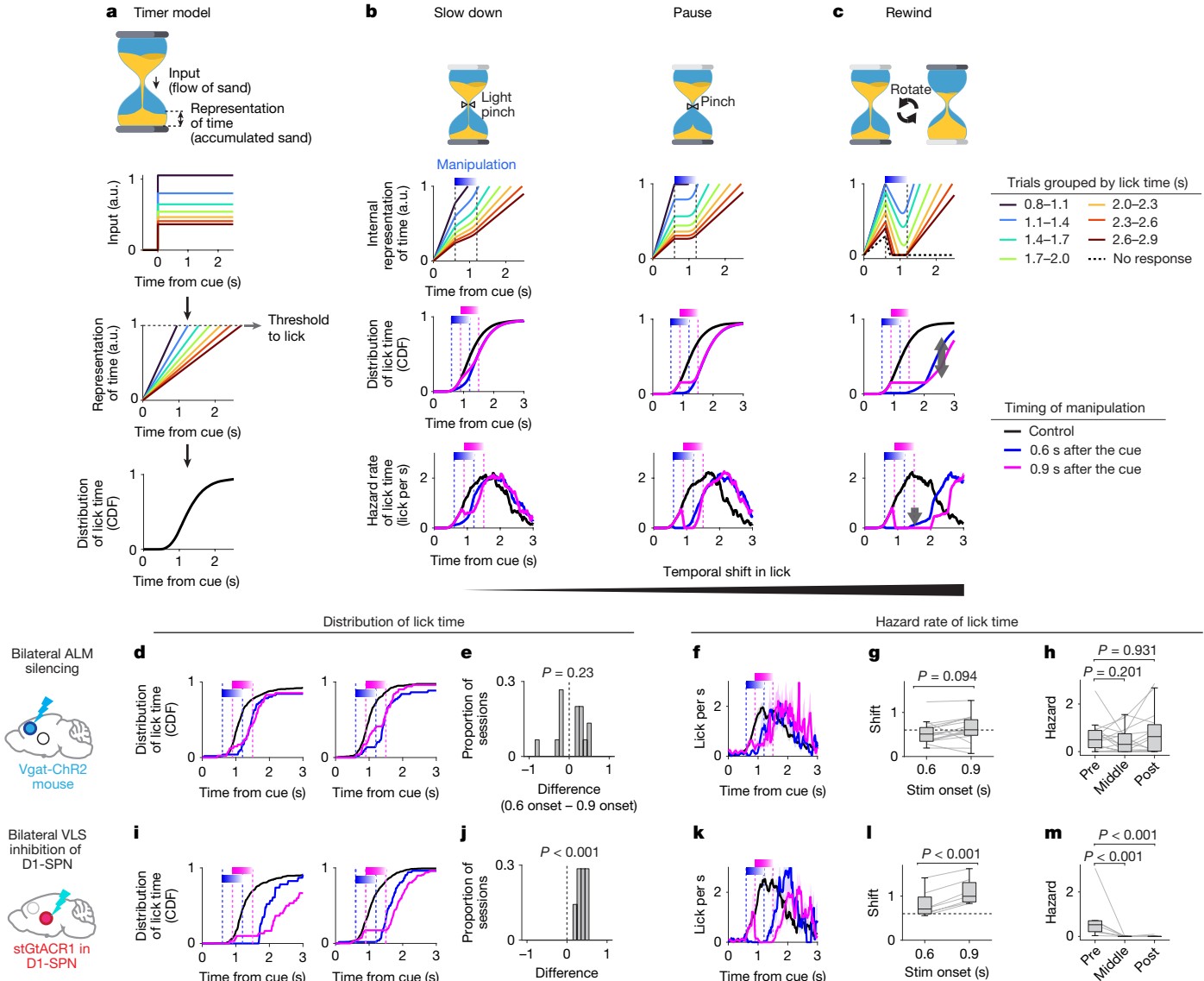

**Fig. 6 | Modelling and testing timer perturbations at different onsets. a**, An hourglass analogy to model timing behaviour. **b**,**c**, Transient manipulation of the timer. Schema (first row), internal time representation (trials with 0.6-s onset; second row), simulated cumulative lick-time distributions (third row; blue and purple, manipulation at 0.6 s and 0.9 s after the cue, respectively), the hazard rate of lick-time distribution (fourth row) are shown. With a 'pause or slowdown', lick-time distributions and the hazard rate shift similarly across onsets, and the hazard rate remaining at more than 0 at the end of manipulation. With a 'rewind', distributions shift differently depending on onset, and the hazard rate dropped to 0 during and after manipulation (black arrows). **d**, Cumulative lick-time distributions with ALM silencing at different onsets (example sessions). **e**, Difference in lick-time distributions (maximum vertical difference) between trials with different ALM silencing onsets (0.6 s versus 0.9 s). *P* values: bootstrap (null hypothesis: no difference). Across sessions, 3 of 15 had

significant positive differences and 1 of 15 had negative differences (Kolmogorov–Smirnov test, *P* < 0.05). **f**, Hazard rate across sessions. *n* = 15 sessions, 8 mice (same for panels **g**,**h**). The line denotes mean, and the shading indicates 95% CI. **g**, Temporal shift in hazard rate between onsets. Data are mean ± 95% CI. *P* values: bootstrap (null hypothesis: no difference). The lines denote individual sessions, and the dotted line indicates manipulation duration. For the boxplot, the line denotes the median, the box indicates 25–75%, and the whiskers show lowest and highest values within 1.5 times the interquartile range of the lower and upper quartile. **h**, Hazard rate before (pre), during (middle; 0.3 s after onset) and after (post) manipulation for 0.9-s onset trials. *P* value: bootstrap (null hypothesis: no difference). Format is as in panel **g**. **i–m**, Same as panels **d–h** but for D1-SPN inhibition. In panel **j**, 6 of 7 sessions showed significant positive differences (Kolmogorov–Smirnov test; *P* < 0.05). *n* = 7 sessions, 7 mice.

To further examine qualitative differences between the two manipulations, we modelled a 'timer' as a generic accumulator (for example, an hourglass; Fig. 6a and Supplementary Discussion). Here a pause or slowdown corresponds to reducing sand inflow, causing the ramp to stop or slowdown but resume after manipulation. This produces an equal-magnitude, parallel shift in the lick-time distribution and its hazard rate (moment-by-moment likelihood of licking given it has not yet occurred; that is, instantaneous drive to lick), regardless of manipulation onset (that is, 'state-independent'; Fig. 6b). By contrast,

a rewind corresponds to flipping the hourglass, lowering the sand level during the manipulation and resuming from a reduced state. This reduces the hazard rate at manipulation offset (as the state of the timer is rewound), and produces large, state-dependent shifts in the lick-time distribution due to a floor effect (when the sand runs out; arrows in Fig. 6c). Consistently, ALM silencing at different onsets during delay produced state-independent shifts with full hazard rate recovery (Fig. 6d–h), whereas D1-SPN inhibition produced larger, state-dependent shifts and drove the hazard rate to zero at inhibition

offset (Fig. 6i–m). These findings support that inhibiting the ALM and D1-SPNs corresponds to pause or slowdown and rewind of the 'timer', respectively.

Together, VLS D1-SPN inhibition produced a stronger behavioural effect than ALM silencing, regardless of onset or duration (Fig. 6 and Extended Data Fig. 10), despite its weaker effect on mean spike rates. This strong behavioural effect probably reflects its on-manifold effect on timing dynamics. The absence of rapid recovery after striatal inhibition suggests no independent timer elsewhere restoring the activity. These results support a model in which the striatum (and/or intermediate subcortical areas) implements an integrator generating timing dynamics, with ALM timing dynamics reflecting those generated by this subcortical integrator (Fig. 5f).

## Discussion

The frontal cortex and striatum often exhibit similar activity patterns and are essential for motor timing and other behaviours[2,3,12,56], posing a challenge in disentangling their roles. To address this, we conducted a series of transient perturbations coupled with multi-regional electrophysiology. All manipulations temporally shifted subsequent timing dynamics, which then evolved in near-parallel with unperturbed conditions and affected lick time. Furthermore, the extent of the shift scaled with perturbation strength (unilateral versus bilateral, duration and laser power) as if perturbations were integrated into the timing dynamics. Across conditions, both ALM and striatal population activity continued to predict lick timing even after perturbations, collectively indicating a tight causal link between dynamics in these areas and motor timing.

Perturbation effects differed qualitatively depending on the manipulated brain areas: silencing the ALM effectively paused the timer without erasing timing information in the striatum, whereas striatal inhibition appeared to rewind the timer in both areas. These findings support a model in which the striatum, potentially along with other subcortical areas, functions as an integrator generating timing dynamics in response to ALM inputs (Fig. 5f). In this model, trial-history information is persistently encoded in the ALM (Fig. 3) and serves as input to the subcortical integrator, determining the slope of ramping activity, thereby adjusting lick timing based on previous trials. The resulting ramping activity is relayed back to the ALM via a multi-synaptic pathway, explaining similar ramping dynamics across brain areas and probably enabling the ALM to trigger a timed lick[12] (ALM ramping is orthogonal to the subcortical integrator, preventing runaway excitation; Fig. 4d).

In dynamical systems, the evolution of activity states is shaped by both the initial conditions and the external inputs[4,21,36]. Models solely based on initial conditions account for temporal scaling[4,5,57], but require additional mechanisms to explain recovery after ALM silencing (Supplementary Discussion). By contrast, the integrator model naturally explains both scaling and recovery, thus, offering a more parsimonious explanation, although initial conditions may also contribute.

We propose that cortical inputs controlling a subcortical integrator provide a general mechanism for temporal integration across motor and cognitive behaviours. Supporting this view, singing mice adjust song durations based on social context, and this context-dependent modulation requires the frontal cortex[58]. Moreover, during decision making, ramping activity associated with evidence accumulation is observed in both the frontal cortex and the striatum, with the striatum implicated as a key integrator[30,31].

Neural correlates and behavioural effects of manipulations alone cannot distinguish multi-regional dynamics models (Extended Data Figs. 1 and 9). Prolonged manipulations (for example, muscimol infusion) particularly have limited ability (Extended Data Fig. 1). Therefore, transient perturbations combined with large-scale electrophysiology are critical[34,35]. Stringent consideration of behavioural adaptation[48] and

calibration of photostimulation conditions to prevent rebound are crucial for interpretable and reproducible perturbation experiments (Extended Data Fig. 6).

In a delayed response task with a cue signalling lick time, transient ALM silencing was followed by a rapid recovery of activity to the original trajectory (akin to Fig. 1c (1)), regardless of inhibition strength[35]. By contrast, identical ALM manipulation in the timing task produced temporal shifts proportional to inhibition strength. Moreover, the ALM encoded trial history only when behaviourally relevant (Extended Data Fig. 5). Thus, ALM dynamics operate under distinct regimes optimized for task demands.

Because the striatum consists predominantly of inhibitory neurons with sparse lateral connectivity[54], D1-SPN alone is unlikely to implement an integrator. The VLS receives excitatory input from the ALM, neighbouring cortical areas and the thalamus[41,42]. Cortical optogenetic screening and ALM silencing (Figs. 2 and 4) suggest that the cortex provides input but is not the integration site. Instead, integration may emerge from a subcortical loop involving the striatum, substantia nigra reticulata and thalamus, forming a disinhibitory loop[41,42,54,59,60]. Alternatively, D1-SPN and D2-SPN may exert a push–pull control over downstream integrators, although weak trial-history activity in the striatum argues against this idea (Extended Data Fig. 5a–d). Mutual inhibition between direct and indirect pathways or in downstream regions[61] may also contribute. Together, we identified the VLS as a key contributor to integration. Further perturbation experiments in the cortico-basal ganglia loop will be essential to fully elucidate the implementation of the integrator.

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

## Method

### Experimental model and participant details

**Mice.** This study is based on both adult male and female mice (aged > P60). We used five mouse lines: C57BL/6 J (JAX# 000664), VGAT-ChR2-eYFP[62] (JAX #14548), *Drd1–cre* FK150 (ref. 63), *Adora2–cre* KG126 (ref. 63) and R26-LNL-GtACR1-Fred-Kv2.1 (ref. 47) (JAX #33089). See Supplementary Table 1 for mice used in each experiment.

All procedures were in accordance with protocols approved by the MPFI IACUC committee. We followed the published water restriction protocol[64]. Mice were housed in a 12–12 reverse light–dark cycle and behaviourally tested during the dark phase. Ambient temperature was 74 °F and humidity ranged between 35% and 60%. A typical behavioural session lasts between 1 h and 2 h. Mice obtained all of their water in the behaviour apparatus (approximately 0.6 ml per day). Mice were implanted with a titanium headpost for head fixation[64] and single housed. For cortical photoinhibition, mice were implanted with a clear skull cap[37]. For bilateral D1/D2-SPN silencing, tapered fibre optics[65] (1.0-mm taper, NA 0.37 and core diameter of 200 μm, Doric lenses) were bilaterally implanted during the headpost surgery around the following target coordinates (Bregma): anteroposterior −0.3 mm, mediolateral ±3 mm and dorsoventral 3.5 mm for the VLS; and anteroposterior 0.6 mm, mediolateral ±1.5 mm and dorsoventral 3 mm for the dorsal medial striatum. Craniotomies for recording were made after behavioural training.

**Viral injection.** To virally express stGtACR1 (ref. 66) in the striatum, we followed published protocols[67] for virus injection. AAV2/5 CamKII-stGtACR1-FusionRed (titre: $9.5 \times 10^{12}$) was injected into anteroposterior −0.3 mm, mediolateral 3 mm, dorsoventral 2.75 and 3.5 mm, 100 nl each depth. The same tapered fibre optics described above were bilaterally implanted at dorsoventral 3.5 mm.

**Behaviour.** At the beginning of each trial, an auditory cue was presented, which consisted of three repeats of pure tones (3 kHz, 150-ms duration with 100-ms inter-tone intervals, 74 dB). A delay epoch started from the onset of the cue presentation. Licking during the delay epoch aborted the trial without a water reward, followed by a 1.5-s timeout epoch. Licking during the 10-s answer epoch following the delay was considered a 'correct lick', and a water reward (approximately 2 μl per drop) was delivered immediately, followed by a 1.5-s consumption epoch. If mice did not lick during the 10-s answer period, the trial would end without a reward. Trials were separated by an ITI randomly sampled from an exponential distribution with a mean of 3 s, with 1-s offset (with a maximum ITI of 7 s). This prevented mice from predicting the trial onset without a cue. Animals had to withhold licking during the full ITI epoch for the next trial to begin (otherwise, the ITI epoch repeated). In approximately 10% of randomly interleaved trials, the auditory cue was omitted to assess spontaneous lick rate ('no-cue' trials). No water reward was delivered in no-cue trials.

We followed the protocol described in Majumder et al.[68] for training. In brief, the delay duration increased from 0.1 s to 1.8 s gradually based on the performance of the animal[68]. Once mice reached 1.8-s delay, we started either the switching delay, the random delay or the constant delay conditions (see Supplementary Fig. 3 for example sessions). In the switching delay condition, we switched the delay between 1 s versus 3 s or 1 versus 1.8 s every 30–70 trials (the number of trials was randomly selected from 30 to 70 and not contingent upon behaviour). Similarly, in the random delay condition, we randomly switched the delay among 0.5, 1.0, 1.5, 2.0, 3.0 or 5.0 s every 30–70 trials. For the constant delay condition, mice were trained with a constant delay of 1.5 s across sessions for at least 2 weeks. For the cue-intensity experiments (Extended Data Fig. 5), we changed the cue intensity (3-kHz auditory cue, ±15 dB, lasting 0.6 s) in randomly interleaved test trials (approximately 20%). Except for this modification, the task structure was identical.

Cue intensity stayed constant (74 dB) before the cue-intensity experiments. Otherwise, the task design and reward contingency remained the same. ALM and striatal perturbation experiments (Figs. 4 and 5) were performed under the switching delay condition. To avoid human bias, the behaviour was automatically controlled by Bpod (Sanworks) and custom MATLAB codes.

**Optogenetics.** Photostimulation was deployed on less than 25% in randomly selected trials. To prevent mice from distinguishing photostimulation trials from control trials using visual cues, a 'masking flash' (1-ms pulses at 10 Hz) was delivered using 470-nm LEDs (Luxeon Star) throughout the trial. For both ChR2 and stGtACR1, we used a 488-nm laser (OBIS 488–150C, Coherent).

The ChR2-assisted photoinhibition of the dorsal cortices was performed through clear-skull cap[37] (Fig. 2e) or craniotomy (in case of simultaneous recording; Fig. 4). We scanned the 488-nm laser light using Galvo mirrors. We stimulated GABAergic interneurons in Vgat-ChR2-eYFP mice starting at 0.6 s after the cue, lasting for 1.2 s (including 0.2-s ramping down; Fig. 2e) or 0.6-s duration (including 0.3-s ramping down; Fig. 4). Time-averaged laser power was 1.5 mW per spot (or 0.3 mW per spot for Extended Data Fig. 12; 8 spots in total: 4 spots in each hemisphere centred around the target coordinates with 1-mm intervals; we photoinhibited each spot sequentially at the rate of 5 ms per step). For Fig. 2e, the targeted brain area was randomly selected for each photostimulation trial. The target coordinates were anteroposterior 2.5 mm and mediolateral ±1.5 mm for the ALM; anteroposterior 0.5 mm and mediolateral ±1.5 mm for M1B; anteroposterior 0.5 mm and mediolateral ±2.5 mm for S1TJ; anteroposterior −1.0 mm and mediolateral ±1.5 mm for S1TR; anteroposterior −1.0 mm and mediolateral ±3.0 mm for S1B; anteroposterior −2 mm and mediolateral ±1.5 mm for PPC; and anteroposterior −2.5 mm and mediolateral ±3.5 mm for V1, respectively (Bregma).

To silence D1-SPNs using stGtACR1 (Fig. 5), we delivered photostimuli (0.25 mW or 0.5 mW, 488 nm) bilaterally (Fig. 5l–o) or unilaterally (in case of optrode; Fig. 5h–k) in the striatum starting 0.6 s after the cue and lasting for 0.6 s (including 0.3-s ramping down). In precue inhibition trials, photostimuli were delivered 0.81 s, 0.6 s before the cue for the ALM, D1-SPN perturbation, respectively, both lasting for 0.6 s. The light was delivered through implanted fibre optics, and intensity was measured at the fibre tip.

**Extracellular electrophysiology.** A small craniotomy (diameter of 0.5–1 mm) was made over the recording sites 1 day before the first recording session. Extracellular spikes were recorded acutely using 64-channel two-shank silicon probes (H-2, Cambridge Neurotech) for the ALM and Neuropixels probe 1.0 (ref. 69) for the striatum. For the H-2 probes, voltage signals were multiplexed, recorded on a PCI6133 board (National Instruments) and digitized at 400 kHz (14-bit). All recordings were made with the open-source software SpikeGLX (http://billkarsh.github.io/SpikeGLX/). During recordings, the craniotomy was immersed in a cortex buffer (125 mM NaCl, 5 mM KCl, 10 mM glucose, 10 mM HEPES, 2 mM $MgSO_4$ and 2 mM $CaCl_2$; adjusted pH to 7.4). Brain tissue was allowed to settle for at least 5 min before recordings.

For the optrode recordings (Fig. 5h–k), we used 64-channel two-shank silicon optrodes with a 1.0-mm taper fibre optic attached adjacently (NA 0.22, core diameter of 200 μm; Cambridge Neurotech). Optrode was acutely inserted in each session and the light delivery protocol was identical to that used for behavioural experiments described in the section 'Optogenetics'. Neuropixels probe and optrode tracks labelled with CM-DiI were used to determine recording locations[70].

**Histology.** Mice were perfused transcardially with PBS, followed by 4% paraformaldehyde/0.1 M PBS. To reconstruct recording tracks, we either generated coronal sections followed by conventional imaging (protocol described in Inagaki et al.[71]) or cleared the brain followed

by light-sheet microscopy. To clear the brain, we used the EZ Clear method[72]. We followed the previous protocol to map the recording tracks to the Allen Common Coordinate Framework[70,73].

### Quantification and statistical analysis

**Behavioural analysis.** We analysed the time of the first lick after the cue onset in each trial. Lick time was measured by detecting the contact of the tongue with the lick port using an electrical lick detector. For optogenetic experiments, we analysed trials with the first lick occurring after the onset time of photostimulation (0.6 s after the cue) in both control and photostimulated trials to compare the effect of photostimulation on behaviour. The no-lick rate was calculated as the probability of mice not responding within 5 s after the cue. The shift in lick time (Δlick time) was based on the median lick time. The post-stimulation lick rate (Extended Data Fig. 6) was calculated as the probability of mice licking within 0.6 s after the photostimulation offset time in no-cue trials. To analyse behaviour while the mice were engaged in the task, we analysed all trials between the first occurrence of five consecutive cue trials with licks and 20 trials before the last occurrence of three consecutive no-lick trials without photostimulation.

Owing to the attenuation of behavioral effects of optogenetic manipulation (Extended Data Fig. 6), we restricted analyses of both behavioural and physiological data to the first (for striatal manipulation) or the first two (for ALM manipulation) manipulation sessions per mouse. All analyses, including the calculation of confidence intervals and $P$ values, were performed using a hierarchical bootstrap, unless stated otherwise. First, we randomly selected animals with replacements. Second, we randomly selected sessions for each animal with replacement. Third, we randomly selected trials for each session with replacements. Then, we calculated the behavioural metrics described above. This procedure was repeated 1,000 times to estimate the mean, confidence intervals and statistics.

**Timer model and hazard rate analyses.** To interpret the effects of optogenetic manipulations, we numerically simulated how different operations influence a timer, an accumulator that infers passage of time by integrating a constant input or periodic event, such as a water clock, hourglass, pendulum clock and quartz watch (Fig. 6). We modelled time as a scalar variable representing the temporal integration of a constant inflow signal. Specifically, the internal representation of time $T(t)$ evolves according to the equation:

$$T(t) = \int_0^t r$$

where $r$ is the inflow rate. A lick was triggered when $T(t)$ reached a fixed threshold $\theta = 1$. $r$ was varied across trials (but constant within each trial) to match the empirically observed distribution of lick times (inverse Gaussian distribution $IG(\mu, \lambda)$ with $\mu = 1.3$ and $\lambda = 12$; 10,000 iterations).

In addition to analysing lick-time distributions, we computed the hazard rate, defined as the instantaneous probability of a lick occurring at time $t$, given that no lick has occurred yet. Mathematically, the hazard rate $h(t)$ is computed as:

$$h(t) = \frac{f(t)}{1 - F(t)}$$

where $f(t)$ is the probability density function and $F(t)$ is the cumulative distribution function of lick times. This measure captures the moment-by-moment drive to lick and provides insight into the temporal dynamics of lick probability. We applied the same procedure to calculate hazard rate in the data. Both the data and the models were binned at 20 ms and smoothed with a boxcar filter over 5 bins. To quantify the temporal shift in hazard rate, we fitted the hazard rate (up to the time point where the cumulative distribution function reaches 80%;

beyond that, the hazard function becomes noisier as the denominator becomes small) with a sigmoid function and estimated its 50% point.

We simulated the effects of two types of transient perturbation to the timer: pause (slowdown), in which the inflow rate $r$ is transiently reduced by the speed coefficient $c$:

$$r_{\text{during manipulation}} = c \times r_{\text{before manipulation}}$$

Here $c = 0$ represents a complete pause, whereas larger values correspond to a slowdown (0.5 was used in Fig. 6b). By contrast, in rewind, the timer state $T(t)$ is transiently decreased as follows:

$$T(t) = T(t_{\text{stim on}}) + \int_{t_{\text{stim on}}}^{t_{\text{stim off}}} r_{\text{decay}}$$

Where $r_{\text{decay}}$ is a negative value, and $t_{\text{stim on}}$ and $t_{\text{stim off}}$ are the times of stimulation on and off, respectively. If $T(t) < 0$, $T(t)$ was set to zero. If $T(t)$ remained at zero for more than 320 ms, it was fixed at zero for the remainder of the trial, resulting in a no-lick outcome.

In all cases, the manipulation lasted for 600 ms and linearly decayed over the final 300 ms, matching the experimental condition. These manipulations were applied across 10,000 trials to assess their effects on lick timing and hazard rate dynamics.

**Trial-history regression analysis.** For the linear regression analysis in Fig. 2d, we tested 42 combinations of regressors with 1–6 lags with fivefold cross-validation (see Supplementary Fig. 3 for details). The median absolute deviation of lick time explained by different regression models was calculated as $1 - R1/R2$, where R1 is the median of the absolute value of the model residuals, and R2 is the median of the absolute value of the null model residuals.

**Extracellular recording analysis. Spike sorting and cell-type classification.** JRClust[74] (https://github.com/JaneliaSciComp/JRCLUST) with manual curations was used for spike sorting. We used quality metrics (described in Majumder et al.[68]) to select single units. Units with a total trial number of less than 75 were excluded from analyses. For the single-session population analysis, units with violated inter-spike interval were included.

For ALM recording, units with a mean spike rate above 0.5 Hz and spike width of 0.5 ms or more[37] (putative pyramidal neurons) were analysed. For striatal recording, units within the striatum (regions annotated as 'striatum', 'caudoputamen', and 'fundus of striatum' after registration to the Allen Common Coordinate Framework) with a mean spike rate above 0.1 Hz were analysed. We classified striatal neuron types based on spike features: striatal projection neurons (spike width ≥ 0.4 ms and with post-spike suppression duration ≤ 40 ms), fast-spiking interneurons (spike width < 0.4 ms and with less than 10% chance of having a long interspike interval) and tonically active neurons[44]. For the single-session analyses (decoding and projection to modes), only putative pyramidal neurons were analysed for the ALM recording, whereas all neurons were included for the striatal recording data. See Supplementary Table 1 for the number of recorded neurons in each experiment.

**Correlation in neural population activity.** To plot the correlation in neural population activity, we calculated the mean spike activity of individual neurons across trials with different lick-time ranges to yield a population activity matrix, with the number of rows equal to the number of neurons and the number of columns equal to the number of time points (200-ms bin). For Fig. 3, we calculated pairwise Pearson's correlation of these population activity matrices between trials with lick times between 1.40 s and 1.55 s (reference trials) and the trials with other lick-time ranges. For Figs. 4 and 5, we compared the pairwise Pearson's correlation between unperturbed trials with lick times between 1.4 s and 1.7 s (reference trials) and the photostimulation trials with lick times

between 1.7 s and 2.0 s). As a control, we subselected unperturbed trials with lick times closest to the median lick time in the unperturbed condition (the number of trials was matched to the number of trials as in the photostimulation condition). The choice of reference trials did not qualitatively change the results. For each correlation matrix, we identified the points along the $y$ axis with the maximum correlation (above 0.8) for each time point, and repeated this procedure with the hierarchical bootstrap (Figs. 3d,i, 4g,k and 5i,m).

**Single-cell analyses.** To plot the PSTH of example cells, PSTHs were calculated based on 1-ms time bin and smoothed with a 200-ms causal boxcar filter unless specified otherwise. To temporally warp PSTH for individual cells, we linearly scaled the spike timing after the cue, based on the time from cue to lick. Specifically, Spike time$_{warped}$ = Spike time$_{original}$/(LT$_{trial to be warped}$/LT$_{target warp time}$), where LT denotes the first lick time in each trial, and LT$_{target warp time}$ = 1 s.

Across-trial variance (Extended Data Fig. 2e–g) was calculated as the variance of spiking activity across trials for the original or temporally warped data (the across-trial variance was calculated for five 200-ms time windows after the cue and then averaged).

To quantify the number of cells that significantly increase or decrease spike rate before the lick compared with baseline, the trial-averaged spike rate of 0.2–0.5 s before the lick was compared with that of 0–1 s before the cue. Signed-rank tests were performed to determine whether the spike rate difference was significant.

To calculate the proportion of cells affected or unaffected by photostimulation (Figs. 4c and 5e), we analysed the spikes within the time window of 50–250 ms from the photostimulation onset time. To quantify the effect of photostimulation, trials with licks before the photostimulation onset time were excluded from the analysis. For individual cells, the spike rate in control and photostimulation trials was compared using the two-sided rank-sum test. Cells with a mean spike rate above 1 Hz during this window and more than 10 trials per condition were analysed.

In Extended Data Fig. 2, we analysed the partial rank correlation between the spike rate (in specific time windows) and the lick time in previous trials, removing the effect of upcoming lick time, for each cell (we only analysed trials after rewarded trials to avoid confound caused by the representation of rewards; analysis of previous unrewarded trials yielded similar results). Specifically, we calculated the rank correlation between spike rate ($R$) versus previous lick time ($P$; $\rho RP$), rank correlation between $R$ versus upcoming lick time ($U$; $\rho RU$) and rank correlation between $P$ and $U$ ($\rho PU$). Then, the partial correlation between spike rate versus lick time in the previous trial removing the effect of upcoming lick time is as follows:

$$\rho RP \cdot U = \frac{\rho RP - \rho RU \cdot \rho PU}{\sqrt{1 - \rho^2 RU}\sqrt{1 - \rho^2 PU}}$$

As controls, we performed a trial shuffle test, which shuffles the trial order and destroys trial history, and a session permutation test to avoid the confound of nonsensical correlations (1,000 iterations)[45]. The proportion of cells with a correlation higher than the chance level estimated by these controls is shown.

**Single-cell ramping characterization.** In Extended Data Fig. 2a–d, cells with more than 50 trials of lick time between 1.25 s and 1.5 s were used for firing pattern characterization. For each cell, PSTH was smoothed with a 200-ms causal boxcar filter. Trials were randomly split into halves and averaged to generate train and test data. We fit the activity of the train data from cue onset to the first lick time with different orders of polynomial functions (MATLAB polyfit function; tested order 1–8). We then calculated the mean squared error between the fit and the test data. The 'best-fit order' is the one with the lowest mean squared error. We repeated this procedure 10 times and defined the final order as the most frequent order among the 10 iterations. The best-fit data were then used to determine the monotonicity and

the peak firing time of the cell. Monotonic firing cells are those whose derivative of the best polynomial fit remains consistently positive or negative values from cue onset to lick. Peak firing time was the time point between the cue and lick where the best polynomial fit had the highest firing rate.

**Dimensionality reduction.** We characterized population activity patterns between the cue and the lick by defining modes that differentiate the baseline activity during the ITI (0–1 s before the cue) from the activity during specific 300-ms time windows after the cue: 0–0.3 s after the cue (cue mode), 0.5–0.8 s before the lick (middle mode), 0.2–0.5 s before the lick (ramp mode) and 0–0.3 s after the lick (execution mode).

Specifically, to calculate ramp mode for a population of $n$ recorded neurons, we looked for an $n \times 1$ unit vector that maximally distinguished the mean activity before the trial onset (0–1 s before cue; $r_{before cue}$) and the mean activity before the first lick (0.2–0.5 s before the first lick; $r_{before lick}$) in the $n$-dimensional activity space. We defined a population ramping vector: $w = r_{before lick} - r_{before cue}$. Ramp mode is $w$ normalized by its norm. Similarly, we defined cue mode, middle mode and execution mode using different time windows, and middle mode was orthogonalized to ramp mode, and cue mode was orthogonalized to both middle mode and ramp mode using the Gram–Schmidt process. Thus, the upper limit of the sum of square sum of task-modulated spiking activity explained (cue mode + middle mode + ramp mode) in Extended Data Fig. 3 is 1. Execution mode was orthogonalized to ramp mode (Extended Data Fig. 5a$_3$,b$_3$).

To define the trial-history mode, we first calculated the predicted lick time in each trial by applying the linear regression model described in the 'Trial-history regression analysis' section for each recorded session. Specifically, the model included previous lick times and the interaction between previous lick time and outcome at lags 1 and 2. This predicted value estimates what the lick time would be if it were determined solely by recent behavioural history and reinforcement, according to the fitted regression model, thereby summarizing trial history as a single value for each trial. We then calculated the Spearman rank correlation between the spike rate during the ITI (0–1 s before the cue) and the predicted lick time across trials for each neuron, indicating how strongly the ITI activity for each neuron encodes trial history. We obtained an $n \times 1$ unit vector representing the rank correlation of each neuron and normalized it by its norm to calculate the trial-history mode. Trial-history mode is not orthogonalized to any other modes.

In Fig. 3 and Extended Data Figs. 3 and 5a–c, we have pooled cells recorded across sessions (that is, pseudo-sessions). For each cell, we randomly selected 50 unperturbed control trials to define the mode. These unperturbed trials met the following criteria: the first lick occurred within 1–3 s after the cue, and there were no licks 3 s before the cue onset. Then, we selected a different set of trials to project the activity along these modes. Only neurons with more than 10 trials within all six lick time ranges were included. The six lick time ranges were: 0.80–1.10 s, 1.10–1.25 s, 1.25–1.40 s, 1.40–1.55 s, 1.55–1.70 s and 1.70–2.00 s.

To calculate the square sum of spiking activity explained by individual modes (Extended Data Figs. 3 and 5), we calculated the square sum of the activity along individual modes after subtracting the baseline activity (0–0.2 s before the cue), and then divided that by the square sum of the spike rate across neurons after subtracting the baseline activity. For each lick time range, we averaged across at least 10 trials with lick times within that range, and spiking data for each trial were smoothed using a 200-ms causal boxcar filter. To calculate the square sum of spiking activity explained by the sum of cue mode, middle mode and ramp mode reported in the main text, we calculated the square sum of task-modulated spiking activity explained between 0.2 s from cue (around when task modulation started) to lick for individual lick time ranges and then averaged across them. We calculated the square sum of spiking activity explained by trial-history mode activity similarly but without subtraction of the baseline activity (Extended Data Fig. 5a–c). In Extended Data Fig. 5d, we performed a linear regression analysis

between trial-history mode activity during 0–1 s before the cue and the upcoming lick time for each iteration of the hierarchical bootstrap. We then plotted the distribution of the linear regression coefficient (slope) across these iterations as a cumulative distribution function.

To calculate the angle between activity modes (Extended Data Fig. 3m), we computed the cosine similarity between two vectors of interest. Because cosine similarity depends on vector dimensionality (tending towards orthogonality as the number of neurons increases), we assessed statistical significance by shuffling one of the vectors and recalculating cosine similarity. This allowed us to determine whether the observed alignment between modes exceeded chance levels.

**Single-session analyses.** For analyses based on single sessions (Figs. 4 and 5 and Extended Data Figs. 4, 5, 7, 8, 10 and 12), sessions with more than 300 trials and five neurons were analysed. Spiking activity was binned per 50-ms time window. Activity between 1 s before the cue and the first lick in each trial was analysed (that is, post-lick activity was excluded as we focused on timing dynamics before the first lick). Dimensionality reduction was performed in the same manner as in the pseudo-session analysis, but modes were defined individually for each session. To visualize the time course of activity (for example, Fig. 4i), trials across sessions were pooled based on lick time, and only lick-time ranges that exist in at least two-thirds of the analysed sessions were shown. Therefore, the plotted lick-time ranges vary depending on the manipulation conditions.

To decode the $T_{to\,lick}$ from simultaneously recorded neural population activity, we conducted a kNN regression analysis. Within each experimental session, trials were partitioned into two sets: a test set comprising randomly selected 100 unperturbed trials and all perturbed trials, and a training set consisting of the remaining trials. For each moment in a test trial (50-ms window), we searched all time points in the training set to identify $k$ data points with the most similar population activity patterns (Mahalanobis distance based on the top principal components explaining 90% of variance). To estimate the $T_{to\,lick}$ of the test set, we averaged the $T_{to\,lick}$ in these kNNs. We tested '$k$' values between 20–50 (which are close to the square root of the number of data points in the training dataset) and found that they yielded similar results and did not change conclusions (data not shown). In the paper, we have reported the results with $k = 30$. Some sessions showed low decodability due to a small number of recorded neurons, trials and/or lack of task-modulated cells (Extended Data Fig. 4b). We analysed sessions in which the kNN decodability (Pearson's correlation between decoded lick time at the perturbation onset time, that is, 0.6 s after the cue versus actual lick time) was higher than 0.35.

To analyse the effect of perturbations systematically, we compared unperturbed versus perturbed trials after matching the number of trials and decoded time at the perturbation onset time (Extended Data Figs. 7a,c,d,e,g–i,k–m,o,p and 12e,i). Specifically, we randomly resampled animals, sessions and trials hierarchically (hierarchical bootstrap; 1,000 iterations). For each perturbed trial in each bootstrap iteration, we identified an unperturbed trial within the same session with the closest decoded time at the perturbation onset time (0.6 s after the cue). Then, we pooled these trials. This procedure allowed us to examine how decoded time (and projection along each mode) changed after the perturbation in conditions where their activity patterns were similar before the perturbation.

For the two-dimensional plots (Figs. 4e and 5g) and two-dimensional vector field analysis (Extended Data Fig. 11), we analysed how activity evolves in the two-dimensional space defined by ramp mode and middle mode. Spiking activity was binned in 50-ms time windows, and activity between the cue and the first lick in each trial was analysed. For each session, we projected the activity of ALM neurons along ramp mode and middle mode. The projection was normalized by the standard deviation of activity among control trials, but was not subtracted by the mean so that 0 represents 0 spike activity. For individual activity state ($x$) in control trials, we calculated the vector $r_x^{control}$ representing the direction that activity evolves in the next time point (50-ms time bin) in the two-dimensional state. Then, we calculated the mean vector for individual states in the two-dimensional space by averaging all vectors within a spatial bin of 0.5 along both the middle mode and ramp mode axes (if the spatial bin contained more than 30 data points): $r_{XY}^{control}$, where $X$ and $Y$ denote the location of the state along the middle mode and ramp mode axes, respectively. Similarly, we acquired the vector field during inhibition by pooling all time points during inhibition (100–400 ms from the inhibition onset) in photostimulation trials to acquire $r_{XY}^{stim}$. Then, we calculated the direction between $r_{XY}^{control}$ and $r_{XY}^{stim}$ for all states where both control and stim vectors exist. We excluded points where $\boldsymbol{r}_{XY}^{control}$ is within $\pi/6$ from $\tanh(Y/X)$ because if the activity is evolving against the zero point under control conditions, we cannot distinguish between whether the activity is rewinding or moving towards the zero point during the inhibition.

**Network models.** Using a dynamical systems approach, we considered four variables representing the average membrane currents ($h$) and spike rates ($r = f(h)$, where $f(h)$ is the neural activation function) of neuronal populations in the ALM and striatum. Conceptually, in these models, the striatum represents both connections within the striatum and the subcortical loop via the thalamus, which is why there are excitatory connections. In these models, the membrane potential of neuron $i$, $h_i(t)$, was governed by the following non-linear differential equation:

$$\tau \frac{dh_i(t)}{dt} = -h_i(t) + \sum W_{ij} r_j(t) + I_i^{base}(t) + I_i^{ext}(t) + I_i^{stim}(t)$$

$$r_i(t) = f(h_i(t))$$

Where $\tau$ is the membrane time constant (10 ms), $W_{ij}$ is the element of the connectivity matrix between the presynaptic neuron $j$ and the postsynaptic neuron $i$, $I_i^{base}(t)$ is the baseline input current, $I_i^{ext}(t)$ is the external input current, and $I_i^{stim}(t)$ is the negative current mediated by optogenetics to neuron $i$. The membrane current $h_i(t)$ was converted to the spike rate by applying a threshold-linear activation function $f(h) = \max(h,0)$.

For integrators mediated by a positive-feedback loop (Extended Data Fig. 1), we modelled two neurons in each brain area. The baseline input currents were chosen so that the system displays a stable fixed point at a low spike rate (lower attractor) with a spike rate of 5 spikes per second, consistent with the baseline firing rate observed in the experimental data. The connectivity matrix $W$ and the external input $I^{ext}(t)$ are shown in Extended Data Fig. 1. In these models, temporal integration is mediated by a continuous attractor, achieved by having an eigenvalue of 1 in the connectivity matrix. For each area, we defined the ramp mode using the same criteria as in the experimental data, and we then plotted spike rate activity along the ramp mode (Extended Data Fig. 1).

We tested models with different connectivity matrices reflecting distinct computational roles of the ALM and striatum (Extended Data Fig. 1). In the externally driven model (Extended Data Fig. 1a), the ALM received a ramping input that scaled with the desired lick times, progressively shifting the location of the fixed point in time. In the distributed model (Extended Data Fig. 1b), integration was achieved only when interareal connections between the ALM and striatum exist; in the absence of these long-range connections, neither the ALM nor striatum displayed slow-temporal dynamics. Conversely, in the redundant model, the ALM and striatum implemented two identical integrators (Extended Data Fig. 1c). Although weakly connected, their behaviour was independent of each other's input. In the specialized ALM integrator model (Extended Data Fig. 1d), the ALM served as the integrator, whereas the striatum followed ALM dynamics. In the specialized ALM leaky integrator model (Extended Data Fig. 1e), the ALM integrated the input with substantial leakiness. Although this model replicated the rewinding effect of striatal inhibition, it failed to reproduce the effect of ALM silencing. The model that best matched the neural dynamics observed in our data featured

the striatum as a perfect integrator and the ALM as a crucial input region (specialized striatum integrator; Extended Data Fig. 1f).

To mimic transient perturbation experiments, the negative current was introduced at 0.6 s after the cue and lasted for 0.6 s, including a 300-ms ramp down. For prolonged perturbations, the negative current was applied throughout the trial without a ramp down. To simulate ALM silencing, both ALM neurons (A1 and A2 in Extended Data Fig. 1) received a negative current $I^{stim}(t) = -10$. To stimulate D1-SPN inhibition, we injected a negative current into one of the striatum neurons (S1). To maintain similar perturbation effects on striatal ramp mode activity across different conditions, the negative current for D1-SPN inhibition was varied across models as follows: −0.3, −0.1, −0.1, −0.2, −0.02 and −0.03 for Extended Data Fig. 1a–f, respectively.

To generate ramping activity with a feedforward network (Extended Data Fig. 9), we used recurrent network modules composed of four or two cells, connected in a feedforward manner. In these models, the recurrent network modules have an architecture similar to that of the feedback model in Extended Data Fig. 1, but with weaker recurrent connections (eigenvalue < 1), such that each stage does not function as a perfect integrator. Thus, feedforward connections between stages are essential to amplify the input and generate both ramping and sequential activity. To implement temporal scaling, we provided a global inhibition to neurons in the feedforward network, that is, $I_i^{base}(t)$, was set to negative. This allows activity to propagate from one neuron to another when the effect of excitatory input exceeds this inhibition. Consequently, the speed of dynamics is controlled by the strength of the step input into the network. The step input is provided to an ALM neuron (except in model **h**, the striatal integrator model, where it is provided to the STR neuron) in the first recurrent module, allowing it to be amplified by both recurrent and feedforward connections. In addition to the step input, we also provided a transient, cue-like input to both ALM neurons (600-ms duration with 450-ms linear ramp down). This input does not get amplified by the recurrent architecture, as it is provided along an axis orthogonal to the one amplified by the recurrent connections (as in Extended Data Fig. 9a). This helps to generate a robust cue-related response in the model, similar to the experimental data. The connectivity matrix $W$ and input are described in detail in Supplementary Table 2. Transient perturbations were simulated similarly to the positive-feedback network. For ALM complete silencing, $I^{stim}(t) = -10$ was injected into all ALM neurons. To simulate D1-SPN inhibition, a negative current was injected into half of the striatal neurons (s2 and s3; the result did not change regardless of the choice of two inhibited neurons). To maintain similar perturbation effects on striatal ramp mode activity across different conditions, the negative current for D1-SPN inhibition was varied across models as shown in Supplementary Table 2. For each area, we defined the ramp and other modes using the same criteria as in the experimental data, and we then plotted spike rate activity along these modes.

Ramp mode activity was normalized across conditions so that a value of 0 corresponds to the baseline firing rate (5 Hz) of the target neuron (in the positive-feedback model, the target neuron is neuron 3 in all cases, or neuron 4 in the ALM leaky integrator model; in the feedforward model, the target neuron is the ALM neuron in the last recurrent network module), whereas a value of 1 corresponds to the time point when the activity of the target neuron reaches 10 Hz, approximating the activity level just before licking in our data. Thus, the absolute spike rate of the target neuron can be inferred from its activity along the ramp mode, providing a direct mapping between model output and population activity. For the lick time analysis in Extended Data Figs. 1g and 9e, the lick time was defined as when the ramp mode activity reached a value of 1.

**Statistics.** The sample sizes were similar to the sample sizes used in the field. No statistical methods were used to determine the sample size. During spike sorting, experimenters could not tell the trial type and, therefore, were blind to conditions. All signed-rank and rank-sum tests were two sided. All bootstrapping was done over 1,000 iterations.

## Reporting summary

Further information on research design is available in the Nature Portfolio Reporting Summary linked to this article.

## Data availability

The recording data in NWB format is available at the DANDI archive (ID 001610).

## Code availability

The source code (MATLAB R2020b and R2022b, and Python3) to reproduce the results of this study is available on GitHub (https://github.com/inagaki-lab/Yang_et_al_2024) and archived on Zenodo[75] (https://doi.org/10.5281/ZENODO.17343841).

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

**Acknowledgements** We thank P. Dayan, M. Andermann, N. Li, D. Fitzpatrick, T. Wang, M. Sarvestani, L. Colgan, B. Mensh, R. Heldman, K. Citrin, T. Yiu, X. Yu and Inagaki laboratory members for comments on the manuscript; K. Daie, S. Romani and S. Saxena for discussions; P. Scarpinato and N. Spiller for the DeepLabCut pipeline; K. Shirley, A. Ilchenko and MPFI Light Microscopy Core for imaging; H. Shearin and other MPFI ARC members for animal care; and MPFI Mechanical Workshop for machining. This work was funded by the Max Planck Florida Institute for Neuroscience (to H.K.I.), the Max Planck Free Floater Program (to H.K.I.), the National Institutes of Health New Innovator Award (NINDS and OD; 1DP2NS132108; to H.K.I.), the Searle Scholars Program (to H.K.I.), the Klingenstein-Simons Fellowship (to H.K.I.), the McKnight Scholar Award (to H.K.I.), and NIMH-IRP ZIA MH002497-36 (to C.R.G).

**Author contributions** H.K.I. conceptualized the study. Z.Y., M.I., C.R.G., L.F. and H.K.I. conducted the investigation. H.K.I. acquired funding. H.K.I. provided supervision. Z.Y. and H.K.I. wrote the manuscript, with input from all authors.

**Funding** Open access funding provided by Max Planck Society.

**Competing interests** The authors declare no competing interests.

**Additional information**
**Correspondence and requests for materials** should be addressed to Hidehiko K. Inagaki.

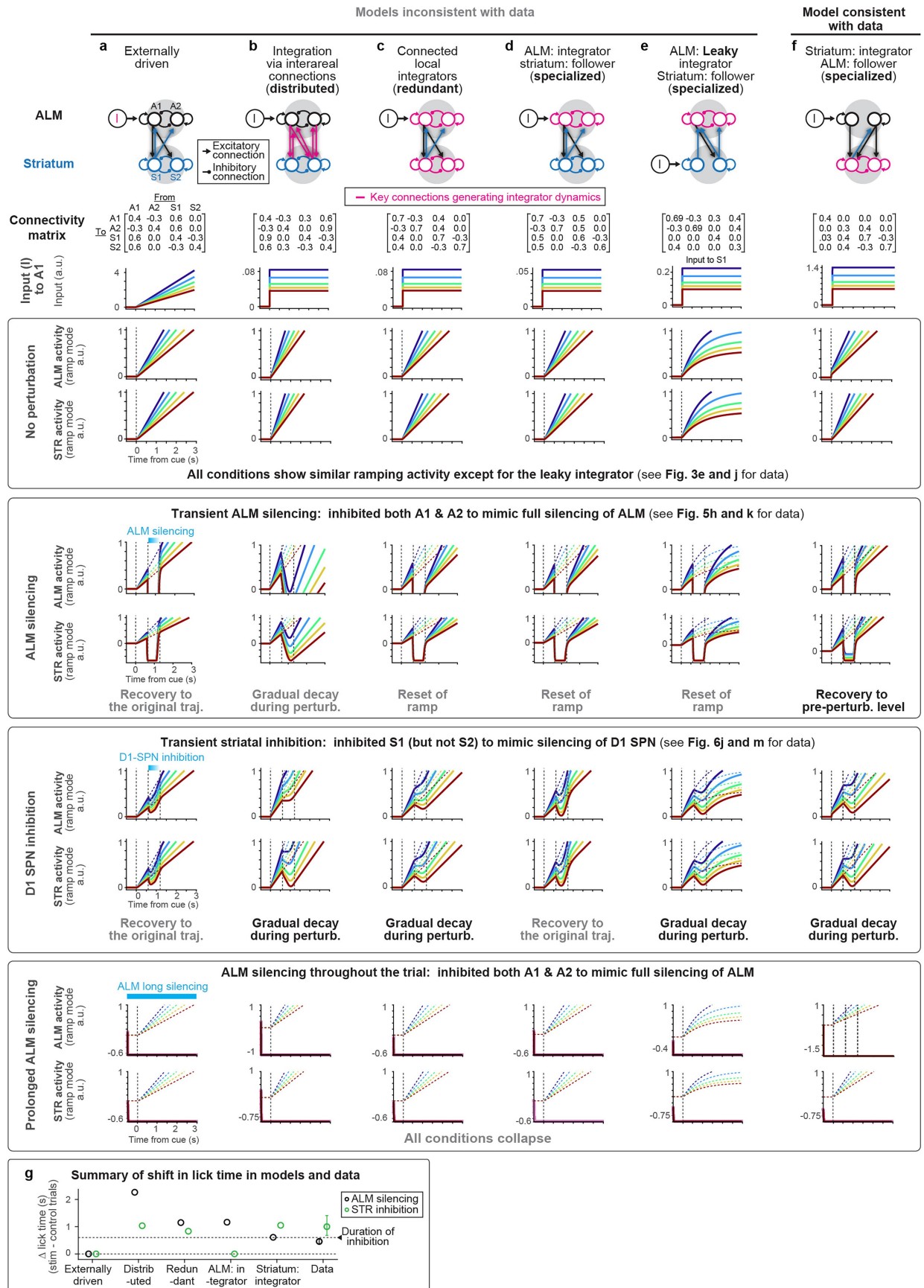

**Extended Data Fig. 1** | See next page for caption.

**Extended Data Fig. 1 | Two-regional network models replicating the ramping activity in ALM and striatum.** Multi-regional network models of ALM and striatum: both regions contain two neurons and exhibit ramping activity with temporal scaling as in Fig. 3. Additionally, a step input mimicking the trial-history mode activity is provided to ALM (except for **e**; most models yield consistent results even if we provide the input to the striatum, and thus only one configuration is shown). The network configuration, i.e., the connectivity matrix, varies across models, leading to different location(s) of integrator(s) and various responses to transient perturbations. ALM silencing was implemented by silencing both neurons in ALM, and striatal inhibition was implemented by silencing one of the striatal neurons (mimicking the silencing of D1-SPNs). Note that prolonged silencing does not distinguish between models (bottom row), underscoring that transient perturbation with concurrent multi-regional recording is essential to differentiate models. **a.** Externally driven model. 1st row, schema of the model (arrows indicate connections between ALM and striatum as in the corresponding connectivity matrix below), and the connectivity matrix used for the simulation. 2nd row, input (I) into the network. 3rd and 4th row, ALM and striatal ramp mode activity without perturbation. 5th and 6th row, ALM and striatal ramp mode activity during transient ALM silencing (the ALM activity along the ramp mode was cropped for visualization purposes. In all cases, ALM activity during ALM silencing decreased to 0). 7th and 8th row, ALM and striatal ramp mode activity during transient striatum inhibition. 9th and 10th row, ALM and striatal ramp mode activity during prolonged ALM silencing throughout the trial. Dashed lines, control conditions overlaid. **b.** Same as in **a** but for integration via interareal connections (distributed) model. **c.** Same as in **a** but for coupled local integrators (redundant) model. **d.** Same as in **a** but for ALM being the integrator and striatum being the follower (specialized) model. **e.** Same as in **a** but for ALM being a leaky integrator and striatum being the follower (specialized) model. This model replicates the rewinding of ALM and striatum dynamics during striatal inhibition (where rewinding is caused by the loss of input to the leaky integrator) but cannot reproduce the effect of ALM silencing. **f.** Same as in **a** but for striatum being the integrator and ALM being the follower (specialized) model that replicates the data. **g.** Comparison of shifts in lick time following perturbation across models and data (Methods; the lick time was estimated in red traces in each plot). The last column (Data) was duplicated from Figs. 4b and 5d for comparison purposes.

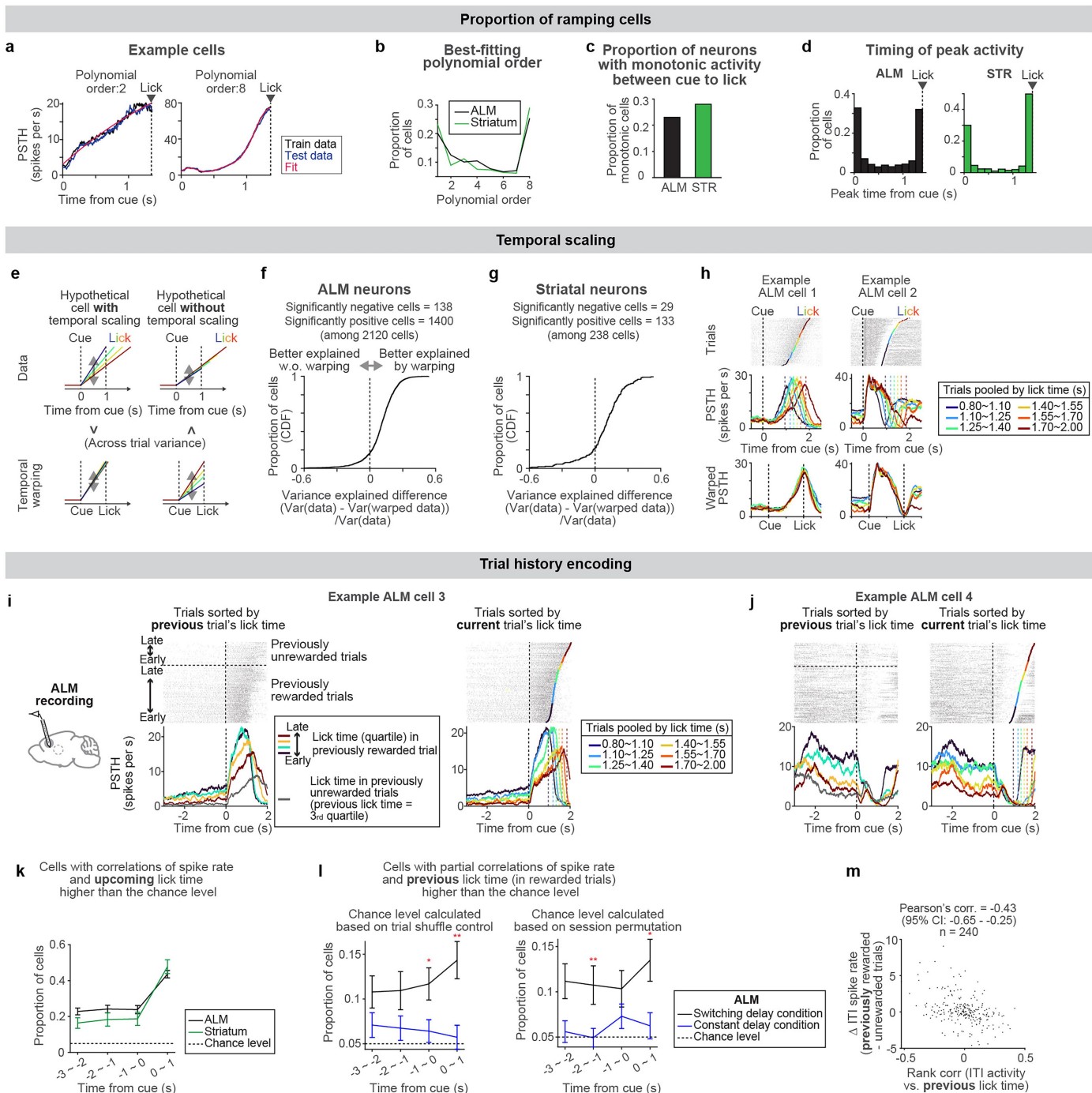

**Extended Data Fig. 2** | See next page for caption.

**Extended Data Fig. 2 | Single-cell characterization of task-modulated activities.** Polynomial fit of individual cells' PSTH showed that about a quarter of neurons exhibit monotonic ramping activity (**a-d**). Temporal warping of spike times revealed that the activity in a large proportion of ALM and striatal neurons can be better explained by temporal scaling rather than a null model without scaling (**e-h**). Some ALM and striatal neurons maintain tonic activity during the ITI, which predicts upcoming lick time (**k**) and encodes trial history (**l-m** and example cells in **i** and **j**). **a**. Example cells with polynomial fitting. Black: training data used to fit the polynomial; blue: test data used to evaluate the fit (mean square error between fit and test data; 10-fold cross-validation); red: best-fitting polynomial. We performed this analysis on activity from cue to lick in trials in which mice licked at 1.25–1.5 s (Methods). **b**. Distribution of best-fitting polynomial order for ALM (black) and striatum (green); 2418 and 553 neurons, respectively (Methods). **c**. Proportion of neurons with monotonic activity between cue and lick, defined as either linear or polynomial fits with derivatives consistently negative or positive from cue to lick. **d**. Distribution of peak activity timing shows that peaks are concentrated at the beginning or just before the lick, indicating that even non-monotonic neurons often resemble ramp-up or ramp-down profiles rather than exhibiting peak activity mid-trial. Histogram bin size, 0.14 s. **e**. Schema illustrating two hypothetical cells that encode time differently through ramping activity[29]. Left, a cell with ramping activity encoding relative time, where the ramping speed changes as the lick time varies (i.e., temporal scaling). In this scenario, the across-trial variance (double-headed arrows) decreases following temporal warping (bottom). Right, a cell with ramping activity encoding absolute time, where the spike rate increases as time progresses. In this case, the across-trial variance increases following temporal warping (right). **f**. The cumulative distribution of the difference of variance explained between data and warped data across ALM neurons (higher value represents data explained better by temporal scaling; see **e**, Methods). n = 2139 neurons, 31 mice. Neurons with more than 200 trials were analyzed. Significant cells, p < 0.05 with bootstrap of trials. **g**. Same as in **b** but for striatal cells, n = 595 neurons, 10 mice. **h**. Two example ALM cells showing temporal scaling. Same format as Fig. 3a: trials are sorted by the current trial's lick time and grouped into six ranges. **i**. An ALM example cell whose ITI activity is modulated by the lick time and reward outcome in the previous trial and anticipates the upcoming lick time. The same cell as in Fig. 3k for comparison. Top, spike raster, grouped by reward outcome in the previous trial and sorted by the lick time in the previous trial. In this example cell, the ITI activity is higher in trials after rewarded trials and with earlier licks. Bottom, PSTH. Lick times of the previous rewarded trials were divided into quartiles

indicated by different colors. The gray trace, trials following previously unrewarded trials with previous trial's lick times within the 3rd quartile. The orange trace represents trials that followed rewarded trials with the same range of previous lick times. This allows us to compare the effect of reward on spiking activity in upcoming trials while controlling for the influence of prior lick timing. In this example, the cell shows a lower spike rate during the inter-trial interval following unrewarded trials compared to rewarded ones. Right, the same cell but trials sorted by lick time in current trials. The spike rate during ITI predicts the upcoming lick time. The brain atlas was adapted from the Allen Institute for Brain Science (https://atlas.brain-map.org). **j**. Another ALM example cell showing trial history modulation during ITI, same format as **i**. **k**. The proportion of neurons with a rank correlation between spiking activity (in different time windows indicated on the x-axis) and upcoming lick time higher than the trial shuffle control (α = 0.05). Neurons with more than 100 current trials and a current lick time later than 1 s (to avoid the influence of post-lick activity for the 0–1 s time window) were analyzed. $P$-values < 0.002 for all data points, as determined by a hierarchical bootstrap test under the null hypothesis that the measured values do not differ from 0. Mean ± SEM (hierarchical bootstrap). n = 3966 neurons, 42 mice for ALM., n = 1419 neurons, 16 mice for striatum. **l**. The proportion of ALM neurons with a partial rank correlation between spiking activity (in different time windows indicated on the x-axis) and previous lick time higher than the trial shuffle control (left) and session permutation control (right) (α = 0.05). Partial correlation was calculated to control for the effect of upcoming lick time (Methods). Both yielded consistent results. Neurons with more than 50 current trials and a current lick time later than 1 s (to be consistent with **k**) were analyzed (n = 1280 neurons, 20 mice for ALM, n = 1485 neurons, 10 mice for striatum). * $p < 0.05$, ** $p < 0.01$ following *Bonferroni* correction, hierarchical bootstrap comparing switching vs. constant delay condition. $P$-value from left to right in left panel, 0.045, 0.04, 0.005, 0.002; in right panel 0.016, 0.001, 0.145, 0.007 (values without correction for multiple comparison). Mean ± SEM (hierarchical bootstrap). **m**. Relationship between the encoding of previous lick time (partial rank correlation between spike rate vs the lick time in previous rewarded trials, controlling for the effect of upcoming lick time; Methods) and whether the animal received a reward or not in the previous trial (based on activity during ITI: 0–1 s before the cue) in ALM. Dots, individual neurons. Neurons with more than 20 previous unrewarded trials, 20 previous rewarded trials, 50 current trials with lick time later than 1 s were analyzed. Neurons encoding previous lick time also tend to encode previous reward outcome.

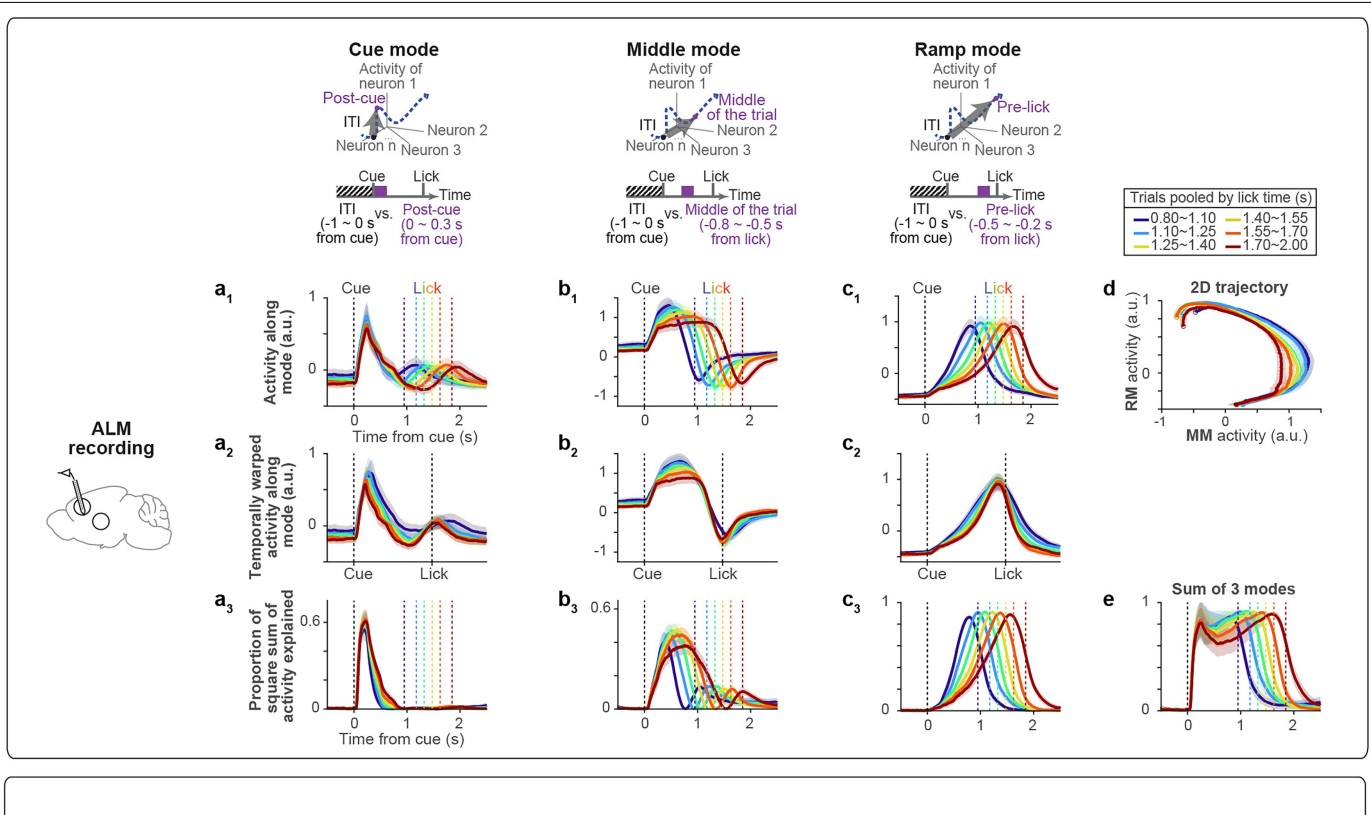

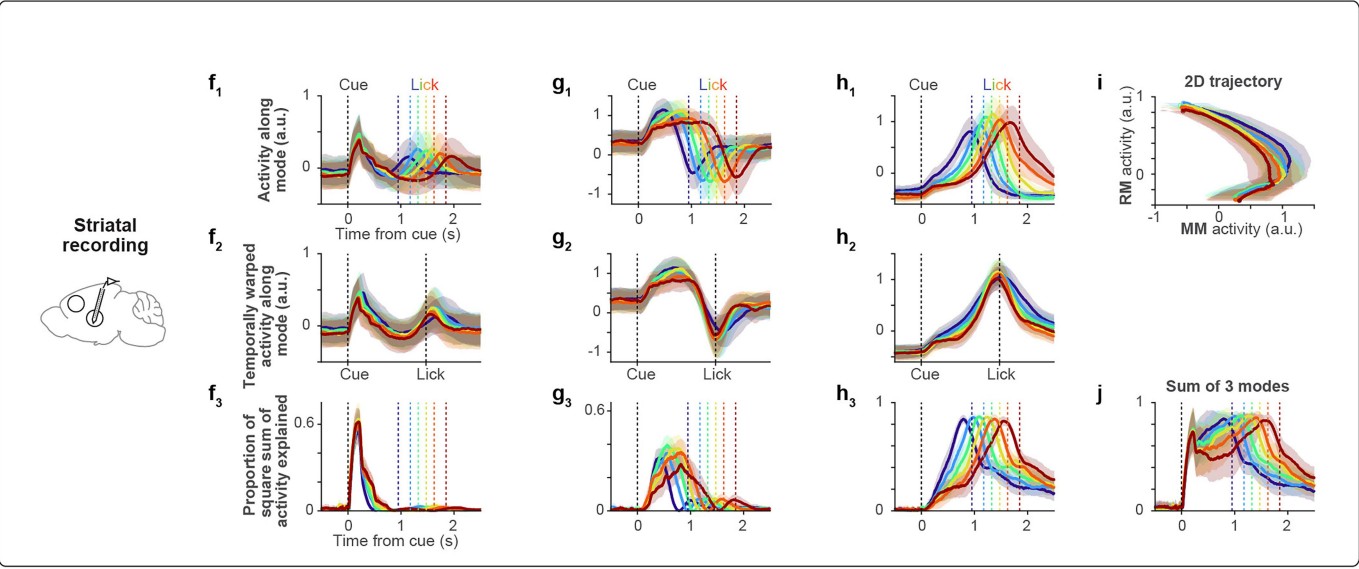

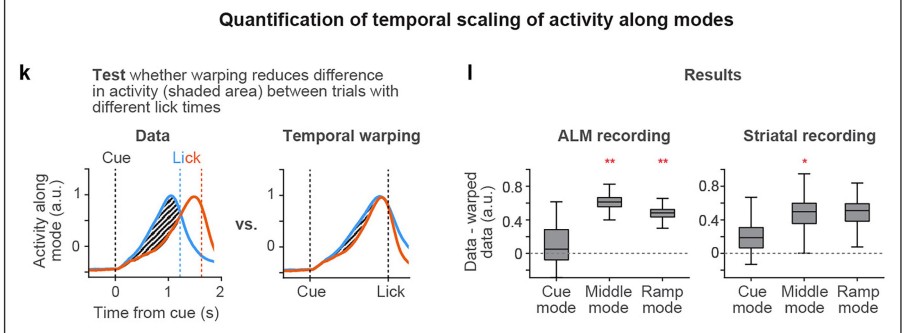

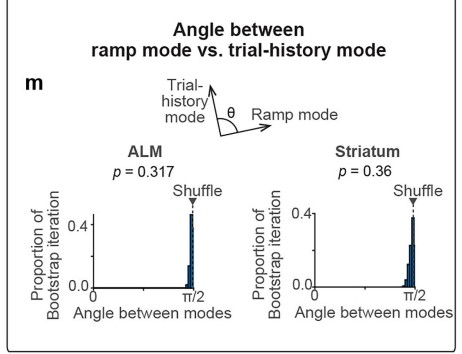

**Extended Data Fig. 3** | See next page for caption.

**Extended Data Fig. 3 | Comparison of activity modes across brain regions.**
In both ALM and striatum, we observed three modes of population activities
(cue mode, middle mode, and ramp mode) that together tiled the time from
trial onset to lick, explaining around 80% of the square sum of task-modulated
spiking activity. Activity along the cue mode did not show temporal scaling,
while activity along the middle and ramp mode showed temporal scaling
(quantified in **k** and **l**). Note that activity patterns along these modes and the
square sum of task-modulated spiking activity explained are qualitatively
similar between ALM and striatum. The angle between ramp mode and trial-
history mode is orthogonal (**m**). **a**. ALM population activity along the cue
mode, under switching delay condition (**a1**). Cue mode activity temporally
warped between cue and lick (**a2**). Square sum of task-modulated spiking
activity explained by cue mode (**a3**). Note that this value was calculated after
trial averaging (10 trials) and smoothing (200 ms causal boxcar filtering).
Colors, different lick times. Lines, grand mean. Shading, SEM (hierarchical
bootstrap), applied to all the shadings in this figure. n = 3261 neurons, 45 mice.
**b**. Same as in **a** but for ALM population activity along the middle mode. **c**. Same
as in **a** but for ALM population activity along the ramp mode. Duplicated from
Fig. 3e for comparison. **d**. Population activity in a two-dimensional space
defined by the ramp mode (RM) and middle mode (MM). Trajectories are
plotted from the cue (filled circles) to the lick (open circles). Regardless of the
lick time, the activity follows a similar trajectory, but the speed varies across
different lick times. **e**. The total square sum of task-modulated spiking activity
explained by the three modes. **f-j**. Same as in **a-e** but for striatal neurons under
switching delay condition. n = 1073 cells, 16 mice. The brain atlas in panels **a**,**f**
was adapted from the Allen Institute for Brain Science (https://atlas.brain-
map.org). **k**. Schema representing the quantification of temporal scaling.
We calculated the difference in activity along a mode between two trial types
(trials with licks occurring 1.1–1.25 s vs. 1.55–1.7 s; the difference was calculated
from the cue to the lick, shaded area). If the population activity along a mode
exhibits temporal scaling, the difference between lick times will be smaller
following temporal warping. **l**. Left, the difference in activity (shaded area) as
described in panel **k** is compared between the data and the temporally warped
data for each mode in the ALM (based on data shown in panels **a-c**). Right, same
for the striatum (based on data shown in panels **f-h**). The central line in the box
plot, median. Top and bottom edges, 75% and 25% points. Whiskers, the lowest/
highest datum within the 1.5 interquartile range of the lower/upper quartile.
$p$ = 0.42, <0.001, <0.001 for ALM Cue, Middle, and Ramp mode; $p$ = 0.102, 0.002,
0.057 for striatum Cue, Middle, and Ramp mode, respectively. **\*\***$p$ < 0.001,
**\***$p$ < 0.01 (significant scaling; P-values, hierarchical bootstrap with a null
hypothesis that the difference between data minus temporal warped data is
less than or equal to 0). **m**. Angle between ramp mode and trial-history mode
(histogram of 1000 bootstrap iterations is shown). Left, ALM recording. Right,
striatum recording. Dotted lines, angles of shuffled control. P-values, bootstrap
test of the null hypothesis that the observed angle is equal to the shuffled control.
In both ALM and striatum, ramp mode and trial-history mode are orthogonal.

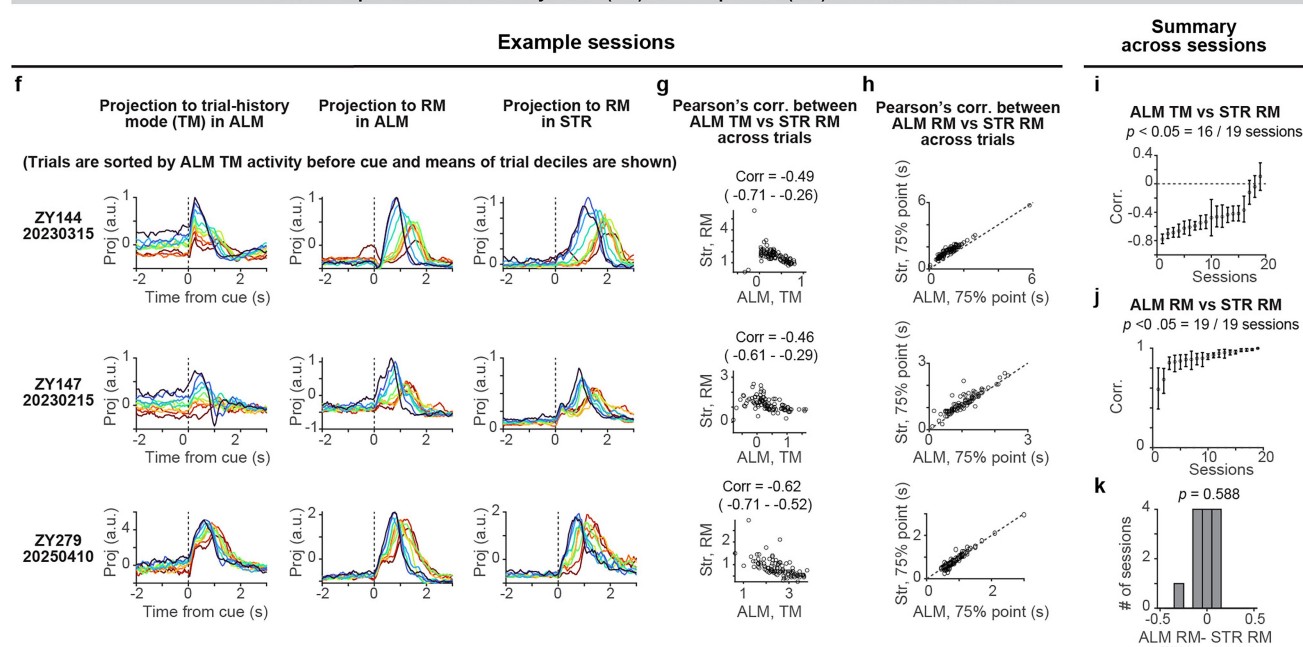

**a**

**Estimate lick timing using kNN decoder**

A data point in a test trial

Activity of neuron 1

1) Find k nearest neighbor (kNN) data points in ref trials → 2) Average "time to lick" of the kNN ref data points → Decoded time to lick of the test data point

Neuron 2
Neuron 3
Neuron n

**b** Pearson's correlation between decoded vs. actual lick time (decoded at cue onset)

Correlation / Number of neurons per session
○ ALM recording
○ STR recording

**c** Example trials
ALM & striatum
Decoded time to lick (s) / Time from cue (s)
ALM
Striatum

**d** Example session
Corr = 0.79
Shuffle: 0.51-0.62
Decoded time to lick (s) based on ALM activity / Decoded time to lick (s) based on STR activity

**e** Corr. = 0.726 ± 0.09 (mean ± std)
*p* < 0.05 in 19 / 19 sessions
Data / Shuffle
□ Session

**Relationship between trial history mode (TM) and ramp mode (RM) in individual sessions**

**Example sessions**

**Summary across sessions**

**f**
Projection to trial-history mode (TM) in ALM | Projection to RM in ALM | Projection to RM in STR

**g** Pearson's corr. between ALM TM vs STR RM across trials

**h** Pearson's corr. between ALM RM vs STR RM across trials

**i** ALM TM vs STR RM
*p* < 0.05 = 16 / 19 sessions
Corr. / Sessions

(Trials are sorted by ALM TM activity before cue and means of trial deciles are shown)

ZY144 20230315
Proj (a.u.) / Time from cue (s)

Corr = -0.49 ( -0.71 - -0.26)
Str, RM / ALM, TM

Str, 75% point (s) / ALM, 75% point (s)

**j** ALM RM vs STR RM
*p* <0 .05 = 19 / 19 sessions
Corr. / Sessions

ZY147 20230215
Proj (a.u.) / Time from cue (s)

Corr = -0.46 ( -0.61 - -0.29)
Str, RM / ALM, TM

Str, 75% point (s) / ALM, 75% point (s)

**k**
*p* = 0.588
# of sessions / ALM RM- STR RM 75% point (s)

ZY279 20250410
Proj (a.u.) / Time from cue (s)

Corr = -0.62 ( -0.71 - -0.52)
Str, RM / ALM, TM

Str, 75% point (s) / ALM, 75% point (s)

**Extended Data Fig. 4** | See next page for caption.

**Extended Data Fig. 4 | Tight trial-by-trial coupling of dynamics between ALM and striatum.** In sessions with simultaneous ALM-striatum recordings, decoded time and ramping activity are highly correlated between ALM and striatum on a trial-by-trial basis. In addition, the amplitude of ALM TM activity is correlated with the slope of the striatal ramping on a trial-by-trial basis (**f**, **g**, and **i**). **a**. Schema depicting a k-nearest neighbor (kNN) method to decode the time to lick ($T_{to\,lick}$) using population neural activity at each time point (Methods). **b**. The performance of the kNN decoder as a function of the number of simultaneously recorded neurons. Decoding accuracy was quantified by Pearson's correlation between actual lick time vs. lick time decoded at cue onset. The performance increased with more recorded neurons. **c**. Decoded $T_{to\,lick}$ was highly correlated between ALM and striatum at a single-trial level. Four example trials from an example session are shown. Traces end at the time of lick. The brain atlas was adapted from the Allen Institute for Brain Science (https://atlas.brain-map.org). **d**. The relationship between decoded lick times estimated from ALM and striatal neurons in an example session. Dots, all time points (50 ms bin; from cue to lick) in the example session. Pearson's correlation across all time points (0.79) was significantly higher than that of the trial shuffle control (0.51–0.62, 95% CI). **e**. Pearson's correlation of decoded time between ALM and striatum was significantly higher than trial shuffle controls in all simultaneously recorded sessions (*P*-value, hierarchical bootstrap with a null hypothesis that observed data are lower than or equal to shuffled data, 19 sessions). Thus, ALM and striatal timing dynamics are synchronized at a single-trial level. **f**. Three example ALM-striatum simultaneous recording sessions (from top to bottom). Left, ALM activity projected to trial-history mode (TM). Middle, ALM activity projected to ramp mode. Right, striatum (STR) activity projected to ramp mode. Line, mean of trial deciles. For all modes, trials are sorted and grouped by ALM activity along the trial-history mode before the cue (blue, trials deciles with the highest activity; red, trial deciles with the lowest), allowing direct comparison of activity in the same trial deciles. Trials with high activity along TM (blue) exhibit steeper ramping activity in both ALM and striatum. **g**. Across-trial Pearson's correlation between the amplitude of ALM activity along the trial-history mode and the slope (time to reach 75% of peak activity after the cue) of striatum activity along the ramp mode for the three sessions shown in **f**. Circles, individual trials. **h**. Across-trial Pearson's correlation between the slope of ALM and STR activity along the ramp mode for the three sessions shown in **f**. Circles, individual trials. Dotted line, the unity line. **i**. Across-trial Pearson's correlation between the amplitude of ALM activity along the trial-history mode and the slope (time to reach 75% of peak activity level after the cue) of STR activity along the ramp mode for all simultaneously recorded sessions. Data are presented as mean ± 95% CI. (hierarchical bootstrap). 16 out of 19 sessions show a significant negative correlation between ALM TM activity and STR RM activity. *P*-value, hierarchical bootstrap with a null hypothesis that the correlation is equal or higher than 0. n = 14 mice. **j**. Across-trial Pearson's correlation between the slope of ALM activity along the ramp mode and STR activity along the ramp mode for all sessions. Data are presented as mean ± 95% CI (hierarchical bootstrap). All 19 sessions show a significant positive correlation between ALM RM activity and STR RM activity. *P*-value, hierarchical bootstrap with a null hypothesis that the correlation is equal to or lower than 0. **k**. Temporal difference between ALM and STR RM activity in reaching 75% of its maximum level. Two-sided sign rank test. No significant difference was detected.

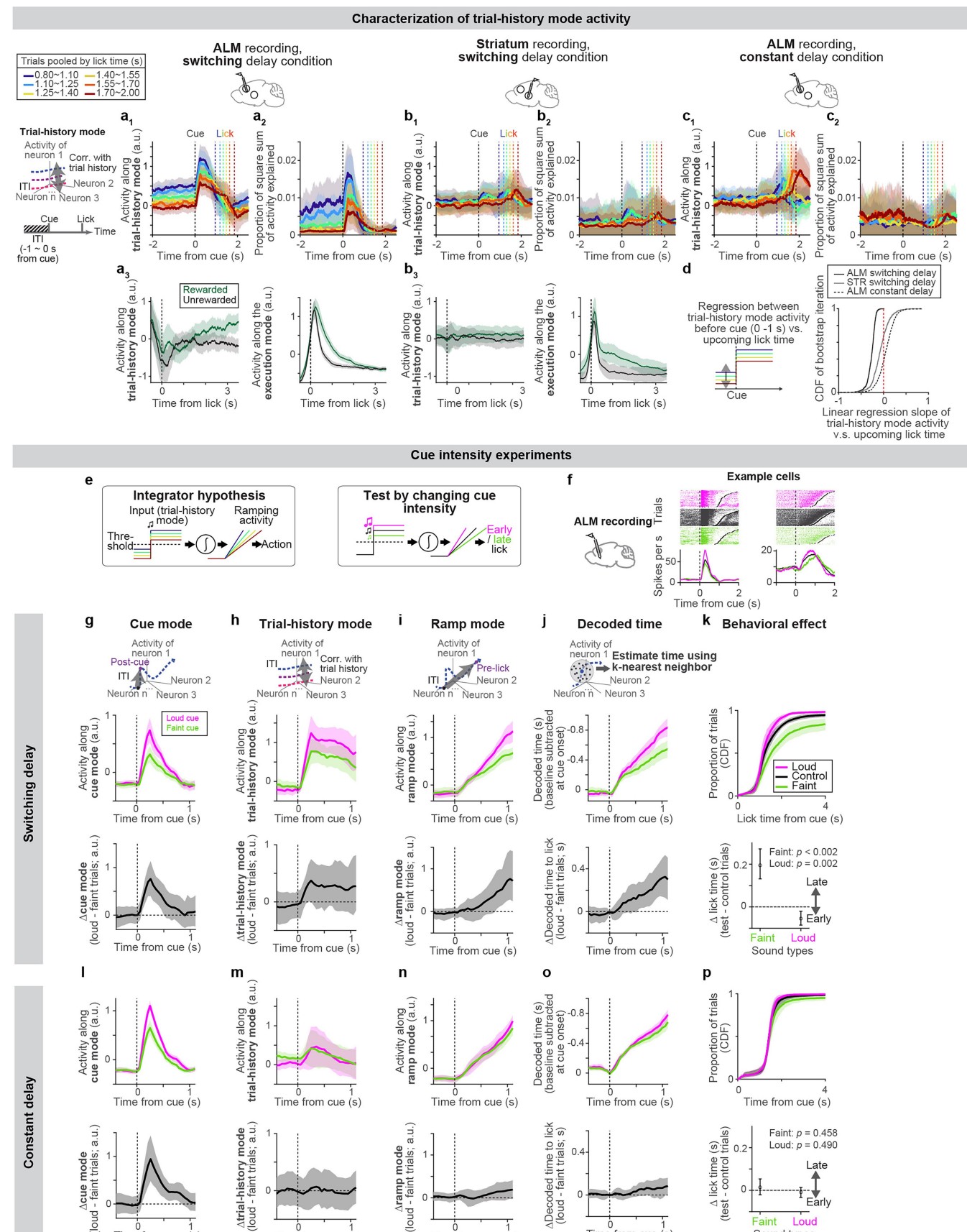

**Extended Data Fig. 5 |** See next page for caption.

**Extended Data Fig. 5 | Tonic activity in ALM predicts lick time and transient sensory manipulation to test the integrator hypothesis.** Activity along the trial-history mode in the ALM under the switching delay condition predicts upcoming lick time (**a** and **d**). However, activity along the trial-history mode in the striatum under the switching delay condition (**b**) was weaker. Additionally, ALM activity under the constant delay condition (**c**) did not predict upcoming lick time, consistent with single cells (Extended Data Fig. 2l). Consistent with the integrator hypothesis (**e**), cue intensity manipulation is integrated into ramping dynamics to influence upcoming action only under the switching delay condition (**e-k;** but not under the constant delay condition without a prominent trial-history mode; **l-p**). This suggests that the cue intensity–dependent modulation of lick time is context dependent and cannot be attributed solely to changes in reaction time driven by sound intensity. **a**. ALM population activity along the trial-history mode, under switching delay condition (**a1**). Duplicated from Fig. 3l for comparison. Square sum of task-modulated spiking activity explained by trial-history mode (**a2**). Colors, different lick times. Lines, grand mean. Shading, SEM (hierarchical bootstrap), applied to all the traces in **a-c**. n = 3261 neurons. ALM activity along the trial-history mode gradually diverged after rewarded vs. unrewarded licks (left) under the switching delay condition (**a3**). In contrast, ALM activity along the execution mode, which captures the activity during the lick (Methods), showed a transient change in activity after the lick (up to -2 s). This transient change along the execution mode most likely reflects differences in lick patterns between these trial types. Since the trial-history mode activity started diverging after the execution modes converged, the divergence of trial-history mode activity is likely not due to movement. Rewarded (green) and unrewarded (black) trials with similar lick times (lick between 1.4 and 1.8 s after the cue) were analyzed. **b**. Same as in **a** for striatal recording during the switching delay condition. **c**. Same as in **a** for ALM recording during the constant delay condition. **d**. The relationship between trial-history mode activity and upcoming lick time. We performed a linear regression analysis between trial-history mode activity during 0–1 s before the cue and the upcoming lick time for each iteration of the hierarchical bootstrap (left, schema). We then plotted the distribution of the linear regression coefficient (slope) across these iterations. A negative value indicates that higher trial-history mode activity precedes earlier licks. $p < 0.001$, = 0.3274, and = 0.6310 for ALM under switching delay, striatum under switching delay, ALM under constant delay, respectively (with a null hypothesis that the slope of the linear regression is larger than or equal to 0). Thus, only under the switching delay condition in ALM, the trial-history mode activity significantly predicts the upcoming lick time. **e**. Left, schema of integrator hypothesis. Right, schema showing the prediction of the cue intensity experiment: changes in trial-history mode in response to different cue intensities are integrated into lasting changes in ramping activity, affecting lick timing. **f**. Two example ALM neurons showing bidirectional modulation in cue response (left) or ramping activity (right) in response to different cue intensities. Magenta, loud cue. Green, faint cue. The brain atlas in panels **a**–**c**,**f** was adapted from the Allen Institute for Brain Science (https://atlas.brain-map.org). **g**. ALM population activity in response to different cue intensities. Top, schema of cue mode. Middle, ALM population activity projected along cue mode (faint vs loud cue trials). Bottom, the difference in activity along the cue mode between loud and faint cue trials. Line, mean. Shade, 95% CI (hierarchical bootstrap). n = 788 neurons, 15 sessions, 5 mice. **h-j**. The same as in **g**, but for trial-history mode, ramp mode, and decoded time to lick. Note that the decoded time to lick was baseline-subtracted relative to the value at cue onset (time 0), since the mean lick-time in control trials varies across sessions. This subtraction causes the decoded time to lick to start at 0 and take on negative values as it approaches the actual lick. **k**. Lick times following different cue intensities. Top, cumulative distribution of lick time. Shade, 95% CI (hierarchical bootstrap). Bottom, quantification of the change in lick time. Data are presented as mean ± 95% CI (hierarchical bootstrap). P-value, hierarchical bootstrap with a null hypothesis that there is no change from the control condition. n = 22 sessions, 8 mice. **l-p**. Same as **g-k** but for the constant delay condition. n = 977 neurons, n = 15 sessions, 5 mice. Despite bidirectional modulation of cue and trial-history modes, there was no change in ramp mode and decoded time. Note that the amplitude of the trial-history mode is small under the constant delay condition (**c**).

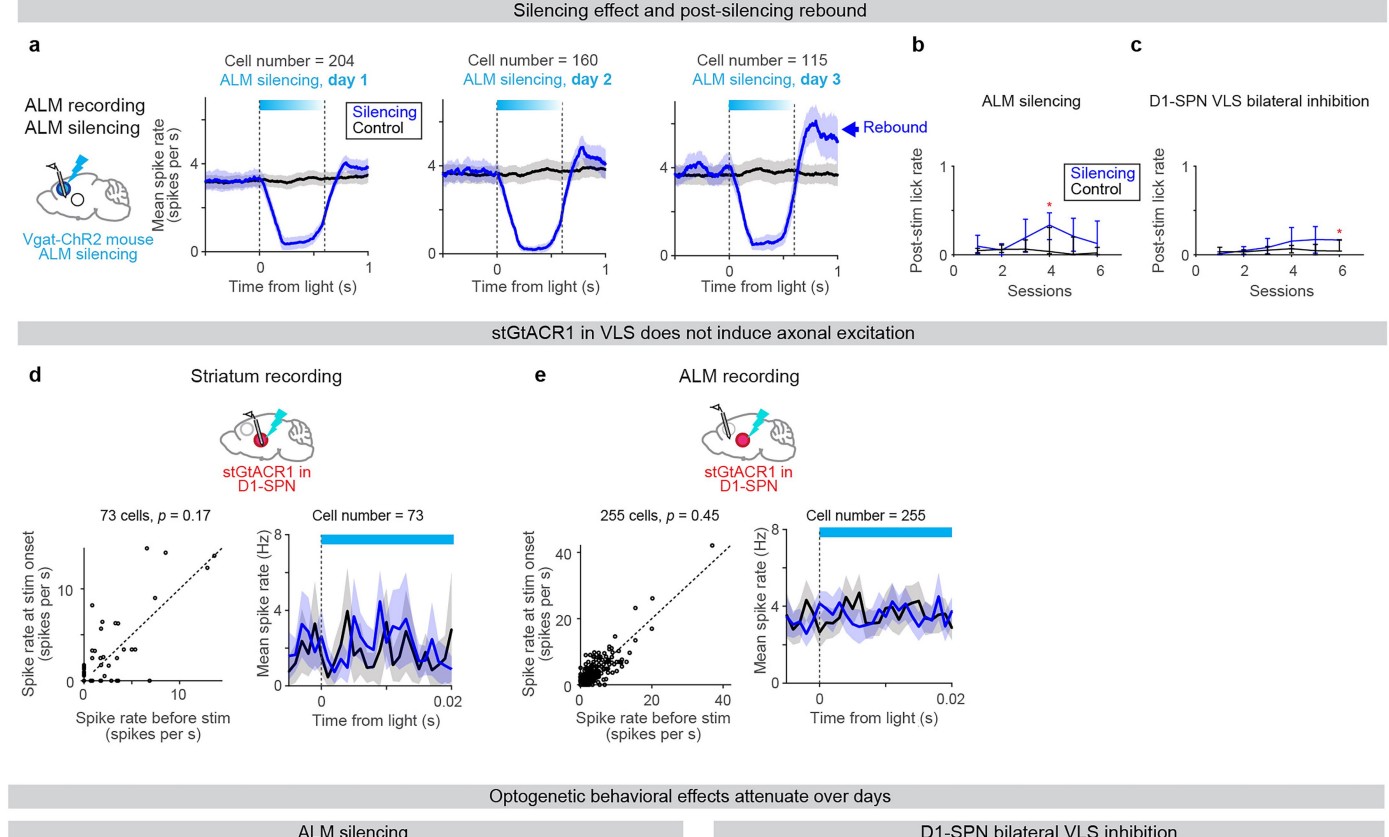

**Extended Data Fig. 6 | Characterization of optogenetic effect across sessions.**
Here, we characterized possible caveats of optogenetic manipulations. First,
we measured the post-silencing rebound activity and lick, over days (**a-c**). We
noticed that for ALM silencing, post-silencing rebound activity increased over
days (**a**), accompanied by the increase of post-silencing lick (**b**), which explains
the decrease in the shift of lick timing over sessions (**f-g**). Therefore, we
restricted our analysis to data from the first two days. Second, we confirmed the
lack of axonal excitation in both striatum and ALM when we used stGtACRl[47,76,77]
in D1-SPN (**d-e**). Third, we observed that the behavioral effect of D1-SPN inhibition
showed a drastic decline over sessions, so we focused our analysis on the data
from the first manipulation day (**h-i**). These observations are consistent
with the short-lived effect of manipulations observed in other species and
manipulations[48,78]. **a**. Mean spike rate of ALM putative pyramidal neurons
during ALM silencing (during no-cue trials) over 3 consecutive days (spike rate
was smoothed by a 200-ms boxcar causal filter). Lines, grand mean. Shading,
SEM (hierarchical bootstrap). Note the increase in the post-silencing rebound
activity over days (blue arrow). **b**. Post-stim lick rate (Proportion of trials with
lick within 600 ms following the photostimulation offset time in no-cue trials)
across days for ALM silencing. *$p < 0.001$ (hierarchical bootstrap with *Bonferroni*
correction for multiple comparisons; null hypothesis is that the post-stim lick
rate in photostimulation trials is lower than or equal to that in control trials).
Data are presented as mean values ± 95% CI. n = 14 mice. **c**. Same as in **b** but for
D1 VLS bilateral inhibition. Data are presented as mean ± 95% CI. n = 6 mice.

**d**. When stGtACR1 exhibits leaky expression at the axon, it can induce short-
latency axonal excitation at the onset of photostimulation[47,76,77]. Thus, we
examined the spiking activity at the onset of D1-SPN inhibition (during no-cue/
precue silencing trials) in VLS, which did not show signs of axonal excitation.
Left, spike rate during baseline (20 ms before the photostimulation) vs. spike
rate within 20 ms of the photostimulation onset for individual striatal neurons.
*P*-value, two-sided signed-rank test. Right, mean spike rate of SPNs (1 ms bin, no
smoothing of spike rate). n = 73 neurons. Shading, SEM (hierarchical bootstrap).
**e**. Same as in **d** but for ALM neurons during bilateral D1 VLS silencing. n = 255
neurons. Neither the striatum (**d**) nor ALM (**e**) showed signs of axonal excitation
caused by GtACR1. The brain atlas in panels **d,e** was adapted from the Allen
Institute for Brain Science (https://atlas.brain-map.org). **f**. The shift in lick time
caused by ALM silencing became weaker over consecutive days with ALM
silencing. Data are presented as mean ± 95% CI (hierarchical bootstrap). The
red dashed line, half of the effect observed in session 1. *P*-value, bootstrap with
a null hypothesis that the effect of delay silencing on session 1 is smaller than or
equal to the subsequent compared day. n = 14 mice. **g**. The change in no-lick
rate over consecutive days with ALM silencing. Data are presented as mean ±
95% CI (hierarchical bootstrap). *P*-value, bootstrap with a null hypothesis that
the effect of delay silencing on session 1 is smaller than or equal to the subsequent
compared day. n = 14 mice. **h-i**. Same as in **f-g** but for bilateral D1-SPN inhibition
in VLS. n = 6 mice.

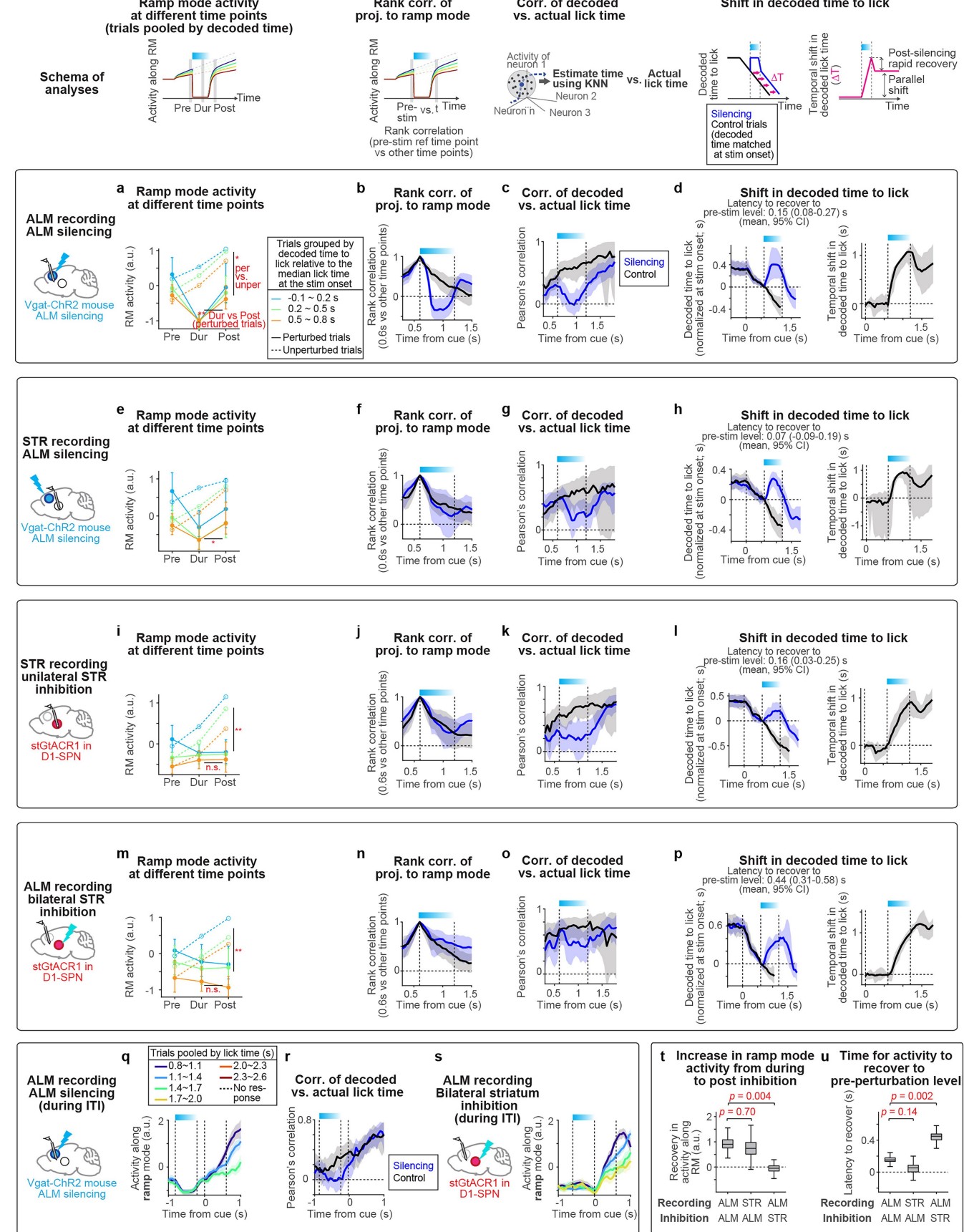

**Extended Data Fig. 7 |** See next page for caption.

**Extended Data Fig. 7 | Quantification of timing dynamics across perturbation experiments.** In addition to the panels shown in Figs. 4 and 5, here, we present additional quantifications of the timing dynamics across perturbation experiments. Firstly, we quantified the change in RM activity during and after transient perturbation. Upon ALM perturbation, the RM activity in both ALM and striatum showed a V-shaped profile, reflecting rapid silencing and rapid recovery of the activity (**a** and **e**). This contrasts with the linear decay profile observed in both areas upon D1-SPN inhibition (**i** and **m**). Secondly, the rank correlation of projection along the ramp mode collapsed in ALM during ALM silencing (**b**), but not in other conditions (**f**, **j**, and **n**). Thirdly, the decodability of lick time (based on kNN decoder) collapsed in ALM during ALM silencing (**c**), but not in other conditions (**g**, **k**, and **o**). Fourthly, the decoded time to lick was shifted during perturbation, and the shift persisted after the perturbation across conditions (**d**, **h**, **l**, and **p**). In contrast to silencing during the delay epoch, ALM silencing before the cue was followed by a recovery of decoded lick time (**r**). The rewind effect of D1-SPN inhibition is specific to dynamics after the cue as D1-SPN precue inhibition did not rewind ALM dynamics (**s**). **a**. Quantification of the change in RM activity during and after ALM silencing. RM activity at before (0.6 s after the cue, 'Pre'), during (0.9 s after the cue, 'Dur'), and after (1.25 s after the cue, 'Post') silencing is shown. Perturbed and unperturbed trials with matched $T_{to lick}$ at the silencing onset (0.6 s after the cue) are shown. Different colors, trials with different decoded $T_{to lick}$ relative to the median lick time in each session (to normalize for differences in median lick time across sessions). Dotted lines, unperturbed trials. Solid lines, perturbed trials. *P*-values, hierarchical bootstrap. **, $p < 0.005$; *, $p < 0.05$; n.s., non-significant. Mean ± 95% CI (hierarchical bootstrap). n = 308 neurons, 17 sessions, 14 mice. For the single-session population analysis, units with violated inter-spike interval were included. Same for **b**-**d**. **b**. Rank correlation of ALM activity along the ramp mode across trials. The rank order at the pre-silencing condition (0.6 s after the cue) is compared with that at other time points. Lines, grand mean. Shading, 95% CI (hierarchical bootstrap), applies to all the traces in this figure. The rank correlation in silenced trials (blue) collapsed during the silencing but recovered to the control (black) level afterward. **c**. Pearson's correlation of lick time decoded by the kNN decoder (based on ALM population activity) vs. the actual lick time. Lines, grand mean. Shading, 95% CI (hierarchical bootstrap). ALM activity predicted upcoming lick time before and after the silencing, but such a correlation disappeared during the silencing. **b** and **c** imply that after ALM silencing, the time information was recovered. **d**. Decoded $T_{to lick}$ from each time point based on kNN decoding analysis of population activity (left). Lines, grand mean. Shading, 95% CI (hierarchical bootstrap). The decoded $T_{to lick}$ was normalized by subtracting the decoded $T_{to lick}$ at stimulus onset to account for different lick times across trials and sessions (right). Across conditions, the decoded $T_{to lick}$ shifted during the perturbation, and the shift persisted after the perturbation, implying that the decoded $T_{to lick}$ is shifted in parallel between control and silencing trials. **e**-**h**. Same as in **a**-**d** but for striatum recording during ALM silencing. n = 297 neurons, 12 sessions, 7 mice. **i**-**l**. Same as in **a**-**d** but for striatum recording during unilateral D1-SPN inhibition in VLS. n = 133 neurons, 7 sessions, 5 mice. **m**-**p**. Same as in **a**-**d** but for ALM recording during bilateral D1-SPN inhibition in VLS. n = 243 neurons, 5 sessions, 5 mice. **q**. ALM ramp mode activity recovered after ALM precue silencing. Lines, grand mean. Shading, SEM (hierarchical bootstrap). **r**. Same as in **c** but for ALM recording during ALM precue silencing. Note that the correlation decayed during the silencing, but recovered after the silencing. **s**. Same as **q** but for ALM recording during bilateral D1-SPN precue inhibition. Note that it did not produce a rewind effect on ALM dynamics during precue. The brain atlas in panels **a**,**e**,**i**,**m**,**q**,**s** was adapted from the Allen Institute for Brain Science (https://atlas.brain-map.org). **t**. Increase in activity along the RM from the middle to the end of the inhibition period (summarizing **a**, **e**, and **m** for direct comparison across experimental conditions). ALM silencing leads to a significant recovery in RM activity by the end of inhibition, whereas bilateral D1-SPN (STR) inhibition does not show such recovery. The central line in the box plot, median. Top and bottom edges, 75% and 25% points. Whiskers, the lowest/highest datum within the 1.5 interquartile range of the lower/upper quartile. *P*-value, hierarchical bootstrap with a null hypothesis that the value is equal between conditions. STR recording during STR inhibition is not shown, as the manipulation is unilateral. **u**. Comparison of the time required for decoded time to return to the pre-perturbation level after the perturbation ends (summarizing **d**, **h**, and **p** for direct comparison across experimental conditions). *P*-value, hierarchical bootstrap with a null hypothesis that the value is equal between conditions. Following ALM silencing, activity rapidly recovers, whereas recovery is significantly delayed after D1-SPN (STR) inhibition.

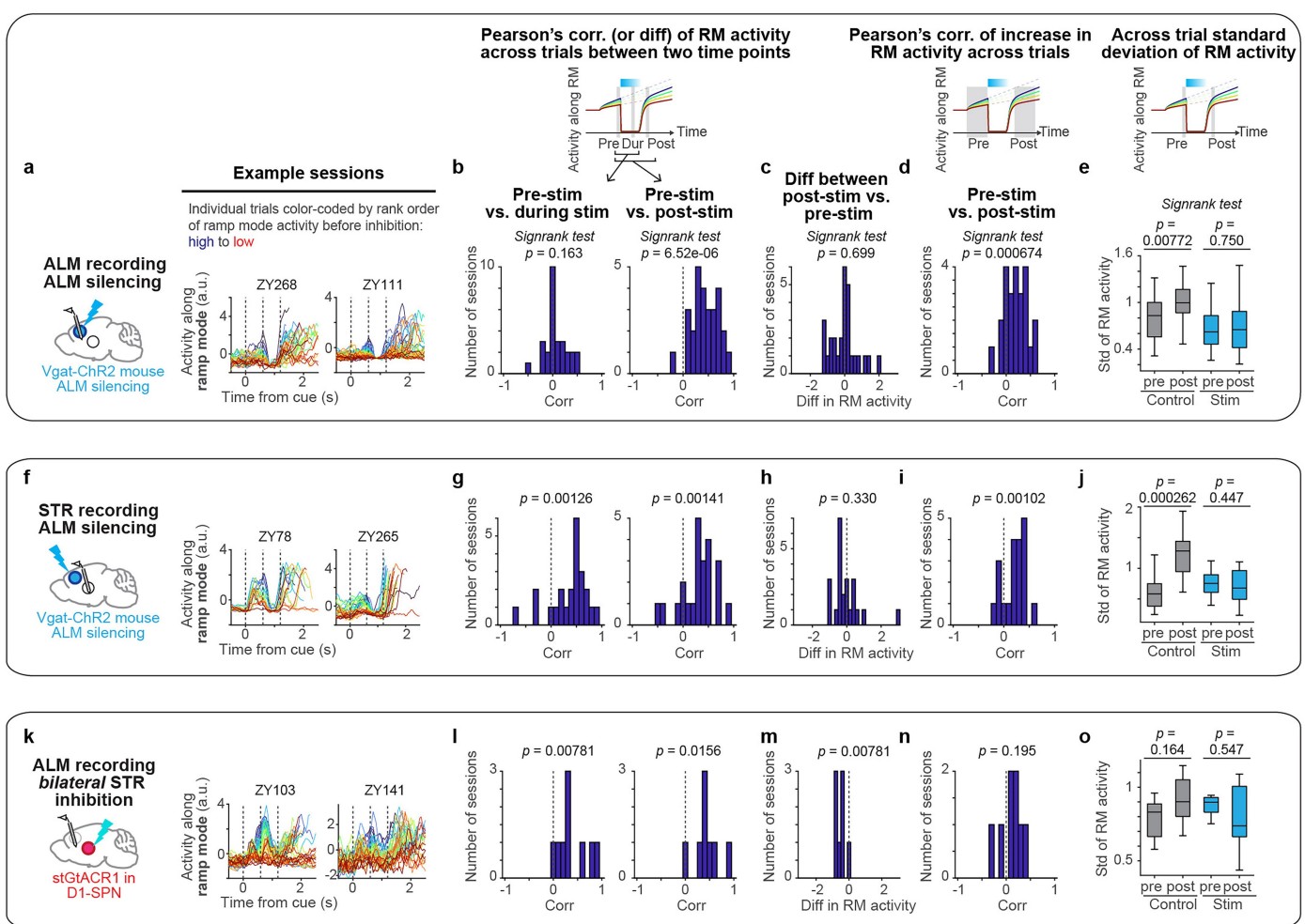

**Extended Data Fig. 8 | Perturbation effects at the single-trial level.** At the single-trial level, (1) ALM ramping activity collapses during ALM silencing but recovers to pre-silencing level afterward (**a**-**c**, **e**), and even the slope of ramping activity is correlated between the periods before and after ALM silencing (**d**); (2) striatal ramping activity retains information (remaining correlated with pre-silencing activity) during ALM silencing (**f**-**g**) and recovers to pre-silencing level afterward (**h** and **j**); and (3) ALM ramping activity gradually decay during D1-SPN inhibition reaching activity level lower than pre-perturbation level (**k**, **m**), while across-trial correlations are preserved during inhibition (**l**). **a.** Two example ALM recording sessions during ALM silencing, projected onto the ramp mode. Lines, individual trials color-coded by activity level before the silencing onset (blue, high; red, low). Although activity collapses during ALM silencing, the order of trials remains similar before and after silencing (quantified in **b**). **b.** Across-trial Pearson's correlation of ramp mode activity between two time points in all recorded sessions. Left, correlation between before (0.6 s after the cue) and during (0.9 s after the cue) the perturbation. Right, correlation between before and after the perturbation (1.25 s after the cue). ALM RM activity is collapsed during the silencing, but RM activity after perturbation was significantly correlated with RM activity before the perturbation. *P*-value, two-sided signed-rank test. n = 28 sessions, 18 mice (same for **c**-**e**). **c.** The mean difference in ALM RM activity between before and after the perturbation across all silencing trials. The difference was not statistically different from 0 (two-sided signed-rank test), implying that ALM RM activity recovered to pre-perturbation levels after ALM silencing. **d.** Across-trial Pearson's correlation

of the derivative (increase) in ramping activity before and after perturbation (We calculated how much the ramp activity increased during the first 0.6 s (dRM/0.6), and compared it to the ramp increase from the end of silencing to where RM activity reaches 2 in each trial). *P*-value, two-sided signed-rank test. The significant correlation indicates that the ramping speed is also recovered and remains correlated with the pre-perturbation condition after ALM silencing. **e.** Across-trial standard deviation of activity along the ramp mode before and after the perturbation. The central line in the box plot, median across sessions. Top and bottom edges, 75% and 25% points. Whiskers, the lowest/highest datum within the 1.5 interquartile range of the lower/upper quartile. *P*-value, two-sided signed-rank test. Across-trial standard deviation increased in control trials from 0.6 s to 1.2 s after the cue, but did not increase in ALM silencing trials, consistent with a "pause" effect of the timer, which suppresses not only the increase in mean (**c**) but also the variance of ramping activity. **f**-**j.** Same as in **a**-**e** but for striatum recordings during ALM silencing. Striatum activity also recovered to pre-stim level after ALM silencing. Unlike recording in ALM, striatum activity stayed correlated with pre-silencing activity even during ALM silencing (**g**). n = 23 sessions, 15 mice. **k**-**o.** Same as in **a**-**e** but for ALM recording during bilateral STR inhibition. ALM activity decayed during inhibition but stayed correlated with the pre-perturbation level during, and after perturbation (**l**). ALM activity did not recover to pre-stim level after STR inhibition (**m**). n = 8 sessions, 8 mice. The brain atlas in panels **a**,**f**,**k** was adapted from the Allen Institute for Brain Science (https://atlas.brain-map.org).

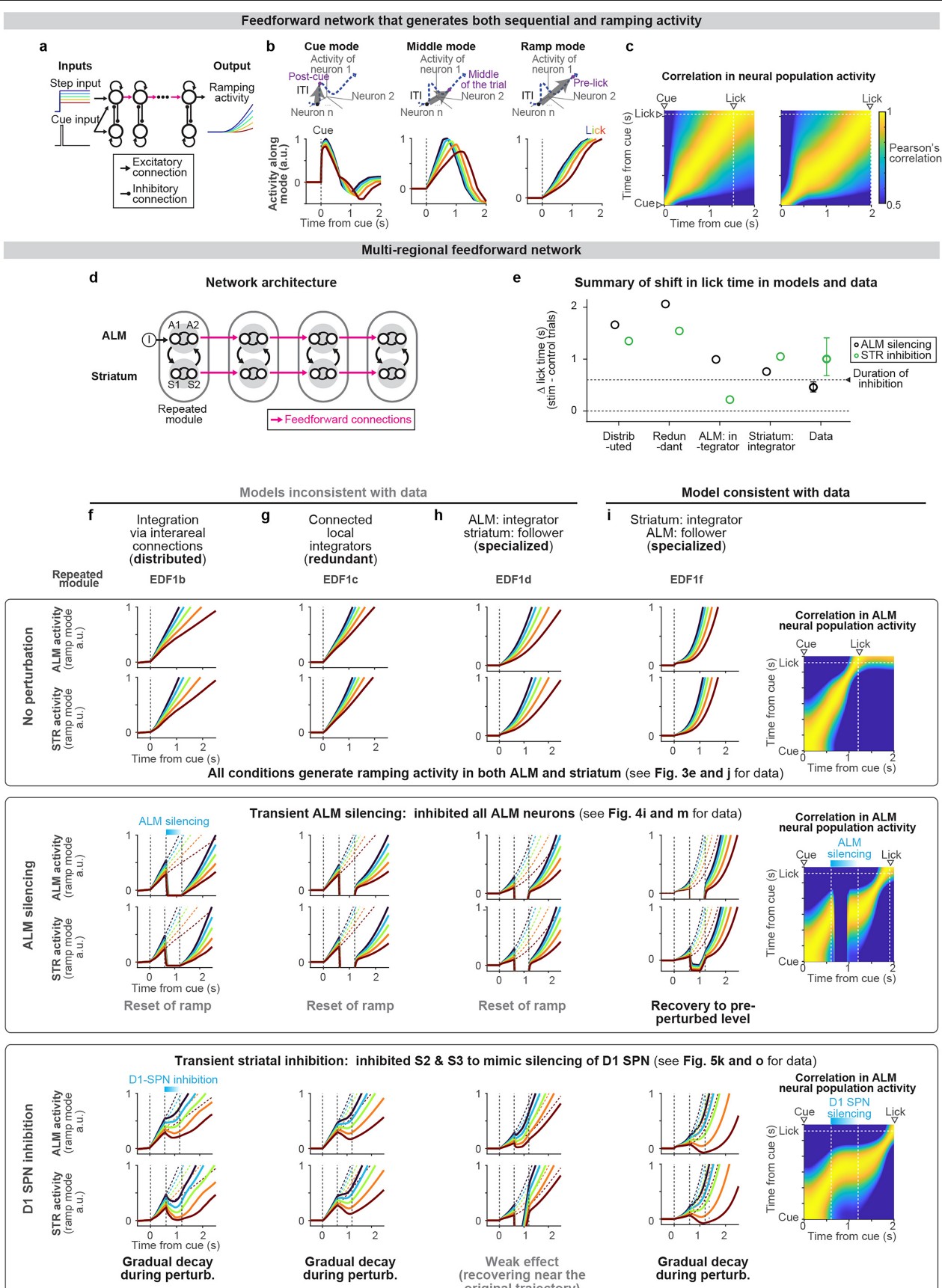

**Extended Data Fig. 9** | See next page for caption.

**Extended Data Fig. 9 | Multi-regional feedforward network that generates sequential and ramping activity.** Recurrent networks with feedforward connectivity can generate ramping activity[52] in response to step input. Activity in this network is high-dimensional and exhibits sequential changes in population activity patterns over time, akin to the data (**a-c**; see Methods). To implement a multi-regional feedforward network, the feedback network modules, similar to those in Extended Data Fig. 1, were duplicated with weaker recurrent connections to prevent perfect integration within each module. These quasi-integrators were then connected through feedforward connections, enabling both sequential and ramping activity (**d**). Each model has: a quasi-integrator implemented by both ALM and STR connected via feedforward connectivity in STR (**f**; distributed, akin to Extended Data Fig. 1b); redundant quasi-integrators and feedforward connectivity in both ALM and STR (**g**; redundant, akin to Extended Data Fig. 1c); a quasi-integrator and feedforward connectivity only in ALM, with STR acting as a follower (**h**; ALM integrator, akin to Extended Data Fig. 1d); and a quasi-integrator and feedforward connectivity only in STR, with ALM acting as a follower (**i**; STR integrator, akin to Extended Data Fig. 1f). See Supplementary Table 2 for connectivity matrix. To replicate ALM complete silencing, we injected a strong negative current into all ALM neurons. To replicate D1-SPN inhibition, we injected negative currents into half of the STR neurons. Consistent with the results in the positive feedback integrator (Extended Data Fig. 1), ALM functioning as an input/follower of the striatal integrator mimics the data, with a pause and rewind in the representation of time during ALM and striatal inhibition, respectively. **a**. Schema of the feedforward network. **b**. Activity along different modes in the network described in **a**. **c**. Correlation in neural population activity patterns in the network described in **a**. Trials with early lick (left) and late lick (right) are shown. Note that similar activity patterns emerge, but unfold at different speeds. **d**. A multi-regional network where the integrator module within ALM and striatum are connected with feedforward connections. **e**. Comparison of shifts in lick time following perturbation across feedforward models and data (Methods; the lick time was estimated in red traces in each plot). The last column (Data) was duplicated from Figs. 4b and 5d for comparison purposes. **f**. A multi-regional network where integration is via interareal connections between ALM and striatum (similar to Extended Data Fig. 1b). Perturbation of ALM resets the integration. Plots are shown up to the time of the lick. **g**. Similar to **f**, but a multi-regional network where both ALM and striatum function as integrators (similar to Extended Data Fig. 1c). Perturbation of ALM resets the integration. **h**. Similar to **f**, but a multi-regional network where ALM functions as the integrator (similar to Extended Data Fig. 1d). Perturbation of ALM resets the integration, while STR inhibition only produces a weak shift of the dynamics after the inhibition. **i**. Similar to **f**, but a multi-regional network where STR functions as the integrator (similar to Extended Data Fig. 1f). ALM silencing results in a pause in the representation of time (both in RM activity, left, and correlation, right). In contrast, STR inhibition decayed the RM activity.

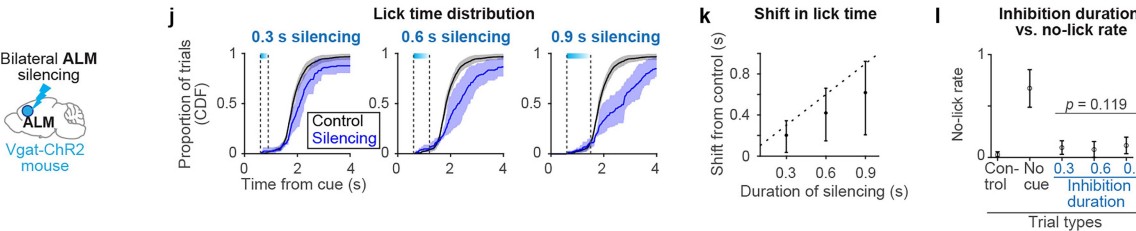

**Extended Data Fig. 10** | See next page for caption.

**Extended Data Fig. 10 | Prolonged rewinding of the timer leads to "no-lick" trials.** No-lick trials can be predicted based on ALM activity preceding the perturbation (**a** and **b**; the kNN-based decoded time was later in no-lick trials before the perturbation). Consistent with a rewind effect, prolonged D1-SPN inhibition caused a longer decay of ALM timing dynamics, shifting lick time more, and leading to more no-lick trials (**c-i**). In contrast, prolonged ALM silencing shifted lick time more without increasing no-lick trials (**j-l**). **a.** Three example ALM recording sessions with bilateral D1-SPN inhibition (0.25 mW) at 0.6 s after the cue onset. The relationship between actual lick time (x-axis) and decoded lick time (kNN) at 0.6 s after the cue (time before inhibition) is shown for both control (black) and inhibition trials (blue). Individual circle, individual trial. In no-lick trials, the decoded time to lick tended to be later than in lick trials. **b.** Summary across sessions. Decoded time to lick before the inhibition onset in lick vs. no-lick trials. The central line in the box plot, median. Top and bottom edges, 75% and 25% points. Whiskers, the lowest/highest datum within the 1.5 interquartile range of the lower/upper quartile. *P*-value, hierarchical bootstrap with a null hypothesis that the decoded time to lick in no-lick trials is smaller than or equal to the ones in lick trials. n = 8 sessions, 8 mice. No-lick trials can be predicted before the inhibition. **c.** Cumulative distribution of lick time in trials with bilateral D1-SPN delay inhibition (0.15 mW; used this power to avoid rebound excitation with the longer inhibition) with different durations (0.3, 0.6, or 0.9 s starting at 0.6 s after the cue). Black, control trials. Blue, inhibition trials. Shading, 95% CI (hierarchical bootstrap). n = 4 sessions, 4 mice, same for **d** and **e**. **d.** Change in median lick time in **c**. Data are presented as mean ± 95% CI (hierarchical bootstrap). Dashed line, unity line. Even with this low laser power, the shift in lick time is larger than the shift caused by ALM silencing (panel **k**) across conditions (*p* = 0.013, 0.021, 0.019 for 0.3, 0.6, and 0.9 s duration, respectively; hierarchical bootstrap with a null hypothesis that the lick time shift by ALM silencing is higher than or equal to that by D1-SPN inhibition). **e.** No-lick rate in **c**. Data are presented as mean ± 95% CI (hierarchical bootstrap). *P*-value, hierarchical bootstrap with a null hypothesis that the slope of the linear regression between no-lick rate and inhibition duration is smaller than or equal to 0. No-lick rate increases as a function of inhibition duration. **f.** ALM population activity along the ramp mode (RM; n = 96 neurons, 3 mice, same for **g-i**), trials pooled by lick time. Lines, grand mean. Shading, SEM (hierarchical bootstrap). Left, control. Right, D1-SPN inhibition trials with different durations. Note that ramping activity continues to decrease during D1-SPN inhibition. No-lick trials with a longer inhibition (0.9 s) show a robust decrease, and activity remains at the baseline level. This is consistent with a hypothesis that a decrease in ramping activity to baseline due to inhibition leads to the absence of a lick. **g.** Quantification of the decrease in ALM RM activity along the RM following bilateral D1-SPN inhibition, calculated as the difference in RM activity between the onset and end of inhibition. ALM RM activity continues to decay more with longer inhibition. Calculated based on lick trials in which the decoded time to lick before inhibition was between −0.1 and 0.2 s relative to the session's median lick time. Box plot: central line, median; box edges, 25th–75th percentiles; whiskers, lowest/highest values within 1.5 interquartile range of the lower/upper quartile, same for **h** and **i**. *P*-value, hierarchical bootstrap with a null hypothesis that there is no difference between 0.3 vs 0.9 s conditions. **h.** Quantification of ALM activity along the RM at the onset of bilateral D1-SPN inhibition in no-lick trials. Baseline before cue onset is subtracted. Even trials with high ALM RM activity can result in no-lick when D1-SPN inhibition is prolonged. *P*-value, hierarchical bootstrap with a null hypothesis that there is no difference between 0.3 vs 0.9 s conditions. **i.** Quantification of ALM activity along the RM at the end of bilateral D1-SPN inhibition in lick (L) and no-lick (N) trials. Baseline before cue onset is subtracted. At the end of inhibition, activity decayed to the baseline level in no-lick trials. *P*-value, hierarchical bootstrap. *: *p* < 0.05, with the null hypothesis that the value is lower than 0 with *Bonferroni* correction. **j-l.** Same as in **c-e** but for ALM delay silencing (1.5 mW) with different durations. n = 4 sessions, 4 mice. Silencing duration shifted lick time accordingly (consistent with a pause) but did not affect the no-lick rate, unlike D1-SPN inhibition. The brain atlas in panels **a,c,j** was adapted from the Allen Institute for Brain Science (https://atlas.brain-map.org).

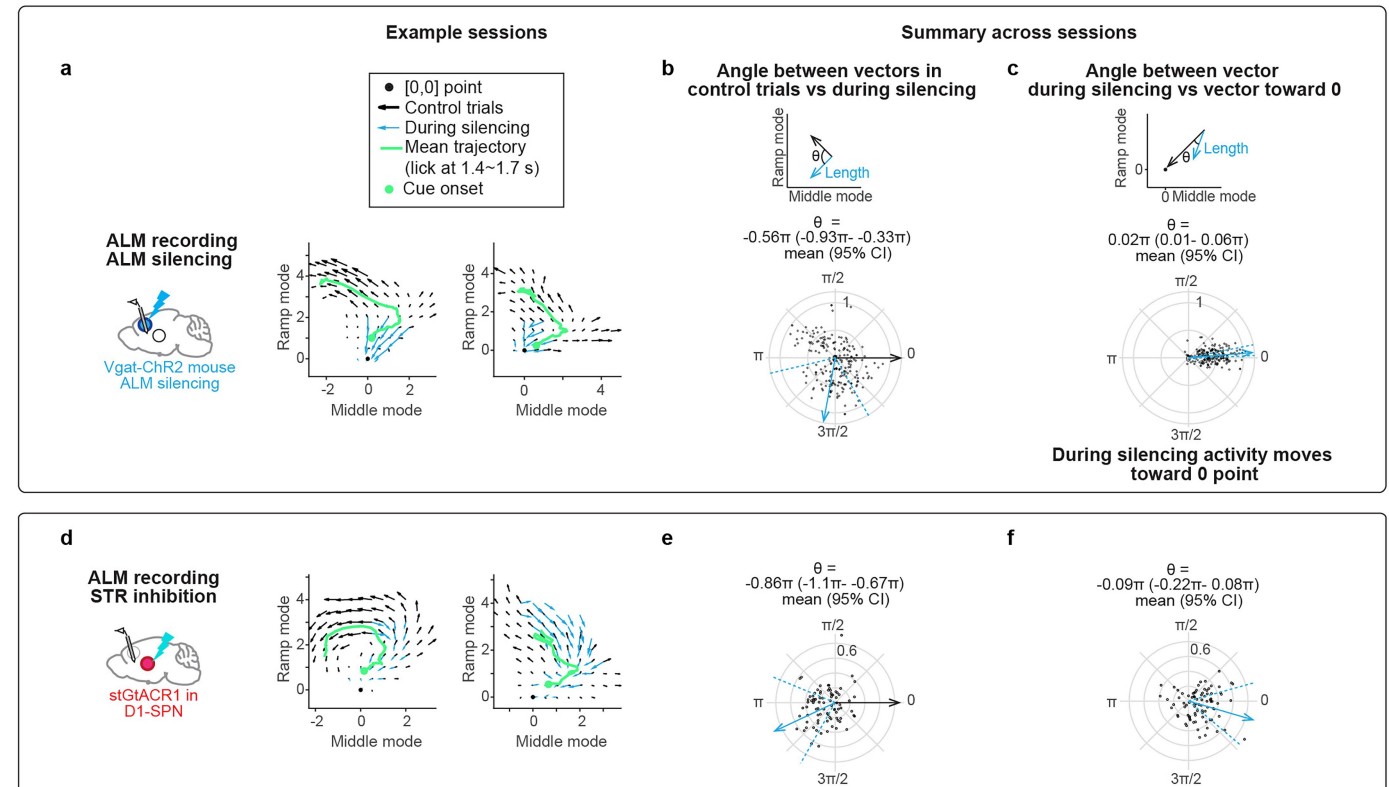

**Extended Data Fig. 11 | Rewinding of ALM dynamics during D1-SPN silencing in VLS.** We analyzed how activity evolves in the two-dimensional activity space defined by RM and MM, which captures a large fraction of task-modulated activity (Extended Data Fig. 3). **a** and **d**, two example sessions. For each session, we calculated the direction in which the activity developed from individual activity states. Specifically, we pooled all activity states (50 ms bin) between cue and lick in unperturbed control trials, calculated the direction of activity evolution (in the next 50 ms bin), and averaged these to acquire the vector field per session (black arrows in **a** and **d**; for states with more than 30 data points). Similarly, we acquired the vector field during inhibition by pooling all activity states during inhibition in silenced trials (cyan arrows; Methods). We then calculated the angle between vectors in control versus inhibition (**b** and **e**;

circles, individual states with both black and cyan arrows) to test whether activity evolved in the opposite direction from the normal trajectory during inhibition. Additionally, we calculated the angle between vectors during silencing and the vector toward the zero point (where spike rate is 0) to test whether activity evolved toward the zero point (**c** and **f**). During ALM silencing, the trajectory is better explained by activity moving toward zero (compare **b** vs. **c**). In contrast, during D1-SPN silencing, the trajectory cannot be solely explained by moving toward zero (**f**) and is likely a mixture of activity evolving in the opposite direction from the normal trajectory and moving toward 0. n = 28 sessions and 9 sessions in total for ALM silencing and D1 inhibition, respectively. The brain atlas in panels **a,d** was adapted from the Allen Institute for Brain Science (https://atlas.brain-map.org).

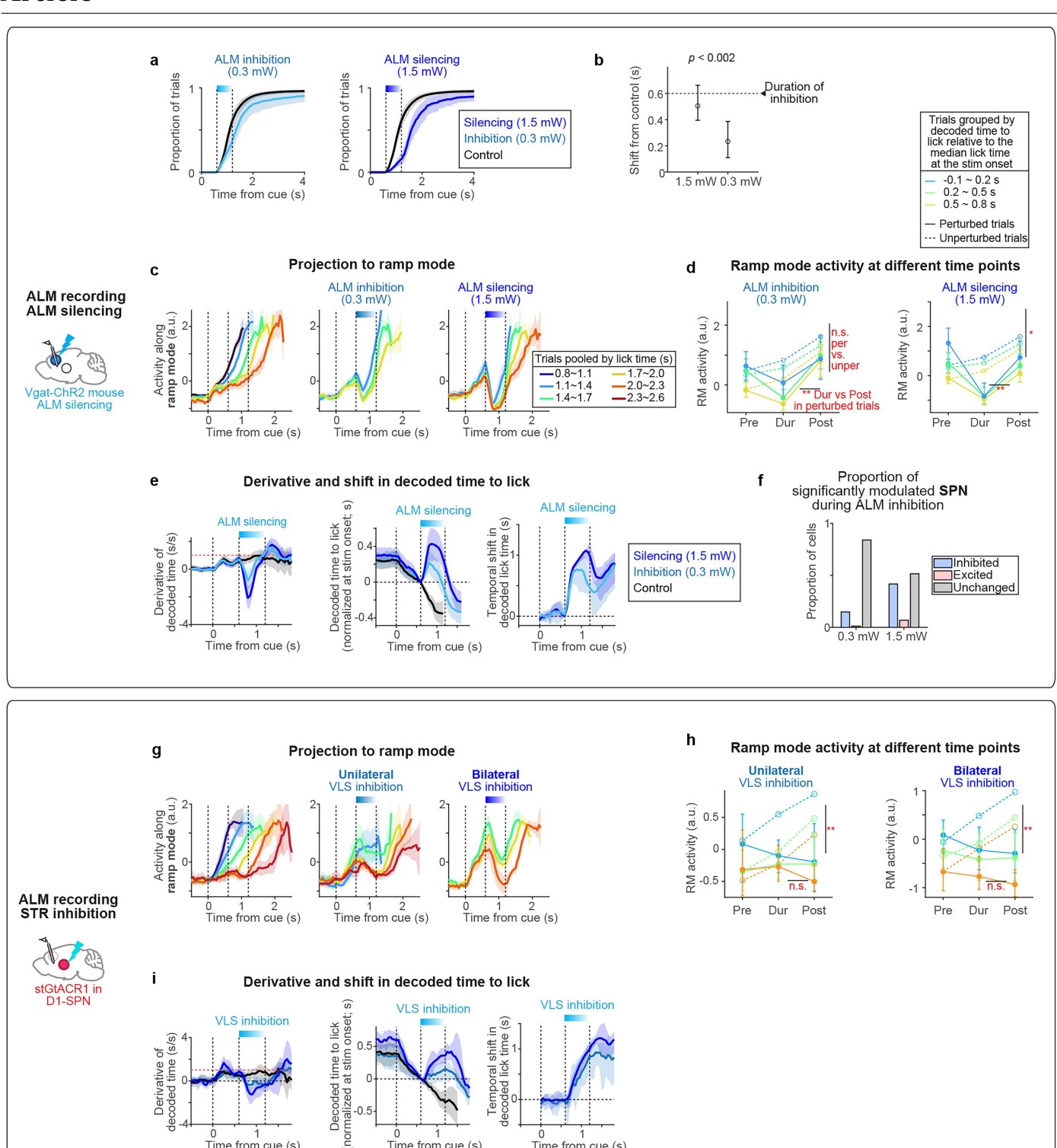

**Extended Data Fig. 12** | See next page for caption.

**Extended Data Fig. 12 | Graded and persistent effect of perturbations.** We recorded ALM activity while perturbing it with different laser powers (0.3 vs 1.5 mW; **a-f**) or while unilaterally (ipsilateral) or bilaterally inhibiting D1-SPN in the VLS (**g-i**). In both cases, the perturbation with different intensities resulted in graded and long-lasting changes in ramping activity and decoded time beyond the duration of perturbations, consistent with the notion that the perturbation was integrated into the timing dynamics. **a.** Cumulative distribution of lick time in trials with weak (left) or strong ALM delay silencing (right). Black, control trials. Blue, silencing trials. Shading, 95% CI (hierarchical bootstrap). n = 23 sessions, 12 mice, same for **b**. **b.** Change in median lick time in **a**. Data are presented as mean ± 95% CI (hierarchical bootstrap). *P*-value, hierarchical bootstrap with a null hypothesis that there is no difference between the lick time shifted by weak and strong perturbations. Dashed line, silencing duration (0.6 s). **c.** ALM population activity along the ramp mode in control trials (left) and ALM silencing trials with different powers (middle, right). Lines, grand mean. Shading, 95% CI (hierarchical bootstrap). n = 256 cells, 12 mice. Trials with different laser powers were randomly interleaved in the same sessions. **d**. Quantification of the change in RM activity during and after ALM silencing. Same format as in Extended Data Fig. 7a. Left, weak ALM silencing. Right, strong ALM silencing. Note that the plot of perturbed trials exhibited a V-shaped profile regardless of laser power, whereas in D1-SPN inhibition (**h**), it showed a linear decay profile regardless of whether the inhibition was unilateral or bilateral. **e**. Analysis of decoded time based on ALM population activity (using kNN decoder). Left: derivative of decoded time (computed over a 200 ms window). In control trials (block), the derivative changes from 0 to 1 (red horizontal dashed line) after the cue. In stimulated trials, the derivative becomes negative at light onset, reflecting rapid decay in activity, and increases after the light, reflecting rapid recovery. Middle and right: same format as in Extended Data Fig. 7d. Lines, grand mean. Shading, 95% CI. **f**. The proportion of striatal projection neurons with spiking activity significantly inhibited or excited ($p < 0.05$; Unchanged, $p >= 0.05$; two-sided rank-sum test) by ALM silencing with weak (left) and strong (right) power. Neurons with a mean spike rate above 1 Hz in control trials during the silencing window were analyzed (169 striatal projection neurons, 23 sessions, 12 mice). The fraction of SPN significantly inhibited by weaker ALM inhibition (15% of neurons; 25 out of 169) is significantly less than the fraction of SPN inhibited by the D1-SPN inhibition (36% of neurons; 9 out of 25, $p = 0.04$, chi-square test). Notably, weak ALM inhibition did not produce a rewind of ramping dynamics (**d** left; ALM ramping activity at the end of inhibition is not significantly different from the control condition, unlike in D1-SPN inhibition in **h**). Therefore, the rewind effect observed with striatum inhibition cannot be attributed to a quantitative difference in the extent of striatal inhibition, but rather reflects a qualitative difference between the targeted brain areas and cell types. **g-i**. Same as in **c-e** for unilateral vs. bilateral inhibition of D1-SPNs in VLS. Data from different mice were combined (bilateral data is duplicated from Fig. 5o; unilateral data, n = 113 cells, 4 mice). Unlike ALM silencing, the derivative of decoded time does not show a large change even during inhibition, reflecting a gradual change in timing dynamics. The brain atlas in panels **c,g** was adapted from the Allen Institute for Brain Science (https://atlas.brain-map.org).

# Reporting Summary

## Statistics

For all statistical analyses, confirm that the following items are present in the figure legend, table legend, main text, or Methods section.

| n/a | Confirmed | |
|---|---|---|
| ☐ | ☒ | The exact sample size ($n$) for each experimental group/condition, given as a discrete number and unit of measurement |
| ☐ | ☒ | A statement on whether measurements were taken from distinct samples or whether the same sample was measured repeatedly |
| ☐ | ☒ | The statistical test(s) used AND whether they are one- or two-sided *Only common tests should be described solely by name; describe more complex techniques in the Methods section.* |
| ☐ | ☒ | A description of all covariates tested |
| ☐ | ☒ | A description of any assumptions or corrections, such as tests of normality and adjustment for multiple comparisons |
| ☐ | ☒ | A full description of the statistical parameters including central tendency (e.g. means) or other basic estimates (e.g. regression coefficient) AND variation (e.g. standard deviation) or associated estimates of uncertainty (e.g. confidence intervals) |
| ☐ | ☒ | For null hypothesis testing, the test statistic (e.g. $F$, $t$, $r$) with confidence intervals, effect sizes, degrees of freedom and $P$ value noted *Give P values as exact values whenever suitable.* |
| ☒ | ☐ | For Bayesian analysis, information on the choice of priors and Markov chain Monte Carlo settings |
| ☐ | ☒ | For hierarchical and complex designs, identification of the appropriate level for tests and full reporting of outcomes |
| ☐ | ☒ | Estimates of effect sizes (e.g. Cohen's $d$, Pearson's $r$), indicating how they were calculated |

*Our web collection on statistics for biologists contains articles on many of the points above.*

## Software and code

Policy information about availability of computer code

| Data collection | spikeGLX, BPod |
|---|---|
| Data analysis | Matlab R2020b, R2022b, Python 3, JRClust, DeepLabCut2.3.5 |

For manuscripts utilizing custom algorithms or software that are central to the research but not yet described in published literature, software must be made available to editors and reviewers. We strongly encourage code deposition in a community repository (e.g. GitHub). See the Nature Portfolio guidelines for submitting code & software for further information.

## Data

Policy information about availability of data

All manuscripts must include a data availability statement. This statement should provide the following information, where applicable:

- Accession codes, unique identifiers, or web links for publicly available datasets
- A description of any restrictions on data availability
- For clinical datasets or third party data, please ensure that the statement adheres to our policy

The recording data in NWB format is available at the DANDI archive (ID 001610). Codes is available at https://github.com/inagaki-lab/Yang_et_al_2024

# Research involving human participants, their data, or biological material

Policy information about studies with [human participants or human data](link). See also policy information about [sex, gender (identity/presentation), and sexual orientation](link) and [race, ethnicity and racism](link).

| | |
|---|---|
| Reporting on sex and gender | N/A |
| Reporting on race, ethnicity, or other socially relevant groupings | N/A |
| Population characteristics | N/A |
| Recruitment | N/A |
| Ethics oversight | N/A |

Note that full information on the approval of the study protocol must also be provided in the manuscript.

# Field-specific reporting

Please select the one below that is the best fit for your research. If you are not sure, read the appropriate sections before making your selection.

☒ Life sciences        ☐ Behavioural & social sciences        ☐ Ecological, evolutionary & environmental sciences

For a reference copy of the document with all sections, see [nature.com/documents/nr-reporting-summary-flat.pdf](http://nature.com/documents/nr-reporting-summary-flat.pdf)

# Life sciences study design

All studies must disclose on these points even when the disclosure is negative.

| | |
|---|---|
| Sample size | The sample sizes are matched to the standadrd in the field (~6 mice per condition; e.g., Li et al, Nature, 2015 ) |
| Data exclusions | Quality metrics were used to determine single units to be analyzed. See Methods for details. |
| Replication | All experiments were performed extensively, as reported in the manuscript (e.g., ALM silencing and D1 SPN silencing during delay were tested for 3 times , n> 4 mice each, independently) and hierarchical bootstrap was used to rigorously test for reproducibility among animals. |
| Randomization | All trial types were randomly determined by the program |
| Blinding | For all experiments (behavior and recording experiments) were fully controlled by computer programs and experimenters did not have control to bias results. Experimenters were blind to experiment during the spike sorting. |

# Reporting for specific materials, systems and methods

We require information from authors about some types of materials, experimental systems and methods used in many studies. Here, indicate whether each material, system or method listed is relevant to your study. If you are not sure if a list item applies to your research, read the appropriate section before selecting a response.

## Materials & experimental systems

| n/a | Involved in the study |
|---|---|
| ☒ | ☐ Antibodies |
| ☒ | ☐ Eukaryotic cell lines |
| ☒ | ☐ Palaeontology and archaeology |
| ☐ | ☒ Animals and other organisms |
| ☒ | ☐ Clinical data |
| ☒ | ☐ Dual use research of concern |
| ☒ | ☐ Plants |

## Methods

| n/a | Involved in the study |
|---|---|
| ☒ | ☐ ChIP-seq |
| ☒ | ☐ Flow cytometry |
| ☒ | ☐ MRI-based neuroimaging |

## Animals and other research organisms

Policy information about studies involving animals; ARRIVE guidelines recommended for reporting animal research, and Sex and Gender in Research

| | |
|---|---|
| Laboratory animals | We used five mouse lines: C57Bl/6J (JAX# 000664), VGAT-ChR2-EYFP87 (JAX #14548), Drd1-cre FK15074, Adora2-cre KG12674, R26-LNL-GtACR1-Fred-Kv2.188 (JAX #33089). See Extended Data Table 1 for mice used in each experiment.  For all experiments, age > P60. |
| Wild animals | No wild animals were used in the study. |
| Reporting on sex | This study is based on both adult male and female mice (age > P60). |
| Field-collected samples | No field collected samples were used in the study. |
| Ethics oversight | All procedures were in accordance with protocols approved by the MPFI IACUC committee. Mice were housed in a 12:12 reverse light: dark cycle and behaviorally tested during the dark phase. Temp 74F. Humidity 35-60% |

Note that full information on the approval of the study protocol must also be provided in the manuscript.

## Plants

| | |
|---|---|
| Seed stocks | N/A |
| Novel plant genotypes | N/A |
| Authentication | N/A |

