## [Peer Review File · Nature]

Integrator dynamics in cortico-basal ganglia loop for flexible motor timing

Corresponding Author: Dr Hidehiko Inagaki

Version 0:

Reviewer comments:

Referee #1

(Remarks to the Author)

The paper explores the mechanisms that underlie ramping activity in the mouse frontal cortex (ALM) and striatum during a flexible timing task. The authors tested how perturbations of ALM and striatum affect behavior and ramping dynamics. While both areas display qualitatively similar signals, perturbations of each had different effects on behavior and dynamics. Based on these effects, the authors conclude that inhibiting ALM “pauses” the timer whereas inhibiting striatum “rewinds” the timer. They also use network models to deduce that the striatum integrates activity in ALM to generate ramping activity and that ramping activity in ALM is then inherited from the striatum (through a multi-synaptic pathway).

The main result—that perturbations of ALM and striatum cause different effects despite similar activity—is compelling and well-supported. An increasingly popular viewpoint in neuroscience is that computations are distributed across brain regions (“everything everywhere”). The results are an important counterpoint and will thus be of wide interest.

The paper has a few important shortcomings. First, the evidence for the authors’ interpretation that inhibiting ALM and striatum pauses and rewinds the animal’s timer, respectively needs stronger support to be convincing. Second, the conclusion that the striatum integrates ALM activity is not directly verified. Third, the hypothesis about integration/inheritance is not well supported. I elaborate on these issues below:

1) The hypothesis that inhibiting ALM pauses the timer needs better evidence. The authors use two approaches to test this: (a) a comparison of activity before the perturbation to that after, and (b) an analysis of the effect of the perturbation on behavior.

A) For (a), the logic is that if the perturbation induces a “pause,” then the post-perturbation activity should return to its pre-perturbation level. Based on my reading, the authors do not directly test this prediction. Instead, they group trials based on the animals’ ultimate lick time and perform analyses on the average activity within these groups. Interpreting these analyses is difficult because the lick time is impacted by the perturbation. It is also not possible to compare dynamics in laser vs no-laser trials with this grouping, because the conditionalization is different for each trial type. Ideally, the authors would be able to leverage their large-scale recordings to do single-trial analysis of the dynamics. But, if grouping trials is necessary, then a more logical way is to group trials by the level of activity in the ramp mode at the time of laser onset (on laser trials) or the equivalent time on no laser trials. This would allow for a direct test of whether the post-perturbation activity returns to its prior level, and for a direct comparison between activity in laser and no-laser trials. It also seems to me that analyzing the variance of activity along the ramp mode would be informative. Without a perturbation, the variance should increase as a function of time in the delay. What is the variance across trials before the laser vs after? Is it also consistent with a pause in the timer?

B) The authors could better leverage the perturbation’s effect on behavior to provide evidence for their claims. A “pause” in the timer makes a few, straightforward predictions about what should occur in the behavior. First, the animals should take longer to lick (as observed) without the increase in variability predicted by the scalar property of timing. Second, the change in behavior should not depend on the state of the timer. The clearest way to test this is to do the same perturbation at different times in the delay. Alternatively, the authors can group lick times based on the level of activity in the ramp mode (as described above). The change in behavior after perturbation should be the same for each group. Confirmation of these

straightforward predictions would make the results convincing.

2) The hypothesis that inhibition of the striatum “rewinds” the timer may not be consistent with the data. First, one of the most salient effects of striatum inhibition is to substantially increase the number of trials in which the animal doesn’t lick, which does not seem consistent with a rewind. The authors do not justify dismissing this effect. Second, with the huge effect of striatal inhibition on lick times, it is difficult for me to interpret the lick-time conditioned analyses in Figure 6. Grouping based on activity (as explained above) would better show how the perturbation influences dynamics. It might also shed light on the no lick trials (i.e. does pre-perturbation activity predict a “no lick” trial). More generally, the rewind hypothesis feels less formulated than the pause hypothesis (which makes straightforward predictions). Given the work the authors put into building models, it would be helpful to see the exact same analyses applied to both the real data and data simulated from the model that has a rewind. Without this, it is not intuitive what one should expect to see from these specific analyses if the hypothesis were correct. A final, related point again pertains to the behavioral effects. My intuition is that a rewind should have behavioral effects that do depend on the state of the timer (unlike a pause), though I am not confident in this. In summary, the paper would be improved if the authors worked out more precise, quantitative predictions for the rewind hypothesis and tested them in neural and behavioral data.

3) The hypothesis that the striatum integrates ALM activity to generate ramping activity and that ALM inherits its ramping from the striatum is not tested and thus not supported by the data. It appears that the authors generated a post hoc hypothesis after the data was collected and did not go back to directly test the hypothesis. As is, the conclusions seem far too strong given the evidence.

A) It’s not clear if there is any evidence for an integration process in the data. Based on my reading, it seems that the authors instead assumed there was an integration process because previous models have shown that it is possible to produce ramping through integration. But, other models have shown that ramping can be produced through other means. Why should we conclude that ramping in the striatum is produced through integration? Indeed, one of the major claims in the discussion is that an integrator mediates the generation of timing dynamics. However, no alternatives to integration are considered or tested.

B) The general conclusion that the authors put forth is not clear to me. I believe it is as follows: the trial history mode in ALM supplies an input to the striatum that is integrated within the striatum to produce ramping dynamics. These ramping dynamics are then inherited by ALM through a multi-synaptic pathway. This took many reads to understand. Regardless of whether my read is correct, I had a lot of trouble figuring out what the authors actually thought was going on. A more general audience will have an even harder time understanding the authors’ conclusions/hypotheses.

C) Why doesn’t striatum also integrate the ramping dynamics in ALM? The trial history mode and the ramp mode were forced to be orthogonal for convenience in the analyses, and they are assumed to be orthogonal in the model. But are the signals actually present in orthogonal subspaces in ALM?

D) There doesn’t seem to be any evidence presented that suggests that striatum is specifically integrating the trial history mode in ALM. Could the simultaneous recordings be used to test this idea? I.e., do fluctuations in the trial-history mode in ALM correlate best with those in the ramp mode in striatum?

E) If ramping in ALM is inherited from the striatum, then there should be a strong trial-by-trial correlation between fluctuations in the ramping of the two areas. The authors did perform one analysis showing that a lick-time decoder trained on the two areas was correlated across trials. This is a good start, but if ramping in ALM is inherited through a multi-synaptic pathway, then there should be a significant lag in the correlations between the two ramping modes. The direction of the lag should be in the opposite direction for correlations between the trial-history mode (or its integral) in ALM and the ramp mode in the striatum.

F) It is unclear to what degree the favored model actually fits with the data. As mentioned above it would be helpful to see the exact same analyses performed on both the real data and the model simulated data so that the reader can directly compare them.

Minor comments:

1) The rainbow (i.e. “Jet”) colormap (used in all of the heat maps) is known to not accurately represent quantitative data and produces illusionary boundaries (D. Borland and R. M. Taylor II, "Rainbow Color Map (Still) Considered Harmful," in IEEE Computer Graphics and Applications). The colormap may also be confused with the colors used in the PSTHs.

2) The definition of “ramping” neurons (¶ starting at line 183) is not convincing. Many non-ramping activity profiles would produce differences in activity between the lick epoch and baseline (e.g., oscillations, motor bursts, etc.).

3) The paper from O’Shea, Duncker, et al (2022), BioRxiv, is highly relevant and should be cited.

(Remarks on code availability)

Referee #2

(Remarks to the Author)

Major comments:

1) The underlying premise of the work is that the timing behavior observed is due to an integration of a step-like (constant) input into a ramp. The step-like input is further speculated to possibly be the same step observed during the pre-cue ‘trial-history’ period, which is then gated into being able to be integrated by the presentation of the cue. These seem like

reasonable hypotheses, but it would be nice if they could be more directly supported by data.

a) The first assumption – that there is a step-to-ramp integration – seems like it is critical for the framework of analysis and modeling. Is there a way to more directly test this hypothesis? For example, if on occasional trials the cue was made stronger or longer, might this lead to greater ‘on manifold’ input that would then be integrated and could be seen experimentally?

b) Could the trial history activity instead just shift the initial condition of the network, thus leading to different ramping dynamics (rather than serving as a constant input to an integrator)? This possibility is mentioned at the start of the text on the trial-history activity, but then dropped in favor of the integrator interpretation, but I wasn’t sure why it was dismissed.

2) I love the modeling idea that ALM serves as the input to an integrator involving the striatum and also as an output to a pattern of ALM activity that does not get integrated by the striatum. However, at least in the model of Extended Figure 1, this idea is implemented by having an ALM population that should be a pure constant step of activity without showing ramping (neuron A1 in Extended Fig. 1f). Is there evidence for this in the data? If, instead, one should think of populations A1 and A2 as two more general patterns of activity that can be intermixed at the level of recorded neurons’ activity (i.e. a given neuron may reflect a certain amount of pattern 1 and a certain amount of pattern 2, in a manner that reflects the experimental recordings), is it still possible to make a model in which ALM is an input and an output in a manner that recapitulates the data but doesn’t form a feedback loop and thus act as part of the integrator?

3) The alternative model of sequences in Extended Data Fig. 11 instead has the external input going to the striatum. However, this model seems to have ALM being part of the integrator (but only weakly due to very weak interconnections between ALM and striatum), as I think is evidenced by the activity decay seen during ALM silencing. Also, it has some strange dynamics where activity can either stay constant or increase during D1 SPN inhibition and it seems to have a very different cue mode (at least in panel b) from the data. It would be helpful to discuss these discrepancies and whether they are contradictory to the data.

4) The models have the striatum driving (exciting) ALM. However, as noted in the text, the experiments in Figure 6e suggest that silencing striatum has little net effect on ALM activity. How does this fit with the models?

5) The models are hard to interpret due to having ‘arbitrary units (a.u.)’. In the methods, it is stated that striatal activity starts from a baseline of 5 Hz, corresponding to experimental recordings. But, then, ‘arbitrary units’ are used. First, are the ‘arbitrary units’ the same for ALM and STR activity and for the different cases d-g? I appreciate that these are population activities, so the same population activity could be due to high firing rates of a smaller number of neurons or low firing rates of a larger number of neurons, but it is very hard to compare the models to data or to each other with such arbitrary (and possibly changing from network to network or even from one population to another?) units. Can the authors instead show the real units that emerge from the model, especially given the methods compares the model to 5 Hz striatal firing rates, and then just explain that these are population values. Most critically here is to have the same, unnormalized units for all nodes of the network and for all simulations so one can get a better idea of what is happening.

6) Is it possible that striatum could instead be an input to, and readout of, the integrator, similar to how ALM is being interpreted but with one difference: in order to explain the rewinding, perhaps the zero point of the integrator occurs when the striatal activity is at baseline, and then silencing of the striatum is effectively providing negative input to the integrator to cause rewinding? For this to work, one would need a model in which (like the ALM model) some striatal neurons (or possibly activity patterns, as discussed in point #2 above) are inputs to the integrator and some are readouts. Anyway, the point here is that, if one considers silencing of the striatum as being negative input, then one could imagine striatum being analogous to how ALM is described in the current model. Is there a contradiction with this possibility?

7) Why isn’t there a plot analogous to Fig. 6g for the ALM silencing experiments of Fig. 5, rather than a cartoon? In Extended Fig. 12b, it’s nice that the activity appears to move parallel to a vector going towards zero in panel c, but it also appears to move somewhat backwards (between 90 and 180 degrees) relative to during control in panel b. It’s not obvious to me that the vector towards 0 is orthogonal to the integrating ‘on manifold’ direction, so the result in panel c doesn’t seem to preclude ALM being partly involved in rewinding of integration.

8) I agree with the authors statement in the discussion that the medium spiny neurons of the striatum seem like an unlikely location for integration on their own. However, it would be nice to provide additional justification for why the authors think the integrator is a subcortical network involving the striatal neurons as opposed to some other set of pathways involving the striatum—is this based on prior inactivation experiments ruling out other neocortical areas, so that it needs to be subcortical? A bit more justification for likely candidates for the feedback pathways would be helpful, and also for the idea of the striatum mediating a mutually inhibitory positive feedback loop between striatal-associated populations that receive sufficiently strong common excitatory input from one ALM population, as in Extended Fig. 1f, in order to not have one population shut off, effectively disconnecting the mutually inhibitory feedback loop.

9) It would be worth discussing (and probably referencing right up front in the introductory paragraphs motivating the current work) more about previous work that likewise suggested neocortical areas to perhaps not be the integrator but striatum being part of the integrator. I’m thinking in particular of the set of papers from the Brody lab that also used recordings, perturbations, and modeling to come to this conclusion in the context of a different, accumulation-of-evidence based decision-making task.

Minor comments:

1) In Extended Fig. 1, the drawings of the model networks should accurately convey the weight matrix. For example, in the

critical 'consistent with data' model, it really is A1 and A2 neurons with self-connections, in which A1 drives S1 and A2 drives both S1 and S2, with only S1 projecting back to A2. This is not remotely obvious from the figure, but is critical for understanding the model and (as noted in Major comments above) critically evaluating whether it does, or does not, reflect the data. When doing this, different notation should be used for excitatory vs inhibitory connections.

2) For figures like Fig. 2b, it would be nice to show an average session plus error bars rather than just an example session (I understand that panel c is cumulative, but it shows somewhat different information if I'm understanding it correctly.).

3) I couldn't understand what cue, middle, ramp mode meant until reading the methods, yet they are critical for the analyses. If possible, try to define briefly in the main text.

4) The trial history mode was even more difficult to understand, as the main text refers to the methods for understanding it and then the methods refers back to Fig. 2d (which beyond a regression formula, doesn't describe it either). Please define clearly in the methods (and, ideally, include a simpler brief explanation in the main text).

5) I'm not a huge fan of the phrase 'similarity matrices' as I think of a similarity matrix as one with the same x and y axes, as in representational similarity analysis matrices. Just calling it correlation as in Fig. 5 seems better. And, as with 'modes', it would be good to define what this graph is more clearly in the main text (it wasn't even obvious until I read the methods that each row is a neuron, given that the y-axis is labeled as 'reference trial', which suggests that each row is a trial).

6) More terminology: 'variance' of spiking activity explained is not really a variance, as it is relative to baseline activity rather than relative to a mean activity. Please rephrase to more accurately convey what is being plotted.

7) Is the value of I_{ext} given for the models? More generally, please check that all parameter values are given somewhere numerically (and not just with a color bar, as in the feedforward network matrices).

8) I didn't understand why gray line in Extended Fig. 4e was compared to the 3rd quartile.

9) Is it surprising that the correlations in Extended Data Fig. 5 go to nearly 1 for potentially noisy spiking data? Or does this reflect some previous averaging out or removal of noise in the data analysis?

10) Extended Data Fig. 8a. I think this should say "cumulative" lick time distribution.

11) Extended Data Figs. 9d and 12a,d captions do not separately refer to/describe what is plotted in the left versus right panels.

12) In Extended Data Fig. 11e, the coupling should perhaps be called 'very' weak (and this is a case where somewhere giving numbers rather than just colors for matrix values would be very helpful).

(Remarks on code availability)

Referee #3

(Remarks to the Author)

This manuscript studies the roles of ALM and striatum in a lick timing task. It reaches the conclusion that ALM and striatum have different roles, with striatum serving as an integrator and ALM providing inputs into this integrator. The paper uses a combination of electrophysiology and optogenetic manipulations to dissociate the roles of these areas. The conceptual conclusions of the paper are exciting, with a potentially significant advance in understanding for the field. The approaches are innovative and applied in a systematic and thorough manner. Overall, the paper is well written and presented. My major reservation is that the main findings of the papers in Figures 5-6 could benefit from additional support. From the current presentation, it is not clear how large the difference in effects are between ALM and striatum inhibition. Further work is needed to clarify and quantify the differences to support the most important and interesting conclusions of this paper.

Specific comments

1. The text (abstract and main text) make major claims about different roles for striatum and ALM. In large part, these claims are based on the inhibition results from Figures 5 and 6. A key part of the argument is that after the ALM inhibition, the population activity returns to the ramping right where it left off at the start of the inhibition. In contrast, the striatum inhibition is interpreted as winding down the ramp of activity and restarting the ramp from a lower position after the inhibition ends.

From the data in Figure 5-6, it is not clear that the post-inhibition starting point for population activity is greatly different for ALM and striatum inhibition. I do not see the difference as being very apparent when comparing panels 5f,g,i,j with panels 6i,j,l,m, specifically when looking at the immediate post-inhibition time points. Similarly, in EDF 9, the pre vs post differences look similar between panels a,e vs i,m. Also, in the time to lick decoding in EDF 9, the confidence intervals for return to pre-stim level are overlapping (and nearly the same) for the ALM vs. striatum inhibition. There are other pieces of evidence that help the interpretation put forward by the authors, but overall more evidence is needed here. I am not sure what is best, but from looking at the panels mentioned, it does not jump out to me that there is a major difference between these manipulations. It would be helpful to see direct comparisons and quantifications with statistics showing the differences.

These direct comparisons would provide stronger evidence about the effect size and about the statistical significance.

The second part that I have reservations about is the winding down of activity. The striatum inhibition appears to be weaker than the ALM inhibition in that fewer neurons are inhibited. From a limited set of recordings, the authors estimate that 70% or so of D1 neurons are inhibited. But, D2 MSNs are not inhibited and 30% of D1 MSNs remain. It seems possible that the wind down could be due to incomplete silencing compared to the ALM inhibition. While the paper provides arguments against this concern, it would be helpful if these arguments could be bolstered with more data. For example, does the weaker stimulation in ALM lead to similar fractions of cells inhibited as in striatum? Is there a way to more strongly inhibit striatum? Since so much of the argument about the different roles rests on this winding down, it seems important to completely rule out the possibility that this is due to incomplete silencing.

2. The terminology and meaning around ramping should be clarified. My understanding from the data as it is presented is that there is a ramping of activity at the level of neural populations. It is less clear to me that ramping happens in single neurons. In a lot of literature regarding, movement planning, timing, and decision making, ramping is shown in single units, especially in recordings from macaques. In Figure 3 panels a,b,f,g, it does not look like each cell is ramping. It looks like cells tile the delay before licking. While this is not a central point of the paper, it is a topic that has been debated in the field and might matter to some readers. It would be helpful to provide more analyses to demonstrate if each cell is indeed ramping or if the tiling that is present in the examples is a more accurate reflection of the substrate underlying the ramp.

(Remarks on code availability)

Version 1:

Reviewer comments:

Referee #1

(Remarks to the Author)

The authors have provided a comprehensive response to my review, including new, convincing experiments. I support publication of the current version.

Nevertheless, I don't understand why the authors decided to bury these new results/analyses in the supplemental materials. In particular, the differential behavioral effects of perturbing ALM and striatum at different times are arguably the most convincing and direct evidence that the two areas are performing distinct functions. Yet, the result is buried in Extended Data Figure 19. More generally, my main complaint about this wonderful paper is that much of the important information is mixed within an enormous amount of supplementary materials. Indeed, most of the figures cited in the discussion are extended data figures. Most readers will not go through the Supplementals in detail and end up viewing the results with unjustified skepticism. I would encourage the authors to revisit their decisions about what to include in the primary figures.

(Remarks on code availability)

Referee #2

(Remarks to the Author)

The authors have addressed my major concerns. I realized that my previous review had its intro paragraph cut off so, for the record, I'll add a couple sentences from that: "The experimental work is very beautiful and an exemplar for the type of work needed to dissect challenging, fundamental computations that involve multi-brain region interactions. Furthermore, the model-based framework for interpreting the data is very elegant and, together with the experiments, provides a beautiful exposition on how to tease apart the source of a neural integrator from its readout and its inputs." [The criticisms that followed this were in the previous Major comments and were generally addressed in this excellent revision, which has made the connections between the conclusions and the data much stronger and more rigorous.]. A few minor comments and suggestions are below, but overall I applaud the authors on a tour de force paper that truly sets the standard for the field.

Minor comments:

1) The overall response to review points out that quantities like state dependence and hazard rate offer more robust means of distinguishing pause/slowdown from rewind. Yet, the experiments doing inactivations at two delays and considering the hazard rate only appear in Supplementary Figure 19 (which I thought was a great figure). Should these therefore be in the main? (There were a couple other things in this spirit that may have been unaddressed or buried, such as reviewer 1's comment that, if one wants to group trials, then a more logical way is to group by level of activity in the ramp mode at the time

of laser onset, rather than sorting by lick time to make comparisons; instead, it seems that single trial analysis was put in the supplement but grouping was kept by lick time in the main text).

I will leave to the authors and editor the ultimate decision on such matters, but I found the reply to review to be really nice in addressing the collective reviewer requests for clearer, more robust connections between the conceptual picture/storyline and the data/results, so it seems like a shame to have these more conclusive (I think?) analyses heavily buried in Extended Figures that many readers won't see. Not sure what the space constraints will be, but I could imagine Fig. 4 getting combined with another figure, and then one could consider which figures/figure panels are the best characterizations for a given point being made, i.e. which main text figures should stay or be cut if freeing up space is required. One possible example: do the peaks of the correlations in Figs. 5 and 6 provide the most rigorous and clear assessment, relative to many of the analyses in the supplements, given they are tracking only the peak of a very broad yellow swath of correlation that has very broad error bars?.

2) As multiple reviewers pointed out, a naïve read of the manuscript would suggest that there could be positive feedback between ALM and STR since each is driving the other. Page 12 (points 1 and 2 paragraphs) and figures 5 and 6 do a nice job of explaining the on vs. off manifold aspects of ALM input to striatum and their consequences on activity, and of the on-manifold optogenetic perturbation of striatal D1 activity. However, I think it's very much buried that the reason ALM can follow striatal input without this causing a positive feedback loop between striatum and ALM, is because striatum is driving the component of ALM firing that projects onto the off-manifold aspect of STR activity, thus avoiding the feedback loop that would occur if STR drove the component of ALM firing that projects back to the 'on-manifold/integrating mode' of the striatum. One really needs to understand Extended Data Fig. 1f (which hardly is explained) to see this. It seems like this STR-to-ALM aspect of the picture needs to be clarified, not only in the supplemental/extended data material but also in the main text.

Also, related to this, I think the critical thing is not (as suggested by the reply to review and Extended data fig. 5m) that the ALM ramp mode and trial-history mode are orthogonal. Indeed, one could imagine that these are orthogonal and each is at 45 degree angles to the (left eigenvector of) the striatal integrating mode, so that both would get integrated. I think the implicit assumption is that the trial-history mode is perfectly aligned with the (left eigenvector of the) striatal integrating mode, so that then the ALM ramp and trial-history modes being orthogonal make the ALM ramp mode not get integrated (and therefore, if the striatum projects to the ALM ramp mode, then there is no positive feedback loop). However, I think the key condition is that the striatum projects to a mode of ALM dynamics that is orthogonal to the (left eigenvector of) the striatal integrating mode.

4) In the PCA analysis of Extended Data Fig. 9, PC2 is explained as looking like the trial history mode. However, unless I'm missing something, it seems very much like the middle mode and not nearly so much like the trial history mode -- in fact, as shown in panel d, while it has a "significant" deviation from orthogonality, the peak of the shown distribution is still very close to $\pi/2$, i.e. close to orthogonal. Clarification is needed here on what to take away from this analysis, and what not to take away from it.

5) In Extended Data Fig. 8, why is decoded time to lick negative on the y-axis? If negative means "longer time preceding lick", then why is the loud cue above the faint cue, which seems like that would mean the loud cue has more time from cue to lick in panel j (which would contradict panel k, which makes intuitive sense in showing there is typically less time from cue to lick for the loud cue). Are the loud and faint lines in this panel switched, or can you clarify the meaning of the y-axis (or maybe I just missed something obvious).

6) In the Extended Data Fig. 6a3-6e row, it would be worth making clear what a value of 1 means for the square sum of spiking activity explained axis.

(Remarks on code availability)

Referee #3

(Remarks to the Author)

The authors have added new data and modeling that support the overall conclusions and interpretations. My concerns have been adequately addressed. I continue to feel that the results are somewhat mixed (or difficult to interpret) between rewinding and pausing the timer. The text makes the distinction between areas to sound black and white, including in the abstract. As I mentioned in my previous review and as other reviewers mentioned as well, some of the results appear more intermediate. I feel it would be appropriate for the authors to scale back any language that makes the result seem black and white and/or add in some caveats or more nuanced interpretation throughout. This will both be more accurate to the data, in my opinion, and likely help the paper to have longevity in case future studies reveal that there are somewhat mixed functions of these areas.

(Remarks on code availability)

Dear Reviewers,

We thank the reviewers for their supportive and constructive comments. In response, we conducted substantial new experiments (n = 43 mice). To improve clarity and readability, we first summarize the key new results that address similar comments raised by multiple reviewers in **Sections I and II**, before responding to each reviewer individually in **Section III**.

We made numerous changes in the text and figures to address comments made by the reviewers and to streamline the manuscript. In the revised manuscript, we have highlighted edits for clarity. In this letter, we follow the format below:

Blue italic text: reviewer's comments

Black text: author responses

Green italic text: copied from the revised manuscript

All changes from the original manuscript are highlighted in yellow.

Table of contents

I. Modeling and Testing Pause vs. Rewind Effects of Optogenetic Manipulations

1-1: Modeling the Effects of Optogenetic Manipulations on an Internal Timer	-P. 2
1-2: Inhibiting ALM and D1-SPN at Different Time Points to Test Pause vs. Rewind	-P. 4
1-3: Explanation of D1-SPN Inhibition-dependent Increase in No-lick Rate	-P. 6

II. Tests for the integrator hypothesis

2-1: Tight Trial-by-trial Coupling of Dynamics Between ALM and Striatum	-P. 11
2-2: Manipulating Cue Intensity to Test the Integrator Hypothesis	-P. 13
2-3: Evaluation of Alternative Mechanisms Underlying Ramping Activity	-P. 16

III. Point-by-Point Response to Reviewer Comments

Referee #1	-P. 18
Referee #2	-P. 29
Referee #3	-P. 45

References	-P. 50
-------------------	---------------

I. Modeling and Testing Pause vs. Rewind Effects of Optogenetic Manipulations

Our manuscript reported that although the premotor cortex and striatum exhibit similar neural dynamics, their inhibition leads to distinct effects on timing behavior and neural dynamics, which we referred to as "pause" and "rewind." To address reviewer concerns and rigorously distinguish these two effects in a hypothesis-driven manner, we conducted new computational modeling followed by targeted experiments for validation.

1-1: Modeling the Effects of Optogenetic Manipulations on an Internal Timer

We modeled the 'timer' as a generic accumulator, whose dynamics can be perturbed in ways that mimic optogenetic manipulations - either by slowing or reversing accumulation (Revision Fig. 1a). The 'timer' infers the passage of time by integrating a constant input or periodic event, as in an hourglass. This framework makes it explicit how distinct perturbation types (i.e. pause vs. rewind) reshape internal time tracking.

Revision Fig. 1. Modeling and testing timer perturbations at different onset times (these panels are now in Extended Data Fig. 19)

a. Schema of the model (Methods).

b-c. First row, schema of transient manipulation of the timer. Second row, simulated cumulative lick time distributions. Blue, trials with manipulation at 0.6 s after the cue. Magenta, trials with manipulation at 0.9 s after the cue. Third row, hazard rate of lick time distribution based on the lick time distribution above. Fourth row, simulated internal representation of time in trials with manipulation at 0.6 s after the cue (Methods).

In this analogy, time is represented by the rising sand level in the bottom chamber, reflecting the accumulation (or temporal integration) of a constant sand flow, and when the sand reaches a

threshold level, it triggers a lick. We varied inflow rates across trials so that the lick time distribution under control conditions followed an inverse Gaussian distribution, consistent with the data. In addition to lick time distribution, we analyzed lick hazard rate: the moment-by-moment likelihood of a lick occurring, given that it has not yet occurred. This measure reflects the instantaneous drive to lick, allowing us to analyze the temporal dynamics of lick probability.

Then, we simulated how two types of transient perturbations, **pause/slowdown** and **rewind**, influence behavior across trials. The models are circuit-agnostic, allowing for the interpretation of behavior without assuming a specific network implementation.

Pause/slowdown (Revision Fig. 1b) corresponds to reducing the sand inflow by pinching the bottleneck of the hourglass. The extent of slowdown is scaled by a speed coefficient representing the strength of the optogenetic manipulation, where 1 indicates normal flow (no manipulation) and 0 represents a complete pause. Intermediate values indicate proportionally slower accumulation. Thus, here pause and slowdown lie on a continuum of varying manipulation strengths. During manipulations belonging to this category, the ramping increase in sand level stops (or slows down) and then resumes from a level close to (or above) the pre-perturbation level, depending on the speed coefficient. This causes a parallel shift in the lick time distribution and lick hazard rate, equal to (if complete pause) or shorter than the manipulation duration. Importantly, the extent of this shift does not depend on when the manipulation starts, i.e., state-independent. In addition, the hazard rate may decrease during the manipulation, but recovers to the pre-perturbation level at the end of the manipulation, as the timer's internal state recovers close to its pre-perturbation state.

Rewind (Revision Fig. 1c), in contrast, is modeled as flipping the hourglass (the rate of sand flow after the flip may differ from the original inflow rate, depending on the strength of the optogenetic manipulation). In this condition, the sand level in the original bottom chamber gradually decreases during the manipulation and then resumes from the reduced level once the manipulation ends (i.e., when the hourglass is flipped back). This causes a larger shift in both the lick time distribution and hazard rate than the manipulation duration. Importantly, different from pause/slowdown, these shifts are state-dependent: the later the manipulation starts, the greater the shift. This state-dependency arises from a floor effect: once the timer rewinds to zero, it cannot rewind further (thus resulting in a functional 'reset' of the accumulator). This aligns with our recording showing that ramping activity does not fall below baseline (Fig. 6 and new Extended Data Fig. 16). In addition, unlike pause/slowdown, the hazard rate is 0 at the end of manipulation, as the timer has rewound and must begin tracking the passage of time before the lick again.

These distinct behavioral signatures — shift extent, hazard rate recovery, and state dependence — distinguish pause/slowdown from rewind. While the brain's internal time representation (e.g., ramping activity) may reflect the modeled timer's dynamics, not all behavioral readouts are equally powerful to separate the effect of perturbations. Specifically, shift magnitude (used in our original manuscript) is a quantitative measure that depends on the strength of the perturbation.

In contrast, state dependence and hazard rate recovery represent qualitative signatures, offering a more robust means of distinguishing pause/slowdown from rewind, as rightly pointed out by Reviewer 1.

1-2: Inhibiting ALM and D1-SPN at Different Time Points to Test Pause vs. Rewind Effects

Guided by the model's predictions described in **section 1-1**, we conducted new experiments inhibiting ALM or D1-SPN at two time points after the cue (0.6 and 0.9 s after cue onset).

a) ALM silencing (n = 8 mice)

Transient ALM silencing shifted the lick time distribution and hazard rate (Revision Fig. 2d and f) close to the duration of silencing (Revision Fig. 2g). Importantly, the lick time distribution and shift in hazard rate after silencing were indistinguishable between the two onset times (Revision Fig. 2e and g), and the hazard rate recovered to the pre-silencing level by the end of the silencing period (Revision Fig. 2h). Silicon probe recordings during these behavioral experiments confirmed effective ALM silencing, and ALM activity quickly returned to pre-perturbation level at the end of the manipulation, regardless of onset time (Revision Fig. 2n and o). This pattern matches the timer's state in the pause/slowdown model, except for the collapse during silencing (this pattern is expected if ALM provides input to a downstream integrator, i.e., the striatum, rather than performing the temporal integration itself). Weak ALM silencing produced a smaller shift in lick time: 0.23 s (Extended Data Fig. 18ab), consistent with a milder slowdown. Together, the effect of ALM silencing is consistent with a pause/slowdown, with the extent of the slowdown depending on laser power (approaching a pause at the highest power we used). The pause/slowdown supports the idea that ALM provides input to the integrator, analogous to sand inflow in the hourglass.

b) D1-SPN inhibition (n = 7 mice)

Transient D1-SPN inhibition shifted lick timing and hazard rate (Revision Fig. 2i and k) longer than the duration of the manipulation (Revision Fig. 2l). In contrast to ALM silencing, in this case, the shift was significantly larger when inhibition began later (0.9 s after the cue; Revision Fig. 2l), and the lick time distributions were distinct between the two onset times (Revision Fig. 2j). During inhibition, the hazard rate dropped and remained at zero even at the end of the inhibition (Revision Fig. 2m). Additionally, ALM ramping activity gradually decayed during D1-SPN inhibition and did not rapidly recover afterward, mirroring the internal time representation in the rewind model (Revision Fig. 2pq). Thus, the effect of D1-SPN inhibition is consistent with a rewind of the timer. This supports the idea that D1-SPN plays a key role in temporal integration, analogous to the bottom chamber in the hourglass.

To summarize, all behavioral measures and recording data consistently support the interpretation that ALM and striatal inhibition result in pause (slowdown when laser power is weak) and rewind of the timer, respectively.

Behavioral data

Revision Fig. 2. Modelling and testing different perturbations of the timer (these panels are now in Extended Data Fig. 19)

d. Cumulative lick time distribution in trials with ALM silencing at different onsets. Two example sessions are shown.

e. Difference in lick time distribution (CDF) between trials with ALM silencing at different delay onsets (0.6 s vs 0.9 s). The maximum vertical difference for each session, the value used for Kolmogorov–Smirnov (KS) test is shown in the histogram (trials with 0.6 s onset minus trials with 0.9 s onset is shown). This value is expected to be zero or positive in pause/slowdown or rewind condition, respectively, according to b and c). P-value; bootstrap with null hypothesis that the mean maximum vertical difference is 0. At the individual session level, 3 out of 15 sessions had a significantly positive difference, and 1 had a significant negative difference (KS test; $p < 0.05$).

f. Hazard rate across sessions. $n = 16$ sessions, 8 mice. Line, mean. Shade, 95% confidence interval.

g. Temporal shift in the 50% point of the hazard rate caused by manipulations at different onsets. Value, mean and 95% confidence interval. P-value: bootstrap with null hypothesis that there is no difference between conditions. Line, each session. The central line in the box plot, median. Top and bottom edges, 75% and 25% points. Whiskers, the lowest/highest datum within the 1.5 interquartile range of the lower/upper quartile. P-value, hierarchical bootstrap.

h. Values of hazard rate of lick time before (Pre; manipulation onset), during (Middle; 0.3 s after manipulation onset), and after (Post; at the end of the manipulation) the manipulation (for trials in which manipulation started 0.9 s after the cue). P-value: bootstrap with null hypothesis that there is no

difference in hazard rate between time points. Same format as in g. In some sessions hazard rate decreased during the silencing, but after the silencing, the hazard rate recovered.

i-m. Same as d-h but for D1-SPN inhibition at different delay onsets. In k, 6 out of 7 sessions had a significantly positive difference in maximum vertical difference(KS test; $p < 0.05$). $n = 7$ sessions, 7 mice.

n. ALM dynamics during ALM silencing at different delay onsets. Trials were pooled by lick time. Lines, grand mean. Shades, SEM (hierarchical bootstrap). $n = 16$ sessions, 8 mice.

o. ALM population activity along the ramp mode before, during, and after the manipulation. ALM activity recovered to the pre-silencing level regardless of the silencing onsets. Trials with decoded time to lick before inhibition between -0.1 and 0.2 seconds relative to the session's median lick time were analyzed.

p-q. Same as n-o but for D1-SPN inhibition at different delay onsets. $n = 7$ sessions, 7 mice. In both manipulation onsets, ALM activity did not recover to the pre-silencing level.

1.3: Explanation of D1-SPN Inhibition-dependent Increase in No-lick Rate

Reviewers noted an increase in no-lick trials following D1-SPN inhibition, in addition to the significant shift in lick timing. Here, we test the hypothesis that no-lick trials result from a rewinding of the internal timer, rather than representing an independent phenomenon.

Hypothesis and predictions: At the cue onset that initiates the internal ‘timer’, both ALM and the striatum exhibit dramatic changes in activity patterns. If timing dynamics rewind back to baseline, it may be difficult to restart the timer, as there is no second cue to re-initiate it. This could lead to a stochastic failure to resume timing dynamics once activity returns to baseline. If so, no-lick trials following D1-SPN inhibition should be those that are more likely to reach the baseline during inhibition, i.e., trials that start with lower pre-perturbation activity. Therefore, it should be possible to predict no-lick outcomes from neural activity preceding the perturbation (**Prediction I**). In addition, longer D1-SPN inhibition, which brings activity closer to baseline, should increase the rate of no-lick trials, in addition to producing a greater shift in lick timing (**Prediction II**).

To test **Prediction I**, we decoded the time to lick on individual trials using a k-nearest neighbors (kNN) decoder and compared lick and no-lick trials prior to inhibition onset (Revision Fig. 3ab). Consistent with our hypothesis, we can predict no-lick trials based on the activity preceding the perturbation: the decoded time to lick was significantly later in no-lick trials, consistent with these trials starting with lower pre-perturbation activity.

Revision Fig. 3. Prolonged rewinding of the timer leads to “no-lick” trials (these panels are now in Extended Data Fig. 16)

a. Three example ALM recording sessions with bilateral D1-SPN inhibition (0.25mW) at 0.6s after the cue onset. The relationship between actual lick time (x-axis) and decoded lick time (kNN) at 0.6 s after

the cue (time before inhibition) is shown for both control (black) and inhibition trials (blue). Individual circle, individual trial. In no-lick trials, the decoded time to lick tended to be later than lick trials.

b. Summary across sessions. Decoded time to lick before the inhibition onset in lick vs. no lick trials. The central line in the box plot, median. Top and bottom edges, 75% and 25% points. Whiskers, the lowest/highest datum within the 1.5 interquartile range of the lower/upper quartile. P-value, hierarchical bootstrap. $n = 8$ session, 8 mice. No-lick trials can be predicted before the inhibition.

c. Cumulative distribution of lick time in trials with bilateral D1-SPN delay inhibition (0.15 mW; used this power to avoid rebound excitation with the longer inhibition) with different durations (0.3, 0.6, or 0.9 s starting at 0.6 s after the cue). Black, control trials. Blue, inhibition trials. Shades, 95% confidence interval (hierarchical bootstrap). $n = 4$ sessions, 4 mice.

d. Change in median lick time in c. Error bars, 95% confidence interval (hierarchical bootstrap). Dashed line, unity line. Even with this low laser power, the shift in lick time is larger than the shift caused by ALM silencing (panel j) across conditions ($p = 0.013, 0.021, 0.019$ for 0.3, 0.6, and 0.9 s duration, respectively; hierarchical bootstrap with a null hypothesis that the lick time shift by ALM silencing is higher than or equal to that by D1-SPN inhibition).

e. No-lick rate in c. Error bars, 95% confidence interval (hierarchical bootstrap). P-value, hierarchical bootstrap with a null hypothesis that the slope of the linear regression between no lick rate and inhibition duration is smaller than or equal to 0. No lick rate increases as a function of inhibition duration.

f. ALM population activity along the ramp mode (RM; $n = 96$ neurons, 3 mice), trials pooled by lick time. Lines, grand mean. Shades, SEM (hierarchical bootstrap). Left, control. Right, D1-SPN inhibition trials with different durations. Note that ramping activity continues to decrease during D1-SPN inhibition. No-lick trials with a longer inhibition (0.9 s) show a robust decrease and activity remains at the baseline level. This is consistent with a hypothesis that a decrease in ramping activity to baseline due to inhibition leads to the absence of a lick.

g. Quantification of the decrease in ALM RM activity along the RM following bilateral D1-SPN inhibition, calculated as the difference in RM activity between the onset and end of inhibition. ALM RM activity continues to decay more with longer inhibition. Calculated based on lick trials in which the decoded time to lick before inhibition was between -0.1 and 0.2 seconds relative to the session's median lick time. P-value, hierarchical bootstrap comparing 0.3 vs 0.9s.

h. Quantification of ALM activity along the RM at the onset of bilateral D1-SPN inhibition in no-lick trials. Baseline before cue onset is subtracted. Even trials with high ALM RM activity can result in no lick when D1-SPN inhibition is prolonged. P-value, hierarchical bootstrap comparing 0.3 vs 0.9s.

i. Quantification of ALM activity along the RM at the end of bilateral D1-SPN inhibition in lick (L) and no-lick (N) trials. Baseline before cue onset is subtracted. At the end of inhibition, activity decayed to the baseline level in no-lick trials. P-value, hierarchical bootstrap. *: $p < 0.05$, with the null hypothesis that the value is lower than 0.

j-l. Same as in c-e but for ALM delay silencing (1.5 mW) with different durations. $n = 4$ sessions, 4 mice. Silencing duration shifted lick time accordingly (consistent with a pause) but did not affect the no-lick rate, unlike D1-SPN inhibition.

To test **Prediction II**, we varied the duration of inhibition (we reduced the laser power to 0.15 mW to avoid rebound effects associated with prolonged strong inhibition¹; Revision Fig. 3c-f). Consistent with our prediction, the no-lick rate increased as a function of inhibition duration, and was accompanied by a larger shift in lick timing (Revision Fig. 3c-e). Simultaneous ALM recordings revealed that D1-SPN inhibition caused a gradual decay of ramping activity during manipulation (Revision Fig. 3f). This decay continued with longer inhibition, leading to no-lick trials even when ramping activity was high prior to inhibition onset (Revision Fig. 3f-h), as the

inhibition brought activity close to baseline (Revision Fig. 3i). This is consistent with our hypothesis that extended inhibition pushed activity closer to baseline, increasing the likelihood of a no-lick outcome.

As a control, we also varied the duration of ALM silencing (Revision Fig. 3j-l). While longer silencing led to larger shifts in lick timing (as expected for 'pause'), it did not increase the no-lick rate. Thus, not all manipulations that cause larger shifts in lick timing lead to an increase in the no-lick rate.

Altogether, the increase in no-lick trials is specific to D1-SPN inhibition and is likely due to its effect of driving the timing dynamics back to baseline. Thus, both extended shift in lick timing and increased proportion of no-lick can be explained within the same framework of timing dynamics decaying back to baseline. We also note that if returning to baseline always led to a no-lick outcome, there would be no state dependency. However, if the no-lick outcome is stochastic or requires a prolonged stay at baseline, it can give rise to both state dependency and no-lick as implemented in Revision Fig. 1c (Methods).

Summary of Section I

Together, these new results demonstrate that ALM and D1-SPN manipulations produce qualitatively different effects on the internal timer, as evidenced by both behavior and neural dynamics. In addition, our new results confirm that the distinct effects of ALM and D1-SPN manipulations on lick timing and cortical dynamics are robust across different onset times and manipulation durations.

The new model and behavioral results from the onset experiments are now included in Extended Data Fig. 19, and the experiments to test no-lick trials are summarized in Extended Data Fig. 16. Due to space constraints in the manuscript, we have included the text of **Section I** in the Supplementary Notes. In addition, we have added the following text to the main manuscript:

Page 15, Line 457

“ALM RM activity and decoded Tto lick gradually decayed during VLS D1-SPN inhibition but resumed ramping in near-parallel to control trials after inhibition ended, without a rapid recovery phase (Fig. 6m and Extended Data Figs. 12m, p, and t and 13k-m). Consistently, the time for Tto lick to recover to the pre-perturbation level was significantly longer than in ALM silencing (Extended Data Fig. 12u). This decay in timing dynamics explains both the extended shift in lick timing and the increase in no-lick trials (Extended Data Fig. 16; Supplementary notes).”

Page 16, Line 480

“In addition, a 'pause' in a timer should shift lick timing equally regardless of inhibition onset, whereas a 'rewind' should produce a larger shift with later onsets. Consistently, ALM silencing shifts lick timing indistinguishably regardless of onset, while inhibiting D1-SPNs at later times shifts lick timing further, highlighting a qualitative difference in the impact of these manipulations on lick timing (Extended Data Fig. 19; Supplementary notes).”

Page 16, Line 486

“Overall, VLS D1-SPN inhibition has a stronger impact on behavior than ALM silencing, regardless of manipulation onsets or durations (Extended Data Figs 16 and 19), despite its weaker effect on mean spike rates.”

II. Tests for the integrator hypothesis

In our original manuscript, we proposed that ramping dynamics representing time arise from the temporal integration of step-like inputs (trial-history mode; TM), inspired by observed ALM activity patterns and prior theoretical studies²⁻¹⁰. To further test this integrator hypothesis, we conducted additional experiments and analyses (**Sections 2-1 to 2-2**) and evaluated alternative scenarios (**Section 2-3**).

2-1: Tight Trial-by-trial Coupling of Dynamics Between ALM and Striatum

As pointed out by reviewers, if striatal ramping activity arises from the integration of ALM trial-history mode (TM) activity, ALM TM activity must be tightly correlated with striatal ramping on a trial-by-trial basis. Furthermore, if ALM ramping reflects striatal ramping activity, then ramping activity and decoded time should be tightly correlated between these areas. To test this, we analyzed sessions with simultaneous recordings from ALM and striatum. In these sessions:

1) The amplitude of ALM TM activity is correlated with the slope of the striatal ramping (quantified as the time when ramp activity reaches 75% of peak) on a trial-by-trial basis (Revision Fig. 4a, b, and d).

2) Ramping activity and decoded time are highly correlated between ALM and striatum on a trial-by-trial basis (Revision Fig. 4a, c, e, and i-k).

These findings are consistent with our hypothesis. The new analyses are now included in Extended Data Fig. 7, and we have added the following text to the main manuscript:

Page 9, Line 257

“Consistent with the integrator hypothesis, the amplitude of ALM activity along the trial-history mode and the slope of ramping activity in both ALM and striatum were highly correlated on a trial-by-trial basis across sessions (Extended Data Fig. 7a-e).”

Revision Fig. 4. Tight trial-by-trial coupling of dynamics between ALM and striatum (these panels are now in Extended Data Fig. 7)

a. Three example ALM-striatum simultaneous recording sessions (from top to bottom). Left, ALM activity projected to trial-history mode (TM). Middle, ALM activity projected to ramp mode. Right, striatum (STR) activity projected to ramp mode. Line, mean of trial deciles. For all modes, trials are sorted and grouped by ALM activity along the trial-history mode before the cue (blue, trials deciles with the highest activity; red, trial deciles with the lowest), allowing direct comparison of activity in the same trial deciles. Trials with high activity along TM (blue) exhibit steeper ramping activity in both ALM and striatum.

- b. Across-trial Pearson's correlation between the amplitude of ALM activity along the trial-history mode and the slope (time to reach 75% of peak activity after the cue) of striatum activity along the ramp mode for three sessions shown in a. Circles, individual trials.*
- c. Across-trial Pearson's correlation between the slope of ALM and STR activity along the ramp mode for three sessions shown in a. Circles, individual trials. Dotted line, the unity line.*
- d. Across-trial Pearson's correlation between the amplitude of ALM activity along the trial-history mode and the slope (time to reach 75% of peak activity level) of STR activity along the ramp mode for all simultaneously recorded sessions. Error bars, 95% confidence interval (hierarchical bootstrap). 16 out of 19 sessions show a significant negative correlation between ALM TM activity and STR RM activity. $n = 14$ mice.*
- e. Across-trial Pearson's correlation between the slope of ALM activity along the ramp mode and STR activity along the ramp mode for all sessions. Error bars, 95% confidence interval (hierarchical bootstrap). All 19 sessions show a significant positive correlation between ALM RM activity and STR RM activity.*
- f. Temporal difference between ALM and STR RM activity in reaching 75% of its maximum level. No significant difference was detected.*
- g. Schema depicting a k -nearest neighbor (kNN) method to decode the time to lick (Tto lick) using population neural activity at each time point (Methods).*
- h. The performance of the kNN decoder as a function of the number of simultaneously recorded neurons. Decoding accuracy was quantified by Pearson's correlation between actual lick time vs. lick time decoded at cue onset. The performance increased with more recorded neurons.*
- i. Decoded Tto lick was highly correlated between ALM and striatum at a single-trial level. Four example trials from an example session are shown. Traces end at the time of lick.*
- j. The relationship between decoded lick times estimated from ALM and striatal neurons in an example session. Dots, all time points (50 ms bin; from cue to lick) in the example session. Pearson's correlation across all time points (0.79) was significantly higher than that of the trial shuffle control (0.51 - 0.62, 95% confidence interval).*
- k. Pearson's correlation of decoded time between ALM and striatum was significantly higher than trial shuffle controls in all simultaneously recorded sessions (19 sessions). Thus, ALM and striatal timing dynamics are synchronized at a single-trial level.*

2-2: Manipulating Cue Intensity to Test the Integrator Hypothesis

The integrator hypothesis predicts that transient manipulations affecting trial-history mode (TM) activity will be integrated into persistent changes in ramp mode (RM) activity, thereby influencing lick timing. To test this, we leveraged the observation that TM activity exhibits a step-like increase at cue onset, while RM activity does not respond transiently. This may allow us to manipulate TM activity by modifying cue intensity as proposed by Reviewer 2 (Revision Fig. 5e).

We varied cue intensity (3 kHz auditory cue, ± 15 dB, 0.6 s) in 20% of randomly interleaved trials (cue intensity was constant, 74 dB, before test sessions). ALM recordings revealed that both faint and loud cues bidirectionally modulated the transient response of ALM neurons to the cue, including TM activity (Revision Fig. 5f-h). RM activity and kNN decoded time were also bidirectionally modulated, but importantly, the change developed gradually and persisted beyond cue presentation, consistent with the integrator hypothesis (Revision Fig. 5i-j). Consistent with the modulation of RM and decoded lick time, the faint and loud cues significantly shifted lick time later and earlier, respectively (Revision Fig. 5k).

Revision Fig. 5. Contextual tonic activity in ALM predicts lick time (these panels are an excerpt from Extended Data Fig. 8)

e. Left, schema of integrator hypothesis. Right, schema showing the prediction of the cue intensity experiment: changes in trial-history mode in response to different cue intensities will be integrated into lasting changes in ramping activity, affecting lick timing.

f. Two example ALM neurons showing bidirectional modulation in cue response (left) or ramping activity (right) in response to different cue intensities. Magenta, loud cue. Green, faint cue.

g. ALM population activity in response to different cue intensities. Top, schema of cue mode. Middle, ALM population activity projected along cue mode (faint vs loud cue trials). Bottom, the difference in activity along the cue mode between loud and faint cue trials. Line, mean. Shade, 95% confidence interval (hierarchical bootstrap). $n = 788$ neurons, 15 sessions, 5 mice.

h-j. The same as in **g**, but for trial-history mode, ramp mode, and decoded time.

k. Lick times following different cue intensities. Top, cumulative distribution of lick time. Shade, 95% confidence interval (hierarchical bootstrap). Bottom, quantification of the change in lick time. Error bar,

95% confidence interval (hierarchical bootstrap). *P*-value, hierarchical bootstrap with a null hypothesis that there is no change from the control condition. *n* = 22 sessions, 8 mice.

I-p. Same as *g-k* but for the constant delay condition. *n* = 977 neurons, *n* = 15 sessions, 5 mice. Despite bidirectional modulation of cue and trial-history modes, there was no change in ramp mode and decoded time. Note that the amplitude of the trial-history mode is small under the constant delay condition (*c*).

The observed effects were unlikely due to a sound-intensity-dependent difference in reaction time. Under the constant delay condition, where TM activity was weak, cue intensity did not shift lick timing, nor modulate RM or decoded time (Revision Fig. 5l-p). Together, the brief and bidirectional sensory perturbation produced lasting changes in timing dynamics and behavior, but only when robust TM activity is present, consistent with the integrator hypothesis.

In addition, in the original manuscript, we have reported that inhibiting ALM at different intensities produced lasting effects on the subsequent ramp activity and lick timing, with the magnitude proportional to inhibition strength (Extended Data Fig. 18). These results are consistent with the view that ALM activity is integrated to generate ramping activity.

We would like to point out that these results are non-trivial: in a memory-guided licking task (which requires memory but it is not a timing task, as action timing is instructed by a cue), ALM exhibited ramping dynamics associated with motor planning, mediated by discrete attractors¹¹. When ALM was bilaterally silenced in that task, ALM dynamics fully and rapidly recovered to the original ramping trajectory regardless of inhibition strength, i.e., without temporally integrating the perturbation into the ramp^{11,12} (Revision Fig. 6). In contrast, in our present timing task, ALM ramping is shifted in time by the perturbation relative to the original trajectory. This suggests that the integration of perturbations into ramping activity is specific to the timing task, where temporal integration can play a functional role, unlike in the memory-guided licking task.

Revision Fig. 6. ALM activity in a memory-guided licking task does not show a temporal shift in ramping activity after ALM silencing. Adapted from Inagaki et al., 2019. ALM was inhibited in *Vgat-ChR2-EYFP* mice trained on the memory-guided licking task. Regardless of inhibition strengths, ALM activity returned to the original trajectory, unlike in the timing task, which shows a temporal shift in ramping activity (Fig. 5f-h).

These new results are now included in Extended Data Fig. 8, and we have added the following text to the main manuscript:

Page 9, Line 259

“The hypothesis further predicts that modulation of the step-like increase in trial-history mode at cue onset would be integrated into persistent changes in ramping dynamics, thereby influencing lick timing. Indeed, modulation of the cue sound intensity in randomly interleaved trials altered the step amplitude and produced lasting changes in ramping dynamics and lick timing: a fainter cue led to a shallower ramp and delayed licking, while a stronger cue had the opposite effect (Extended Data Fig. 8e–k; Methods), consistent with the integrator model.”

2-3: Evaluation of Alternative Mechanisms Underlying Ramping Activity

So far, we have presented additional experimental evidence consistent with the integrator hypothesis. We now consider the question of whether there are alternative circuit-level mechanisms that could also explain our data.

In any recurrent network, activity evolves based on both initial conditions and external inputs. Depending on the network regime, which is shaped by both the strength and the structure of recurrent and external connections, one factor may dominate. In *input-driven* regimes, sustained external inputs primarily determine how activity evolves, as in integrator models. In contrast, in *initial-condition-driven* regimes, a brief input at the cue onset may set the initial state, but the subsequent dynamics unfold autonomously. Examples of this regime include ramping activity generated by slow drift toward a point attractor, or by runaway excitation (Revision Fig. 7a). In these relatively simple autonomous ramping models, ramping dynamics do not scale with time because trajectories are fixed once initiated. Ramping slopes remain the same across conditions with different action timing, except for the initial response to external input (Revision Fig. 7a), and therefore cannot account for the temporal scaling in dynamics observed during timing tasks (Fig. 3d, e, i, and j).

Revision Fig. 7. Schema of ramping activity in initial-condition-driven regime

a. Ramping activity generated by discrete attractors. At cue onset, input drives the activity away from the baseline attractor, and the dynamics unfold toward the other attractor via a positive feedback loop with nonlinearity. The resulting ramping activity is identical across conditions, except for the initial externally driven increase, i.e., there is no temporal scaling.

b. To implement temporal scaling, multiple such attractors can be arranged in parallel in state space (black arrows), with the initial condition determining which trajectory unfolds (red). This can implement

temporal scaling, yet, requires an additional mechanism to explain recovery to pre-perturbation level following silencing (green).

To implement temporal scaling, initial-condition-driven models need to be extended by incorporating parallel trajectories in a higher-dimensional state space, where different initial conditions lead to trajectories that unfold at different speeds^{13,14} (Revision Fig. 7b). However, to our knowledge, these extended models cannot intrinsically recover from transient silencing without an additional memory mechanism that stores and restores the internal state to its pre-perturbation level (as observed during ALM silencing). Such a memory mechanism would need to store a continuous state, which may be implemented by a mechanism analogous to an integrator. Thus, while these models can account for temporal scaling, they cannot explain recovery in timing dynamics after transient ALM silencing without additional complexity.

In contrast, input-driven integrator models naturally account for both temporal scaling and recovery. Varying the input strength modulates the ramping speed, thereby implementing temporal scaling. On-manifold perturbations (aligned with the integration axis) that cancel out the input can pause the ramp, while off-manifold perturbations (orthogonal to the integrating axis) allow rapid recovery to the pre-perturbation state without requiring additional mechanisms. Thus, while it remains possible that initial conditions also play a role in shaping neural dynamics, integrator-based architectures provide a more parsimonious explanation for both robustness to perturbation and the temporal scaling observed in our experiments.

To reflect this reasoning, we have added the following paragraph to the Discussion:

Page 17, Line 520

“In a dynamical system, the evolution of activity states is shaped by both the initial conditions and external inputs of a network^{9,31,50,83}. Models solely based on initial-condition-dependent parallel trajectories can account for temporal scaling^{9,10,84,85}, but require additional mechanisms to restore timing dynamics to the pre-perturbation state after ALM silencing. In contrast, the integrator model scales naturally with input strength, and silencing the input leads to a pause followed by recovery without an extra mechanism. Because it accounts for both scaling and recovery through a single mechanism, the integrator model offers a more parsimonious explanation, though initial conditions may also contribute to shaping the dynamics.”

In addition, as we cannot fully rule out an alternate scenario, we have revised the Abstract as follows:

Page 1, Line 36

“These findings are consistent with a model in which the striatum is part of a network that temporally integrates input from the frontal cortex and generates ramping activity that regulates motor timing.”

III. Point-by-Point Response to Reviewer Comments

Referee #1 (Remarks to the Author):

The paper explores the mechanisms that underlie ramping activity in the mouse frontal cortex (ALM) and striatum during a flexible timing task. The authors tested how perturbations of ALM and striatum affect behavior and ramping dynamics. While both areas display qualitatively similar signals, perturbations of each had different effects on behavior and dynamics. Based on these effects, the authors conclude that inhibiting ALM “pauses” the timer whereas inhibiting striatum “rewinds” the timer. They also use network models to deduce that the striatum integrates activity in ALM to generate ramping activity and that ramping activity in ALM is then inherited from the striatum (through a multi-synaptic pathway).

The main result—that perturbations of ALM and striatum cause different effects despite similar activity—is compelling and well-supported. An increasingly popular viewpoint in neuroscience is that computations are distributed across brain regions (“everything everywhere”). The results are an important counterpoint and will thus be of wide interest.

The paper has a few important shortcomings. First, the evidence for the authors’ interpretation that inhibiting ALM and striatum pauses and rewinds the animal’s timer, respectively needs stronger support to be convincing. Second, the conclusion that the striatum integrates ALM activity is not directly verified. Third, the hypothesis about integration/inheritance is not well supported. I elaborate on these issues below:

We thank the reviewer for their thorough comments, support of our findings, and important, insightful, and constructive suggestions that have significantly strengthened the manuscript.

1. The hypothesis that inhibiting ALM pauses the timer needs better evidence. The authors use two approaches to test this: (a) a comparison of activity before the perturbation to that after, and (b) an analysis of the effect of the perturbation on behavior.

A) For (a), the logic is that if the perturbation induces a “pause,” then the post-perturbation activity should return to its pre-perturbation level. Based on my reading, the authors do not directly test this prediction. Instead, they group trials based on the animals’ ultimate lick time and perform analyses on the average activity within these groups. Interpreting these analyses is difficult because the lick time is impacted by the perturbation. It is also not possible to compare dynamics in laser vs no-laser trials with this grouping, because the conditionalization is different for each trial type. Ideally, the authors would be able to leverage their large-scale recordings to do single-trial analysis of the dynamics. But, if grouping trials is necessary, then a more logical way is to group trials by the level of activity in the ramp mode at the time of laser onset (on laser trials) or the equivalent time on no laser trials. This would allow for a direct test of whether the post-perturbation activity returns to its prior level, and for a direct comparison between activity in laser and no-laser trials.

It also seems to me that analyzing the variance of activity along the ramp mode would be informative. Without a perturbation, the variance should increase as a function of time in the delay. What is the variance across trials before the laser vs after? Is it also consistent with a pause in the timer?

While grouping based on lick time is useful for visualizing activity patterns leading up to the lick, we agree that this approach has limitations, particularly when quantifying the effects of manipulations. To address this, we performed a single-trial analysis of ramping dynamics in each session (Revision Fig. 8).

Revision Fig. 8. Perturbation effects at single-trial level (these panels are now Extended Data Fig. 13)

a. Two example ALM recording sessions during ALM silencing, projected onto the ramp mode. Lines, individual trials color-coded by activity level before the silencing onset (blue, high; red, low). Although activity collapses during ALM silencing, the order of trials remains similar before and after silencing (quantified in b).

b. Across-trial Pearson's correlation of ramp mode activity between two time points in all recorded sessions. Left, correlation between before (0.6 s after the cue) and during (0.9 s after the cue) the perturbation. Right, correlation between before and after the perturbation (1.25 s after the cue). ALM RM activity is collapsed during the silencing, but RM activity after perturbation was significantly correlated with RM activity before the perturbation. $n = 28$ sessions, 18 mice.

c. The mean difference in ALM RM activity between before and after the perturbation across all silencing trials. The difference was not statistically different from 0 (signrank test), implying that ALM RM activity recovered to pre-perturbation levels after ALM silencing.

d. Across-trial Pearson's correlation of the derivative (increase) in ramping activity before and after perturbation (We calculated how much the ramp activity increased during the first 0.6 s (dRM/0.6), and compared it to the ramp increase from the end of silencing to where RM activity reaches 2 in each trial). The significant correlation indicates that the ramping speed is also recovered and remains correlated with the pre-perturbation condition after ALM silencing.

e. *Across-trial standard deviation of activity along the ramp mode before and after the perturbation. The central line in the box plot, median across sessions. Top and bottom edges, 75% and 25% points.*

Whiskers, the lowest/highest datum within the 1.5 interquartile range of the lower/upper quartile. P-value, hierarchical bootstrap. Across-trial standard deviation increased in control trials from 0.6 s to 1.2 s after the cue, but did not increase in ALM silencing trials, consistent with a “pause” effect of the timer, which suppresses not only the increase in mean (c) but also the variance of ramping activity.

f-j. *Same as in a-e but for striatum recordings during ALM silencing. Striatum activity also recovered to pre-stim level after ALM silencing. Unlike recording in ALM, striatum activity stayed correlated with pre-silencing activity even during ALM silencing. n = 23 session, 15 mice.*

k-o. *Same as in a-e but for ALM recording during bilateral STR inhibition. ALM activity decayed during inhibition but stayed correlated with the pre-perturbation level during, and after perturbation. ALM activity did not recover to pre-stim level after STR inhibition. n = 8 session, 8 mice.*

Notably, these results show that, even at the single-trial level: (1) ALM ramping activity collapses during ALM silencing but recovers afterward (Revision Fig. 8a-c), and even the slope of ramping activity is correlated between the periods before and after ALM silencing (Revision Fig. 8d); (2) striatal ramping activity retains information (remaining correlated with pre-silencing activity) during ALM silencing (Revision Fig. 8f, and g). These findings support the conclusions of the original manuscript.

We note that in the original manuscript, we performed a series of single-session analyses using a k-nearest neighbor decoder (Extended Data Fig. 12). Specifically, trials were grouped based on decoded lick time before perturbations (instead of lick time), and we quantified how neural activity changed during and after perturbations compared to control trials, similar to what suggested by the reviewer. These results support the recovery of activity to the pre-silencing level following ALM silencing, consistent with the single-trial analyses.

Lastly, as the reviewer suggested, we also calculated the across-trial standard deviation (std) of activity along the ramp mode for individual sessions (Revision Fig. 8e). As predicted by the reviewer, the std increased over time (from 0.6s to 1.2s after the cue) in control trials. In contrast, the std of ALM and striatum ramping activity remained at a similar level following ALM silencing (Revision Fig. 8e and j), consistent with the idea that ALM silencing pauses the timer not only in terms of mean activity (Revision Fig. 8c and h) but also in terms of variability.

To discuss these points, we have modified the main text as follows:

Page 11, Line 301

“Consistently, at the end of ALM silencing, RM activity rapidly recovered (Fig. 5h and Extended Data Fig. 12ab, and t), yet it was significantly lower than in the unperturbed condition (Extended Data Fig. 12a), indicating that dynamics did not return to their original trajectory (unlike in Fig. 1c1). Instead, RM activity recovered close to the pre-silencing level, as assessed by both the mean and standard deviation (Extended Data Fig. 13ce). Even at the single-trial level, RM activity collapses during ALM silencing, but recovers afterward, as indicated by the correlation between activity before and after silencing (Extended Data Figs. 13a-c). Following silencing, RM

activity resumed ramping at a rate similar to that observed before silencing (Extended Data Fig. 13d). These findings suggest that following silencing, ALM activity rapidly reverted to a state closely resembling its pre-perturbation condition at the individual trial level (akin to Fig. 1c5)."

B) The authors could better leverage the perturbation's effect on behavior to provide evidence for their claims. A "pause" in the timer makes a few, straightforward predictions about what should occur in the behavior. First, the animals should take longer to lick (as observed) without the increase in variability predicted by the scalar property of timing. Second, the change in behavior should not depend on the state of the timer. The clearest way to test this is to do the same perturbation at different times in the delay. Alternatively, the authors can group lick times based on the level of activity in the ramp mode (as described above). The change in behavior after perturbation should be the same for each group. Confirmation of these straightforward predictions would make the results convincing.

We thank the reviewer for these important suggestions. Please see **Section I** in this letter for detailed modeling and analyses addressing this point. In brief, as the reviewer speculated, our model predicts that the pause effect does not depend on the state of the timer, in contrast to the rewind effect. Consistently, our new experimental results show that the behavioral effects of ALM silencing at different onset times were state-invariant and support the pause interpretation.

Additionally, we quantified the coefficient of variation (CV) of lick times (restricted to licks occurring after the silencing onset time in both control and ALM-silencing trials) and found that the CV was significantly lower in ALM-silencing trials compared to controls regardless of the silencing onset time (Revision Fig. 9). These results are consistent with the idea that ALM silencing shifts lick timing without the corresponding increase in variability expected under the scalar timing property. Nonetheless, interpreting CV requires caution, as it reflects only the mean and variance, and the analysis is based on a truncated lick time distribution restricted to licks occurring after the silencing onset time. Manipulations may alter the lick time distribution in unintended ways, for example, by triggering rebound licking. To address this potential confound, we also analyzed the hazard rate of lick timing, which quantifies the dynamics of the moment-by-moment likelihood of licking (Revision Fig. 2f). We observed a near-parallel shift in the hazard rate following ALM silencing, with no evidence of a transient increase in licking after the silencing period. This rules out rebound licking and supports the interpretation that the entire lick time distribution is shifted later in time, consistent with a pause. We now report the hazard rate in the revised manuscript. See **Section I** for the corresponding edits in the manuscript.

Revision Fig. 9. ALM silencing decreases the coefficient of variation (16 sessions)

Coefficient of variation (CV) of lick timing in control trials and ALM silencing trials (only trials with a lick after the stim onset time were considered). $p = 0.034$ and 0.002 , respectively, for trials with silencing starting at 0.6 s, 0.9 s from the cue. Hierarchical bootstrap with a null hypothesis that there is no difference in CV between ALM silencing and control trials.

2) The hypothesis that inhibition of the striatum “rewinds” the timer may not be consistent with the data.

A) First, one of the most salient effects of striatum inhibition is to substantially increase the number of trials in which the animal doesn't lick, which does not seem consistent with a rewind. The authors do not justify dismissing this effect.

B) With the huge effect of striatal inhibition on lick times, it is difficult for me to interpret the lick-time conditioned analyses in Figure 6. Grouping based on activity (as explained above) would better show how the perturbation influences dynamics. It might also shed light on the no lick trials (i.e. does pre-perturbation activity predict a “no lick” trial).

We agree with the reviewer that the increase in no-lick trials required further explanation. We have addressed this in **Sections 1–3**. In brief, we can decode no-lick trials before inhibition onset, and found that longer perturbations increased the no-lick rate, consistent with the idea that when ramping (timing) dynamics decays closer to baseline, it becomes more difficult for the system to reinitiate the timer, increasing the likelihood of a no-lick trial. Thus, both the extended shift in lick timing and the increased proportion of no-lick can be explained within the same framework of timing dynamics decaying back to baseline during D1-SPN inhibition.

We also agree with the reviewer regarding the grouping method and performed a series of single-session analyses using a k-nearest neighbor decoder (Extended Data Fig. 12), as well as single-session and single-trial analyses (Revision Fig. 8), similar to those conducted for ALM silencing. Together, these results support the conclusion that activity decays below the pre-perturbation level during D1-SPN inhibition at the single-trial level. Specifically, during bilateral D1-SPN inhibition, ALM ramping activity decayed (Revision Fig. 8m), while maintaining its correlation with pre-perturbation activity (Revision Fig. 8l). After the inhibition, activity gradually recovered without a rapid recovery phase, resulting in a significantly longer recovery time compared to ALM silencing (Extended Data Fig. 12tu). For the direct statistical comparison with bilateral ALM silencing in Extended Data Figs. 12tu and 13, we did not include STR recordings during unilateral STR inhibition, as this manipulation was unilateral, unlike all others.

C) More generally, the rewind hypothesis feels less formulated than the pause hypothesis (which makes straightforward predictions). Given the work the authors put into building models, it would be helpful to see the exact same analyses applied to both the real data and data simulated from the model that has a rewind. Without this, it is not intuitive what one should expect to see from these specific analyses if the hypothesis were correct.

D) A final, related point again pertains to the behavioral effects. My intuition is that a rewind should have behavioral effects that do depend on the state of the timer (unlike a pause), though I am not confident in

this. In summary, the paper would be improved if the authors worked out more precise, quantitative predictions for the rewind hypothesis and tested them in neural and behavioral data.

We sincerely thank the reviewer for raising these important points. These points (C and D) are addressed in **Section I**, where we introduce new models that distinguish the effects of a 'pause' versus a 'rewind' on behavior and neural dynamics, and compare them directly with the data in a consistent format (Revision Fig. 1). Consistent with the reviewer's intuition, D1-SPN inhibition shifts lick timing in an onset-dependent manner (Revision Fig. 2), further supporting the notion of a 'rewind' mechanism. See **Section I** for the corresponding edits in the manuscript.

3) The hypothesis that the striatum integrates ALM activity to generate ramping activity and that ALM inherits its ramping from the striatum is not tested and thus not supported by the data. It appears that the authors generated a post hoc hypothesis after the data was collected and did not go back to directly test the hypothesis. As is, the conclusions seem far too strong given the evidence.

A) It's not clear if there is any evidence for an integration process in the data. Based on my reading, it seems that the authors instead assumed there was an integration process because previous models have shown that it is possible to produce ramping through integration. But, other models have shown that ramping can be produced through other means. Why should we conclude that ramping in the striatum is produced through integration? Indeed, one of the major claims in the discussion is that an integrator mediates the generation of timing dynamics. However, no alternatives to integration are considered or tested.

In the original manuscript, we proposed that ramping dynamics representing time arise from the temporal integration of step-like inputs (trial-history mode; TM), inspired by observed ALM activity patterns and prior theoretical studies²⁻¹⁰. To further support the integrator hypothesis, we performed a series of new analyses and causal experiments, including:

- 1) demonstrating a tight trial-by-trial correlation between ALM trial-history mode dynamics and striatal ramping activity (Revision Fig. 4);
- 2) additional manipulations (sensory cue) showing that transient perturbations exert graded and lasting effects on ramping activity (Revision Fig. 5).

In addition, we have included a detailed comparison of alternative models in the **Section 2-3** of this letter and Discussion section of the manuscript. Please refer to **Section II** for full details. In short, we confirmed that temporal integration is consistent with all our experimental results, and provides a parsimonious explanation given that alternative models cannot account for the effects of ALM silencing without additional complexity.

B) The general conclusion that the authors put forth is not clear to me. I believe it is as follows: the trial history mode in ALM supplies an input to the striatum that is integrated within the striatum to produce ramping dynamics. These ramping dynamics are then inherited by ALM through a multi-synaptic pathway. This took many reads to understand. Regardless of whether my read is correct, I had a lot of trouble

figuring out what the authors actually thought was going on. A more general audience will have an even harder time understanding the authors' conclusions/hypotheses.

Yes, the reviewer's summary of our conclusion is correct. We have revised the main text to improve clarity for a broader audience:

Page 17, Line 511

“Our findings support a model in which the striatum, potentially along with other subcortical areas situated between the striatum and ALM, function as an integrator generating timing dynamics in response to inputs provided via ALM (Fig. 5d, 6f, and Extended Data Fig. 1f and 15i). In this model, trial-history information is persistently encoded in ALM across trials (Fig. 4) and serves as input to the subcortical integrator. This input determines the slope of ramping activity, thereby adjusting lick timing based on previous trials. The resulting ramping activity is then relayed back to ALM via a multi-synaptic pathway, explaining similar ramping dynamics across brain areas and likely enabling ALM to trigger a timed lick¹⁹.”

C) Why doesn't striatum also integrate the ramping dynamics in ALM? The trial history mode and the ramp mode were forced to be orthogonal for convenience in the analyses, and they are assumed to be orthogonal in the model. But are the signals actually present in orthogonal subspaces in ALM?

We thank the reviewer for their questions. First, we note that the trial-history mode and ramp mode were not forced to be orthogonal. We added a clarification in the Methods as follows:

Page 26, Line 864

“We obtained an $n \times 1$ unit vector representing the rank correlation of each neuron and normalized it by its norm to calculate the trial-history mode. Trial-history mode was not orthogonalized to any other modes.”

Further, to address the reviewer's comment, we quantified the angle between the trial-history mode and ramp mode (Revision Fig. 10, Methods). In both ALM and striatum, the two modes were effectively orthogonal (no significant difference between data vs. shuffle controls).

Revise Fig. 10. Angle between ramp mode and trial-history mode is orthogonal (these panels are now Extended Data Fig. 5m). Left, ALM recording ($n = 3261$ neurons, 45 mice). Right, striatum recording ($n = 1073$ neurons, 16 mice). Dotted lines, angles of shuffled control. *P*-values, bootstrap test of the null hypothesis that the observed angle is equal to the shuffled control. In both ALM and striatum, ramp mode and trial-history mode are orthogonal.

Temporal integration of ramping activity via a recurrent excitatory loop would result in exponentially unstable dynamics that rapidly diverge (i.e., runaway excitation), which is inconsistent with the stable, reproducible ramping observed in our data. In our model, the trial-history mode in ALM provides input to the striatal integrator (i.e., on-manifold input), while ALM ramping activity projects onto a direction orthogonal to the integrator (off-manifold input). This orthogonal ramping input from ALM modulates the gain of ramping dynamics in the striatum, but does not contribute to the accumulation. This framework explains how striatal activity can decrease during ALM silencing while still preserving timing information. Biologically, this may be implemented through distinct connectivity patterns linking ALM neurons encoding trial-history information and those driving ramping activity to striatal neurons, or through selective gating within striatum.

D) There doesn't seem to be any evidence presented that suggests that striatum is specifically integrating the trial history mode in ALM. Could the simultaneous recordings be used to test this idea? I.e., do fluctuations in the trial-history mode in ALM correlate best with those in the ramp mode in striatum?

E) If ramping in ALM is inherited from the striatum, then there should be a strong trial-by-trial correlation between fluctuations in the ramping of the two areas. The authors did perform one analysis showing that a lick-time decoder trained on the two areas was correlated across trials. This is a good start, but if ramping in ALM is inherited through a multi-synaptic pathway, then there should be a significant lag in the correlations between the two ramping modes. The direction of the lag should be in the opposite direction for correlations between the trial-history mode (or its integral) in ALM and the ramp mode in the striatum.

To address these points (D and E), we calculated 1) the across-trial Pearson's correlation between ALM's trial-history activity and striatum's ramping activity, and 2) the correlation between the ramp mode activity in ALM and striatum, in simultaneously recorded sessions (Revision Fig. 4). Remarkably, ALM trial-history mode activity shows a strong correlation with striatal ramping activity, and ramping activity in ALM and striatum is highly correlated at the single-trial level. These findings are consistent with the hypothesis that the striatum integrates ALM's trial-history activity, which in turn is inherited by ALM. See **Section 2-1** for details, and edits in the main manuscript.

Additionally, in individual simultaneous recording sessions, we estimated the temporal offset between ramping activity in the two areas by identifying the time point at which ramp mode activity reached 75% of its peak (Revision Fig.4f). However, no consistent trend was observed. The absence of a clear temporal lag between the two areas may be due to sampling bias in the temporal profiles of recorded ramping neurons. Within each region, some ramping neurons begin ramping earlier or later than others (Fig. 3b and g), making onset measurements highly

sensitive to neuronal sampling within each session. Thus, accurately estimating the temporal lag between ramping activities remains challenging without a substantial increase in the number of simultaneously recorded neurons. Finally, the ALM–striatal interaction is not purely feedforward: in our model, ALM ramping activity modulates the gain of striatal ramping. Consequently, we do not necessarily expect a temporal lag in ramping activity between these two recurrently connected regions.

F) It is unclear to what degree the favored model actually fits with the data. As mentioned above it would be helpful to see the exact same analyses performed on both the real data and the model simulated data so that the reader can directly compare them.

In Extended Data Figs. 1 and 15, simulated neural activity is projected onto the ramp mode using the same method applied to the experimental data, allowing for direct comparison. We have now included lick-time analysis (determined based on the ramp mode activity; Methods) in these models to allow explicit comparison with the experimental data (Revision Fig. 11). Furthermore, we implemented a model without assuming a specific network architecture to distinguish the computational signatures of pause/slowdown and rewind mechanisms (**Section 1-1**; new Extended Data Fig. 19). We believe these additions enable a more direct comparison between the data and the models.

Revision Fig. 11. Comparison of shifts in lick time following perturbation across positive feedback models and data (this is now in Extended Data Fig. 1g). The last column (Data) was duplicated from Fig. 5b, and 6d for comparison purposes.

Minor comments:

1) The rainbow (i.e. “Jet”) colormap (used in all of the heat maps) is known to not accurately represent quantitative data and produces illusionary boundaries (D. Borland and R. M. Taylor II, “Rainbow Color Map (Still) Considered Harmful,” in IEEE Computer Graphics and Applications). The colormap may also be confused with the colors used in the PSTHs.

We have updated the heatmaps in Fig. 3b, c, g, and h, Fig. 5e and h, and Fig. 6h and k using a perceptually uniform colormap (“parula”). For PSTHs displaying trials with different lick times,

we used the “turbo” colormap, which provides diverse yet more perceptually uniform colors compared to the “jet” colormap

(<https://research.google/blog/turbo-an-improved-rainbow-colormap-for-visualization/>)

2) The definition of “ramping” neurons (¶ starting at line 183) is not convincing. Many non-ramping activity profiles would produce differences in activity between the lick epoch and baseline (e.g., oscillations, motor bursts, etc.).

We performed a detailed characterization of the ramping profile at the single-cell level. To this end, we adapted published methods¹⁵ (Revision Fig. 12, Methods). Specifically, the PSTH of each neuron was fit with polynomial functions of varying orders, and the goodness of fit was evaluated on held-out trials. The polynomial model with the best fit was used to determine the order, assess whether a neuron’s firing pattern was monotonic (i.e., whether its derivative remained consistently positive or negative from cue to lick), and determine the peak firing time (Methods). Based on this analysis, 59% of total ALM neurons showed peak firing within 100 ms at cue or lick onset (Revision Fig. 12d), and 37% of these neurons showed monotonic ramping activity (corresponds to 23% of all ALM neurons; Revision Fig. 12c); 67% of total striatal neurons show peak firing within 100 ms at cue or lick onset (Revision Fig. 12d), and 37% of these neurons showed monotonic ramping activity (corresponds to 28% of all striatum neurons). Overall, around a quarter of neurons show ramping activity. Nonetheless, our conclusions do not rely on individual neurons exhibiting perfect ramping, but rather on the broader temporal structure observed across the population.

Revision Fig. 12. Proportion of ramping cells (these panels are now in Extended Data Fig. 4)

a. Example cells with polynomial fitting. Black: training data used to fit the polynomial; blue: test data used to evaluate the fit (mean square error between fit and test data; 10-fold cross-validation); red: best-fitting polynomial. We performed this analysis on activity from cue to lick in trials in which mice licked at 1.25 - 1.5 s (Methods).

b. Distribution of best-fitting polynomial order for ALM (black) and striatum (green); 2418 and 553 neurons, respectively (Methods).

c. Proportion of neurons with monotonic activity between cue and lick, defined as either linear or polynomial fits with derivatives consistently negative or positive from cue to lick.

d. Distribution of peak activity timing shows that peaks are concentrated at the beginning or just before the lick, indicating that even non-monotonic neurons often resemble ramp-up or ramp-down profiles rather than exhibiting peak activity mid-trial. Histogram bin size, 0.14 s.

We modified the definition of “ramping” neurons of ALM and striatum in the main text as follows:

Page 5, Line 160

“Many ALM neurons displayed ramping activity during the task, peaking around the onset of cue or lick (Fig. 3a; based on single-cell polynomial fitting analysis, 59% of neurons showed peak firing within 100 ms at cue or lick onset, and 37% of these neurons showed monotonic ramping activity; Extended Data Fig. 4a-d; Methods).”

Page 7, Line 200

“and many neurons displayed ramping activity (based on polynomial fitting analysis, 67% of neurons show peak firing within 100 ms at cue or lick onset, and 37% of these neurons showed monotonic ramping activity; Extended Data Fig. 4a-d; Methods).”

3) The paper from O’Shea, Duncker, et al (2022), BioRxiv, is highly relevant and should be cited.

We have added this reference in the text below:

Page 3, Line 86

“Depending on the computational role of the manipulated brain area, multi-regional dynamics are expected to respond to and recover from brief disturbances differently^{47,48} (Fig. 1c and Extended Data Fig. 1).”

Page 17, Line 546

“Therefore, combining transient perturbation with large-scale electrophysiology is critical to dissect multi-regional dynamics across behaviors^{47,48}.”

Referee #2 (Remarks to the Author):

Major comments:

1) The underlying premise of the work is that the timing behavior observed is due to an integration of a step-like (constant) input into a ramp. The step-like input is further speculated to possibly be the same step observed during the pre-cue 'trial-history' period, which is then gated into being able to be integrated by the presentation of the cue. These seem like reasonable hypotheses, but it would be nice if they could be more directly supported by data.

a) The first assumption – that there is a step-to-ramp integration – seems like it is critical for the framework of analysis and modeling. Is there a way to more directly test this hypothesis? For example, if on occasional trials the cue was made stronger or longer, might this lead to greater 'on manifold' input that would then be integrated and could be seen experimentally?

We thank the reviewer for the insightful comments. To address it, we performed the proposed experiment by randomly interleaving trials with louder or fainter cues. As the reviewer speculated, this manipulation altered the step input and resulted in lasting changes in both ramping activity and behavior, consistent with our hypothesis. Please refer to Summary **Section 2-2** for details.

b) Could the trial history activity instead just shift the initial condition of the network, thus leading to different ramping dynamics (rather than serving as a constant input to an integrator)? This possibility is mentioned at the start of the text on the trial-history activity, but then dropped in favor of the integrator interpretation, but I wasn't sure why it was dismissed.

The initial condition hypothesis can indeed account for the dynamics observed in control trials without perturbation. However, it cannot readily explain the recovery of ramping activity following ALM silencing without invoking an additional mechanism to maintain timing information. In contrast, the integrator hypothesis naturally accommodates both the reliable unfolding of ramping activity and its recovery after perturbation, making it a more parsimonious account of the data. Please see **Section 2-3** for a detailed discussion and corresponding edits in the manuscript.

2) I love the modeling idea that ALM serves as the input to an integrator involving the striatum and also as an output to a pattern of ALM activity that does not get integrated by the striatum. However, at least in the model of Extended Figure 1, this idea is implemented by having an ALM population that should be a pure constant step of activity without showing ramping (neuron A1 in Extended Fig. 1f). Is there evidence for this in the data? If, instead, one should think of populations A1 and A2 as two more general patterns of activity that can be intermixed at the level of recorded neurons' activity (i.e. a given neuron may reflect a certain amount of pattern 1 and a certain amount of pattern 2, in a manner that reflects the experimental recordings), is it still possible to make a model in which ALM is an input and an output in a manner that recapitulates the data but doesn't form a feedback loop and thus act as part of the integrator?

In the reduced network model, each unit represents a mode of population activity rather than a single neuron in each area: one corresponding to the step-like input and the other to ramping dynamics. We argue that the activity of individual neurons in ALM can be well approximated as a combination of these orthogonal modes. Here, we present evidence supporting this interpretation.

First, while some individual neurons in ALM exhibit clear ramping or step-like activity (Fig. 3af and Extended Data Fig. 4ij), many neurons display a mixture of both response types. The angle between the trial-history mode and ramp mode is orthogonal in ALM (Revision Fig. 10 in response to Reviewer 1, comment 3-C), consistent with the view that ALM neural responses can be explained as a combination of these two independent modes of activity.

Second, we performed PCA analysis on ALM and striatal recording data to independently investigate the population structure (Revision Fig. 13, Methods). The top 2 PCs in ALM resembled the ramp mode (PC1) and trial-history mode (PC2), respectively (Revision Fig. 13a). The alignment between the ALM ramp mode and PC1, and between trial history mode and PC2, was significantly stronger than that observed in shuffle controls (Revision Fig. 13d). Moreover, in ALM, activity along PC2 showed significant modulation across trials predicting upcoming lick times, similar to the trial-history mode (Revision Fig. 13b). Such modulation was absent in the striatum, consistent with the observation that the trial-history information is minimally represented in the striatum (Revision Fig. 13f, Extended Data Figs. 4k, and 8b). Thus, ramp and trial-history modes closely align with the top two principal components in ALM, suggesting that they are not arbitrarily defined activity patterns but instead represent dominant, orthogonal patterns of activity in ALM.

Together, the diverse neural activity in ALM is well captured by a linear combination of orthogonal step- and ramp-like responses, represented by A1 and A2 in the reduced model. This means the model in principle can be extended to include additional neurons exhibiting diverse mixtures of trial-history and ramp modes, without altering the overall multiregional dynamics.

Revision Fig. 13. Angle between PC and modes (this is now in Extended Data Fig. 9)

a. Projection of ALM activity to PC1 (left) and PC2 (right). $n = 3261$ neurons, 45 mice. Shade, 95% confidence interval.

b. Relationship between actual lick time and amplitude of activity along each PC during ITI (Amplitude is normalized to the activity in trials with the shortest lick times). P -values, bootstrap test of the null hypothesis that the linear regression slope between these two is larger than 0. Notably, PC2 shows a significantly negative slope, similar to the trial-history mode.

c. Cumulative variance explained by PCs.

d. Angle between each PC and ramp mode or trial-history mode (histogram of 1000 bootstrap iterations is shown). Dotted lines, angles of shuffled modes. P -values, bootstrap test of the null hypothesis that the observed angle is equal to the shuffled angle. PC1 and PC2 are significantly aligned with ramp mode and trial-history mode, respectively.

e-h. Same as **a-d** but for striatal activity ($n = 1073$ neurons, 16 mice). PC1 and PC2 are aligned with the ramp mode, but none of the top principal components aligned with the trial-history mode (we tested the top 5 PCs; only the top 2 are shown here).

3) The alternative model of sequences in Extended Data Fig. 11 instead has the external input going to the striatum. However, this model seems to have ALM being part of the integrator (but only weakly due to very weak interconnections between ALM and striatum), as I think is evidenced by the activity decay seen during ALM silencing. Also, it has some strange dynamics where activity can either stay constant or increase during D1 SPN inhibition and it seems to have a very different cue mode (at least in panel b) from the data. It would be helpful to discuss these discrepancies and whether they are contradictory to the data.

Recurrent networks connected via feedforward connectivity can generate both sequential and ramping activity, as observed in the data. However, as the reviewer pointed out, the multi-regional feedforward model presented in the original manuscript had several issues: in the original model, we used a single self-exciting neuron per feedforward layer per brain area. This resulted in an amplification of all inputs by the recurrent/feedforward chain, making it difficult to implement the cue mode like in the data and to separate the functions between brain areas.

To address these issues, we reformulated the model using an idealized architecture in which the functional roles of each region can be more clearly differentiated. Specifically, in the updated model, we have two mutually inhibiting, self-exciting neurons per feedforward layer per brain area (single brain area model is in **a-c** and multi-brain area models are in **d-i**). This allows for selective amplification of inputs depending on how they are provided, enabling the model to reproduce the cue response observed in the data, along with ramping and sequential activity, and to separate the functional roles of the two areas.

To implement this, the recurrent network modules shown in Extended Data Fig. 1 were duplicated with weaker recurrent connections to prevent perfect integration within each module. These quasi (leaky)-integrator modules, with distinct functions assigned to ALM and the striatum (STR), were then connected via feedforward connections, enabling the generation of both sequential and ramping activity in response to step inputs.

Each model has: a quasi-integrator implemented by both ALM and STR connected via feedforward connectivity in STR (Revision Fig. 14f; distributed, akin to Extended Data Fig. 1b); redundant quasi-integrators and feedforward connectivity in both ALM and STR (Revision Fig. 14g; redundant, akin to Extended Data Fig. 1c); a quasi-integrator and feedforward connectivity only in ALM, with STR acting as a follower (Revision Fig. 14h; ALM integrator, akin to Extended Data Fig. 1d); and a quasi-integrator and feedforward connectivity only in STR, with ALM acting as a follower (Revision Fig. 14i; STR integrator, akin to Extended Data Fig. 1f). See Extended Data Table 2 for full description of connectivity matrix and inputs.

Two distinct inputs were provided to ALM (or to STR in the case of model h) in the first feedforward layer (as in Revision Fig. 14a). A step input was delivered to a single neuron, allowing it to be amplified by the recurrent connections and propagated to the next layer. In contrast, a transient input at cue onset was delivered to both neurons in the mutually inhibiting module, preventing its amplification and transmission to downstream feedforward layers.

These changes result in model behavior that more closely matches the cue mode activity and effects of D1-SPN inhibition seen experimentally.

Revision Fig. 14. Multi-regional feedforward network that generates sequential and ramping activity (Excerpt from Extended Data Fig. 15)

- a. Schema of the feedforward network.
- b. Activity along different modes in the network described in a.
- c. Correlation in neural population activity patterns in the network described in a. Trials with early lick (left) and late lick (right) are shown. Note that similar activity patterns emerge, but unfold at different speeds.
- d. A multi-regional network where the integrator module within ALM and striatum are connected with feedforward connections.
- e. Comparison of shifts in lick time following perturbation across feedforward models and data (Methods; the lick time was estimated in red traces in each plot). The last column (Data) was duplicated from Fig. 5b, and 6d for comparison purposes.
- f. A multi-regional network where integration is via interareal connections between ALM and striatum (similar to Extended Data Fig. 1b). Perturbation of ALM resets the integration. Plots are shown up to the time of the lick.
- g. Similar to f, but a multi-regional network where both ALM and striatum function as integrators (similar to Extended Data Fig. 1c). Perturbation of ALM resets the integration.
- h. Similar to f, but a multi-regional network where ALM functions as the integrator (similar to Extended Data Fig. 1d). Perturbation of ALM resets the integration, while STR inhibition only produces a weak shift of the dynamics after the inhibition.
- i. Similar to f, but a multi-regional network where STR functions as the integrator (similar to Extended Data Fig. 1f). ALM silencing results in a pause in the representation of time (both in RM activity, left, and correlation, right). In contrast, STR inhibition decayed the RM activity.

4) The models have the striatum driving (exciting) ALM. However, as noted in the text, the experiments in Figure 6e suggest that silencing striatum has little net effect on ALM activity. How does this fit with the models?

As the reviewer pointed out, the proportion of ALM neurons significantly modulated by D1 SPN manipulation was 35%. Despite this, ALM ramp mode activity and decoded time were significantly reduced during the manipulation (Fig. 6), suggesting that D1 SPN is not a general excitatory drive of ALM but specifically drives the ramping activity patterns. Consistently, ALM neurons that are significantly suppressed during D1 SPN inhibition exhibited ramping-up activity on average in unperturbed trials (Revision Fig. 15). In the model, the striatum specifically drives ramping activity in ALM, consistent with experimental data.

Revision Fig. 15. Activity of ALM neurons significantly inhibited during bilateral VLS silencing (this panel is now in Extended Data Fig. 14q). Blue, silencing trials. Black, control trials. Lines, grand mean. Shades, SEM (hierarchical bootstrap). These neurons exhibit ramping-up activity on average in control trials ($n = 24$ cells).

5) The models are hard to interpret due to having ‘arbitrary units (a.u.)’. In the methods, it is stated that striatal activity starts from a baseline of 5 Hz, corresponding to experimental recordings. But, then, ‘arbitrary units’ are used. First, are the ‘arbitrary units’ the same for ALM and STR activity and for the different cases d-g? I appreciate that these are population activities, so the same population activity could be due to high firing rates of a smaller number of neurons or low firing rates of a larger number of neurons, but it is very hard to compare the models to data or to each other with such arbitrary (and possibly changing from network to network or even from one population to another?) units. Can the authors instead show the real units that emerge from the model, especially given the methods compares the model to 5 Hz striatal firing rates, and then just explain that these are population values. Most critically here is to have the same, unnormalized units for all nodes of the network and for all simulations so one can get a better idea of what is happening.

To better align the models with our recording data, we revised the normalization procedure to ensure consistent units across all models. In the updated projection plots, a value of 0 corresponds to the baseline firing rate (5 Hz) of the target neuron (neuron 3 in all cases, or neuron 4 in the ALM leaky integrator model), while a value of 1 corresponds to the time point when the target neuron's activity reaches 10 Hz, approximating the activity level just before licking in our data. Thus, the absolute spike rate of the target neuron can be inferred from its activity along the ramp mode, providing a direct mapping between model output and population activity.

We show here the raw spike rate of individual ‘neurons’ for the model in Extended Data Fig. 1f (Revision Fig. 16). Yet, due to space constraints in Extended Data Fig. 1, we can not show the raw traces for all neurons. The full simulation code is available on GitHub (link provided in the Methods).

Revision Fig. 16. Activity of all neurons in the model in Extended Data Fig.1f.

A1 and A2 are the 2 neurons in ALM, S1 and S2 are the 2 neurons in striatum. S1 is the target neuron, traces are plotted until the time point when S1 neuron (target neuron) reaches 10 Hz.

6) Is it possible that striatum could instead be an input to, and readout of, the integrator, similar to how ALM is being interpreted but with one difference: in order to explain the rewinding, perhaps the zero point of the integrator occurs when the striatal activity is at baseline, and then silencing of the striatum is effectively providing negative input to the integrator to cause rewinding? For this to work, one would need a model in which (like the ALM model) some striatal neurons (or possibly activity patterns, as discussed in point #2 above) are inputs to the integrator and some are readouts. Anyway, the point here is that, if one considers silencing of the striatum as being negative input, then one could imagine striatum being analogous to how ALM is described in the current model. Is there a contradiction with this possibility?

We thank the reviewer for raising this interesting alternative scenario. We considered this possibility in our work, however, the scenario that the baseline spike rate in the striatum serves as the zero-point is not supported by our data. In particular, D1-SPN inhibition during the inter-trial interval (ITI) did not produce clear effect on ALM dynamics along ramp mode, nor shift the lick time (unlike during the delay epoch; Revision Fig. 17, Fig. 6c and d). Alternatively, we can think of a scenario where D1 and D2 SPNs function as push-pull inputs to a downstream integrator, with both driven by ALM. In this scenario, some striatal neurons would function as inputs to the integrator (i.e., trial-history mode), others as a readout. However, this is not supported by our recordings, as we observed minimal trial-history mode activity in the striatum (Extended Data Fig. 4k, Extended Data Fig. 8b, and new Extended Data Fig. 9e and f). That said, we cannot fully exclude this possibility, as such neurons may not have been sampled. Accordingly, we have added a clarification to the Discussion as follows.

Page 18, Line 571

“Alternatively, integration may occur in two stages: D1-SPN provides excitatory input to the integrator, while D2-SPN suppresses it, making the striatum a push-pull controller of downstream integrators in response to ALM input. Although the weak, trial-history mode activity in the striatum (Extended Data Fig. 4k, Extended Data Fig. 8b, and 9f) argues against this model, it cannot be fully excluded.”

**ALM recording
Bilateral striatum
inhibition
(during ITI)**

Revision Fig. 17. Inhibition D1-SPN during ITI does not affect ALM activity along ramp mode, despite it modulating activity in ALM (Extended Data Fig. 14m). (This panel is now in Extended Data Fig. 12s).

7) Why isn't there a plot analogous to Fig. 6g for the ALM silencing experiments of Fig. 5, rather than a cartoon? In Extended Fig. 12b, it's nice that the activity appears to move parallel to a vector going towards zero in panel c, but it also appears to move somewhat backwards (between 90 and 180 degrees) relative to during control in panel b. It's not obvious to me that the vector towards 0 is orthogonal to the integrating 'on manifold' direction, so the result in panel c doesn't seem to preclude ALM being partly involved in rewinding of integration.

We added a 2D projection depicting the ALM silencing result as Fig. 5e.

Revision Fig. 18. Striatum activity in a two-dimensional space along the ramp and middle mode during ALM delay silencing. (Excerpt from Fig. 5). The mean trajectory of trials with silencing (lick between 1.7 - 2.0 s) is shown. Arrows indicate the direction in which the trajectory evolves.

In addition, we have updated Extended Data Fig. 17 by pooling new recording data (collected for **Section 1**; 28 sessions and 9 sessions in total for ALM silencing and D1 inhibition, respectively) to increase the sample size. With the increased sample size, the difference between panel **b** and panel **c** has become even clearer (although the same trend was already present in the previous version): the vectors are widely distributed in panel **b** (rewind) but highly confined in panel **c** (movement toward 0). The confidence interval of the mean angle in panel **c** is nearly ten times smaller than that in panel **b**. This suggests that ALM activity during ALM silencing is better explained by movement toward the zero point.

Revision Fig. 19. (excerpt from Extended Data Fig. 17)

8) I agree with the authors statement in the discussion that the medium spiny neurons of the striatum seem like an unlikely location for integration on their own. However, it would be nice to provide additional justification for why the authors think the integrator is a subcortical network involving the striatal neurons as opposed to some other set of pathways involving the striatum—is this based on prior inactivation experiments ruling out other neocortical areas, so that it needs to be subcortical? A bit more justification for likely candidates for the feedback pathways would be helpful, and also for the idea of the striatum mediating a mutually inhibitory positive feedback loop between striatal-associated populations that receive sufficiently strong common excitatory input from one ALM population, as in Extended Fig. 1f, in order to not have one population shut off, effectively disconnecting the mutually inhibitory feedback loop.

The orofacial sector of the striatum (VLS) receives excitatory input from ALM and surrounding dorsal cortical areas (which are mostly silenced by our 8 spots silencing centered around ALM¹), as well as from the intralaminar thalamic nuclei^{16,17}. The dorsal cortical optogenetic screening (Fig. 2) and ALM silencing results (Fig. 5) suggest that although the cortex provides input to the integrator, it is not itself part of the integration mechanism. This led us to reason that integration is mediated by subcortical structures.

Given that the striatum is primarily inhibitory, integration is likely implemented via a long-range loop involving the striatum, substantia nigra pars reticulata (SNr), and intralaminar thalamus, which projects back to the striatum and forms a long-range excitatory (disinhibitory) loop.

As the reviewer pointed out, shared excitation effectively linearizes the mutual inhibitory dynamics and ensures that activity in both populations remains within a dynamic range where mutual inhibition serves to regulate the ramp. Without this excitatory input, one population may dominate and suppress the other entirely, breaking the integrator. The mutual inhibition required for integration may arise from interactions between ALM-recipient direct and indirect pathways within the VLS or in VLS-recipient downstream regions such as the SNr¹⁸, a possibility to be explored in future studies.

To incorporate these discussions, we have revised the main text as follows:

Page 18, Line 566

“VLS receives excitatory input from ALM and surrounding dorsal cortical areas, as well as from the intralaminar thalamic nuclei^{57,58}. Our cortical optogenetic screening (Fig. 2) and ALM silencing (Fig. 5) suggest that while the cortex provides input to the integrator, it is not the integration site itself. Instead, integration may be implemented via a subcortical long-range loop involving the striatum, substantia nigra pars reticulata (SNr), and intralaminar thalamus, which projects back to the striatum to form a long-range disinhibitory loop^{57,58,74,75,89,90}. Alternatively, integration may occur in two stages: D1-SPN provides excitatory input to the integrator, while D2-SPN suppresses it, making the striatum a push-pull controller of downstream integrators in response to ALM input. Although the weak, trial-history mode activity in the striatum (Extended Data Fig. 4k, Extended Data Fig. 8b, and 9f) argues against this model, it cannot be fully excluded. The mutual inhibition required for integration may arise from interactions between ALM-recipient direct and indirect pathways within the VLS or in VLS-recipient downstream regions such as the SNr⁹¹. Together, while we identified VLS as a key contributor to the integration process, future perturbation experiments across areas within this long-range subcortical loop may help elucidate how the integrator is fully implemented in these subcortical areas.”

9) It would be worth discussing (and probably referencing right up front in the introductory paragraphs motivating the current work) more about previous work that likewise suggested neocortical areas to perhaps not be the integrator but striatum being part of the integrator. I'm thinking in particular of the set of papers from the Brody lab that also used recordings, perturbations, and modeling to come to this conclusion in the context of a different, accumulation-of-evidence based decision-making task.

We added the discussion of the striatum being part of the integrator in the Introduction, including referencing the Brody lab's work:

Page 2, Line 70

“Additionally, distinct neural correlates⁴² and perturbation effects of the frontal cortex and striatum on evidence accumulation^{43,44} suggest the striatum may play a key role in integration during decision-making. However, most of these studies examined neural correlates and causal manipulations separately, limiting their ability to reveal the specific computational roles of each area and the interactions between them.”

Minor comments:

1) In Extended Fig. 1, the drawings of the model networks should accurately convey the weight matrix. For example, in the critical ‘consistent with data’ model, it really is A1 and A2 neurons with self-connections, in which A1 drives S1 and A2 drives both S1 and S2, with only S1 projecting back to A2. This is not remotely obvious from the figure, but is critical for understanding the model and (as noted in Major comments above) critically evaluating whether it does, or does not, reflect the data. When doing this, different notation should be used for excitatory vs inhibitory connections.

We agree that the cartoons did not accurately reflect the connectivity matrix. We have revised them in Extended Data Fig. 1 as follows to more accurately represent the underlying connectivity matrices.

Revision Fig. 20. (Excerpt from Extended Data Fig. 1)

2) For figures like Fig. 2b, it would be nice to show an average session plus error bars rather than just an example session (I understand that panel c is cumulative, but it shows somewhat different information if I'm understanding it correctly).

Because the duration of each delay block was randomized to prevent animals from predicting block transitions, session-averaged data cannot be presented in the same format as Fig. 2b.

Instead, Fig. 2c shows the cumulative distribution of lick times between different delay blocks to show that mice adapt lick timing. Here, data are pooled across sessions and animals, and the shaded area represents the confidence interval calculated using hierarchical bootstrap. To clarify this, we have added the title “Average across all sessions” to Fig. 2c (Revision Fig. 21).

Additionally, in Extended Data Fig. 2d, we present changes in lick time at delay block transitions across sessions and animals, showing both the mean across mice and individual mouse data. This demonstrates that mice can adapt their lick timing based on the delay duration.

Thus, population-level data are shown (Fig. 2c and Extended Data Fig. 2d) alongside an example session (Fig. 2b) to provide a more comprehensive view.

Revision Fig. 21. (Excerpt from Fig. 1c)

3) I couldn't understand what cue, middle, ramp mode meant until reading the methods, yet they are critical for the analyses. If possible, try to define briefly in the main text.

We appreciate this feedback and have revised the main text as shown below to improve clarity:

Page 7, Line 177

"To further characterize the population activity patterns between the cue and the lick, we used targeted dimensionality reduction to define three modes (directions in population activity space) that capture task-related activity (Methods). Each mode represents changes in activity during a specific task epoch: the cue mode (CM) reflects the transient response to the auditory cue (0 - 300 ms after the cue), the middle mode (MM) captures activity bridging cue and movement preparation (500 - 800 ms before the lick), and the ramp mode (RM) captures the activity preceding lick initiation (200 - 500 ms before the lick), which exhibits a ramping profile."

4) The trial history mode was even more difficult to understand, as the main text refers to the methods for understanding it and then the methods refers back to Fig. 2d (which beyond a regression formula, doesn't describe it either). Please define clearly in the methods (and, ideally, include a simpler brief explanation in the main text).

We apologize for the lack of a detailed description of the trial-history mode. We have now revised both the main text and Methods as shown below:

Page 8, Line 238

"To characterize the evolution of ALM activity encoding trial history at the population level, we defined a "trial-history mode" by constructing a population vector in which each neuron's contribution was weighted by the strength of its correlation with trial history during the ITI. We then projected population activity onto this vector to quantify trial-history-related dynamics (Fig. 4b, left; Methods)."

Page 26, Line 857

"To define the trial-history mode, we first calculated the predicted lick time in each trial by applying the linear regression model described in the 'Trial-history regression analysis' section for each recorded session. Specifically, the model included previous lick times and the interaction between previous lick time and outcome at lags 1 and 2. This predicted value estimates what the lick time would be if it were determined solely by recent behavioral history and reinforcement, according to the fitted regression model, thereby summarizing trial history as a single value for each trial. We then calculated the Spearman rank correlation between the spike rate during ITI (0 - 1 s before the cue) and the predicted lick time across trials for each neuron, indicating how strongly each neuron's ITI activity encodes trial history. We obtained an $n \times 1$ unit vector representing the rank correlation of each neuron and normalized it by its norm to calculate the trial-history mode. Trial-history mode is not orthogonalized to any other modes."

5) I'm not a huge fan of the phrase 'similarity matrices' as I think of a similarity matrix as one with the same x and y axes, as in representational similarity analysis matrices. Just calling it correlation as in Fig. 5 seems better. And, as with 'modes', it would be good to define what this graph is more clearly in the main text (it wasn't even obvious until I read the methods that each row is a neuron, given that the y-axis is labeled as 'reference trial', which suggests that each row is a trial).

We have changed all the "similarity matrices" to "Correlation in neural population activity" or "correlation matrices" for short. We have also added a brief description in the main text as shown below:

Page 6, Line 171

"To quantify this observation, we calculated Pearson's correlations of neural population activity across time points, comparing activity in trials with different lick times (Fig. 3c)."

6) More terminology: 'variance' of spiking activity explained is not really a variance, as it is relative to baseline activity rather than relative to a mean activity. Please rephrase to more accurately convey what is being plotted.

To accurately describe the plot, we have revised the term to "square sum (SS) of task-modulated spiking activity". All changed terms are highlighted in yellow in the revised manuscript.

7) Is the value of I_{ext} given for the models? More generally, please check that all parameter values are given somewhere numerically (and not just with a color bar, as in the feExtended Data Forward network matrices).

We now provide all the parameters for both Extended Data Figs 1 and 11 (11 is now 15 in the revised manuscript) in Extended Data Table 2. In addition, all model codes are available on GitHub.

8) I didn't understand why gray line in Extended Fig. 4e was compared to the 3rd quartile.

The gray trace represents trials that followed **unrewarded** trials, where the lick time on the previous trial was within a specific range (the third quartile). The orange trace represents trials that followed **rewarded** trials with the same range of previous lick times. This allows us to compare the effect of reward on upcoming trials while controlling for the influence of prior lick timing. In this example, the cell shows a lower spike rate during the intertrial interval following unrewarded trials compared to rewarded ones.

To make this point clear, we have revised the legend of Fig. 4a and Extended Data Fig. 4i as follows:

“Bottom, PSTH. Lick times of the previous rewarded trials were divided into quartiles indicated by different colors. The gray trace, trials following previously unrewarded trials with previous trial’s lick times within the 3rd quartile. The orange trace represents trials that followed rewarded trials with the same range of previous lick times. This allows us to compare the effect of reward on spiking activity in upcoming trials while controlling for the influence of prior lick timing. In this example, the cell shows a lower spike rate during the inter-trial interval following unrewarded trials compared to rewarded ones.”

9) Is it surprising that the correlations in Extended Data Fig. 5 go to nearly 1 for potentially noisy spiking data? Or does this reflect some previous averaging out or removal of noise in the data analysis?

We assume the reviewer was referring to panels **e** and **j**. Yes, this high value is partly due to averaging and smoothing: for each lick time range, we averaged across at least 10 trials with lick times within that range, and spiking data for each trial were smoothed using a 200-ms boxcar filter. Also, please note that this represents the squared sum of task-modulated activity, with baseline activity during the inter-trial interval subtracted. To clarify the data processing procedure, we have now added the following sentence to the legend of Extended Data Fig. 5 and in the corresponding Methods section.

Page 42, Line 1436

“Square sum of task-modulated spiking activity explained by cue mode (a3, Methods). Note that this value was calculated after trial averaging (10 trials) and smoothing (200ms causal boxcar filtering).”

Page 27, Line 873

“To calculate the square sum of spiking activity explained by individual modes (Extended Data Fig. 5), we calculated the square sum of the activity along individual modes after subtracting the baseline activity (0 - 0.2 s before the cue), and then divided that by the square sum of the spike rate across neurons after subtracting the baseline activity. For each lick time range, we averaged across at least 10 trials with lick times within that range, and spiking data for each trial were smoothed using a 200 ms causal boxcar filter.”

10) Extended Data Fig. 8a. I think this should say “cumulative” lick time distribution.

We have changed all related titles and captions to “cumulative lick time distribution”.

11) Extended Data Figs. 9d and 12a,d captions do not separately refer to/describe what is plotted in the left versus right panels.

We have revised the legends as follows (Extended Data Figs. 9 and 12 are now Extended Data Figs. 12 and 17, respectively):

Page 50, Line 1679

“Decoded Tto lick from each time point based on kNN decoding analysis of population activity (left). Lines, grand mean. Shades, 95% confidence interval (hierarchical bootstrap). The decoded Tto lick was normalized by subtracting the decoded Tto lick at stimulus onset to account for different lick times across trials and sessions (right).”

Page 57, Line 1872

“a and d, two example sessions.”

12) In Extended Data Fig. 11e, the coupling should perhaps be called ‘very’ weak (and this is a case where somewhere giving numbers rather than just colors for matrix values would be very helpful).

We have significantly revised the feedforward model, and thus, there are no longer very weak connections in the model (see our response to the major comment 3 above).

Referee #3 (Remarks to the Author):

This manuscript studies the roles of ALM and striatum in a lick timing task. It reaches the conclusion that ALM and striatum have different roles, with striatum serving as an integrator and ALM providing inputs into this integrator. The paper uses a combination of electrophysiology and optogenetic manipulations to dissociate the roles of these areas. The conceptual conclusions of the paper are exciting, with a potentially significant advance in understanding for the field. The approaches are innovative and applied in a systematic and thorough manner. Overall, the paper is well written and presented. My major reservation is that the main findings of the papers in Figures 5-6 could benefit from additional support. From the current presentation, it is not clear how large the difference in effects are between ALM and striatum inhibition. Further work is needed to clarify and quantify the differences to support the most important and interesting conclusions of this paper.

We acknowledge the reviewer for thoughtful and important feedback. We have addressed the concern in detail below.

Specific comments

1. The text (abstract and main text) make major claims about different roles for striatum and ALM. In large part, these claims are based on the inhibition results from Figures 5 and 6. A key part of the argument is that after the ALM inhibition, the population activity returns to the ramping right where it left off at the start of the inhibition. In contrast, the striatum inhibition is interpreted as winding down the ramp of activity and restarting the ramp from a lower position after the inhibition ends.

From the data in Figure 5-6, it is not clear that the post-inhibition starting point for population activity is greatly different for ALM and striatum inhibition. I do not see the difference as being very apparent when comparing panels 5f,g,i,j with panels 6i,j,l,m, specifically when looking at the immediate post-inhibition time points. Similarly, in Extended Data Fig 9, the pre vs post differences look similar between panels a,e vs i,m. Also, in the time to lick decoding in Extended Data Fig 9, the confidence intervals for return to pre-stim level are overlapping (and nearly the same) for the ALM vs. striatum inhibition. There are other pieces of evidence that help the interpretation put forward by the authors, but overall more evidence is needed here. I am not sure what is best, but from looking at the panels mentioned, it does not jump out to me that there is a major difference between these manipulations. It would be helpful to see direct comparisons and quantifications with statistics showing the differences. These direct comparisons would provide stronger evidence about the effect size and about the statistical significance.

We thank the reviewer for the comment. To address this and the following comments, we performed additional modeling and analyses to clarify the distinction between pause and rewind (**Section I** in this letter). In short, pause and rewind are expected to produce inhibition-onset-independent and inhibition-onset-dependent shifts in the lick time distribution, respectively. Additionally, the hazard rate is predicted to recover (pause) or remain at zero (rewind) at the end of inhibition (Revision Fig. 1). Our new experiments demonstrate that ALM and striatal manipulations correspond to pause and rewind, respectively, following these qualitative criteria, as well (Revision Fig. 2). Together, these results provide a stronger basis for distinguishing pause and rewind than what was presented in the original manuscript. Please see **Section I** for details.

Regarding the specific comment on Extended Data Fig. 9 (now Extended Data Fig. 12), we believe the reviewer is referring to the comparison between panels **d** and **I**. Please note that panel **I** depicts a unilateral striatum manipulation, which produces a weaker behavioral effect than the bilateral manipulation shown in panel **p**. The appropriate comparison is between panels **d** and **p**, where recordings were made from the same area (ALM) while either ALM or D1-SPNs in VLS were bilaterally inhibited. In this comparison, we observe a significant difference in recovery time between the two manipulations (95% confidence intervals: 0.08–0.27 vs. 0.31–0.58; $p = 0.002$, hierarchical bootstrap under the null hypothesis that the time to recover to the pre-perturbation level is the same between the two manipulations).

In addition, following the reviewer’s suggestion, we have added figure panels and statistics to directly compare the post-inhibition activity and recovery time across conditions, as shown below (Revision Fig. 22; we did not compare striatal recording during unilateral striatal inhibition for the reason described above).

Revision Fig. 22. Comparison of post-perturbation recovery of activity across experiments (this panel is now in Extended Data Fig. 12).

t. Increase in activity along the RM from the middle to the end of the inhibition period (summarizing *a*, *e*, and *m* for direct comparison across experimental conditions). ALM silencing leads to a significant recovery in RM activity by the end of inhibition, whereas bilateral D1-SPN (STR) inhibition does not show such recovery. The central line in the box plot, median. Top and bottom edges, 75% and 25% points. Whiskers, the lowest/highest datum within the 1.5 interquartile range of the lower/upper quartile. P-value, hierarchical bootstrap. STR recording during STR inhibition are not shown, as the manipulation is unilateral.

u. Comparison of the time required for decoded time to return to the pre-perturbation level after the perturbation ends (summarizing *d*, *h*, and *p* for direct comparison across experimental conditions). Following ALM silencing, activity rapidly recovers, whereas recovery is significantly delayed after D1-SPN (STR) inhibition.

We also edited the main text as follows:

“Consistently, the time for $T_{to\ lick}$ to recover to the pre-perturbation level was significantly longer than in ALM silencing (Extended Data Fig. 12u).”

The second part that I have reservations about is the winding down of activity. The striatum inhibition appears to be weaker than the ALM inhibition in that fewer neurons are inhibited. From a limited set of recordings, the authors estimate that 70% or so of D1 neurons are inhibited. But, D2 MSNs are not inhibited and 30% of D1 MSNs remain. It seems possible that the wind down could be due to incomplete silencing compared to the ALM inhibition. While the paper provides arguments against this concern, it would be helpful if these arguments could be bolstered with more data. For example, does the weaker stimulation in ALM lead to similar fractions of cells inhibited as in striatum? Is there a way to more strongly inhibit striatum? Since so much of the argument about the different roles rests on this winding down, it seems important to completely rule out the possibility that this is due to incomplete silencing.

First, as the reviewer suggested, we attempted to inhibit D1-SPN using stronger laser power (Revision Fig. 23). However, this stronger inhibition induced rebound licking, as evidenced by immediate licks occurring after D1-SPN inhibition in trials without a cue. This rebound effect makes it difficult to evaluate the impact of stronger VLS inhibition on timing behavior.

Revision Fig. 23. Bilateral VLS inhibition using a higher power (0.5 mW) than that used in the main manuscript (0.25 mW) resulted in rebound licking after the end of inhibition, even in trials without a cue. Black: control trials; blue: inhibition trials.

Therefore, as an alternative approach, we analyzed the effects of weaker ALM silencing, as also suggested by the reviewer. To this end, we newly recorded the activity of striatal neurons during weak ALM silencing (0.3 mW instead of 1.5 mW used in other experiments). The fraction of SPN significantly inhibited by weaker ALM inhibition (15% of neurons; 25 out of 169; Extended Data Fig. 18f) is significantly less than the fraction of SPN inhibited by the D1-SPN inhibition (36%, $p = 0.04$, chi-square test), or by stronger ALM inhibition (1.5 mW; 42%). Thus, we tested a range of ALM inhibition strengths that affected SPNs either less than or more than direct D1-SPN inhibition.

This weak ALM inhibition delayed the animal's lick time only by 0.23 (0.11-0.39) s (mean and 95% confidence interval), significantly shorter than 1.0 (0.69 - 1.4) s (mean, 95% confidence interval) shifted with D1-SPN inhibition ($p < 0.001$, hierarchical bootstrap). Notably, weak ALM inhibition did not produce a rewind of ramping dynamics (Revision Fig. 24a left; ALM ramping

activity at the end of inhibition is not significantly different from the control condition, unlike in D1-SPN inhibition; Revision Fig. 24b). In addition, even after the strong ALM silencing, activity recovered at the end of silencing (as summarized in Revision Fig. 22).

Therefore, the rewind effect observed with striatum inhibition cannot be attributed to a quantitative difference in the extent of striatal inhibition, but rather reflects a qualitative difference between the targeted brain areas and cell types (Please also see **Section I** for analyses testing qualitative differences between ALM and D1-SPN manipulations).

These new results are now in Extended Data Fig. 18a-f

Revision Fig. 24. Weak ALM inhibition differs qualitatively and quantitatively from D1 SPN inhibition (these panels are excerpts from Extended Data Fig. 18d, f, a, h, and Extended Data Fig. 14i)

a. Quantification of the change in RM activity during and after ALM silencing. Same format as in Extended Data Fig. 12a. Note that the plot of perturbed trials exhibited a V-shaped profile regardless of laser power, whereas in D1-SPN inhibition (**b**), it showed a linear decay profile regardless of whether the inhibition was unilateral or bilateral. Right, the proportion of striatal projection neurons with spiking activity significantly inhibited or excited ($p < 0.05$; Unchanged, $p \geq 0.05$; rank sum test) by ALM silencing with weak (left) and strong (right) power. Neurons with a mean spike rate above 1 Hz in control trials during the silencing window were analyzed (169 striatal projection neurons, 23 sessions, 12 mice).

b. Same as **a** but for striatum recording during unilateral D1-SPN inhibition at VLS. $n = 113$ cells, 4 mice

The main text has been revised as follows:

Page 16, Line 473

“Even a weak ALM inhibition (0.3 mW instead of the 1.5 mW used in Fig. 5) caused a weak yet rapid decay in RM activity at the onset of photostimulation, followed by a recovery of ramping

during photostimulation (Extended Data Fig. 18cd) and a mild behavioral effect (shifted the median lick time by 0.23 (0.22 - 0.39) s; mean, 95% confidence interval; n = 12 mice; Extended Data Fig. 18ab). Thus, the gradual decay in ALM timing dynamics during D1-SPN inhibition cannot be explained by its weak inhibitory effect.”

2. The terminology and meaning around ramping should be clarified. My understanding from the data as it is presented is that there is a ramping of activity at the level of neural populations. It is less clear to me that ramping happens in single neurons. In a lot of literature regarding, movement planning, timing, and decision making, ramping is shown in single units, especially in recordings from macaques. In Figure 3 panels a,b,f,g, it does not look like each cell is ramping. It looks like cells tile the delay before licking. While this is not a central point of the paper, it is a topic that has been debated in the field and might matter to some readers. It would be helpful to provide more analyses to demonstrate if each cell is indeed ramping or if the tiling that is present in the examples is a more accurate reflection of the substrate underlying the ramp.

We appreciate the reviewer’s comment. As the same question was raised by Reviewer 1 (Minor comments 2), please see our response there (**page 27** of this letter). In brief, according to single-cell polynomial fitting analysis, 59% of total ALM neurons showed peak firing within 100 ms at cue or lick onset (Revision Fig. 12d), and 37% of these neurons showed monotonic ramping activity (corresponds to 23% of all ALM neurons; Revision Fig. 12c); 67% of total striatal neurons show peak firing within 100 ms at cue or lick onset (Revision Fig. 12d), and 37% of these neurons showed monotonic ramping activity (corresponds to 28% of all striatum neurons (Revision Fig. 12a-d). Thus, about a quarter of ALM and striatal neurons exhibit a monotonic ramp. We also implemented a model in which the network generates both ramping and sequential activity via recurrent modules connected by feedforward connectivity (Extended Data Fig. 15), which behaves similarly to the model with only ramping activity (Extended Data Fig. 1). Thus, our conclusions do not depend on individual neurons exhibiting perfect ramping, but rather on the broader temporal structure observed at the population level.

References

1. Li, N. *et al.* Spatiotemporal constraints on optogenetic inactivation in cortical circuits. *eLife* **8**, e48622 (2019).
2. Murakami, M., Vicente, M. I., Costa, G. M. & Mainen, Z. F. Neural antecedents of self-initiated actions in secondary motor cortex. *Nat. Neurosci.* **17**, 1574–1582 (2014).
3. Khona, M. & Fiete, I. R. Attractor and integrator networks in the brain. *Nat. Rev. Neurosci.* **23**, 744–766 (2022).
4. Aksay, E. *et al.* Functional dissection of circuitry in a neural integrator. *Nat. Neurosci.* **10**, 494–504 (2007).
5. Simen, P., Balci, F., deSouza, L., Cohen, J. D. & Holmes, P. A Model of Interval Timing by Neural Integration. *J. Neurosci.* **31**, 9238–9253 (2011).
6. Seung, H. S. How the brain keeps the eyes still. *Proc. Natl. Acad. Sci.* **93**, 13339–13344 (1996).
7. Lim, S. & Goldman, M. S. Balanced cortical microcircuitry for maintaining information in working memory. *Nat. Neurosci.* **16**, 1306–1314 (2013).
8. Cannon, S. C., Robinson, D. A. & Shamma, S. A proposed neural network for the integrator of the oculomotor system. *Biol. Cybern.* **49**, 127–136 (1983).
9. Wong, K.-F. & Wang, X.-J. A Recurrent Network Mechanism of Time Integration in Perceptual Decisions. *J. Neurosci.* **26**, 1314–1328 (2006).
10. Balci, F. & Simen, P. A decision model of timing. *Curr. Opin. Behav. Sci.* **8**, 94–101 (2016).
11. Inagaki, H. K., Fontolan, L., Romani, S. & Svoboda, K. Discrete attractor dynamics underlies persistent activity in the frontal cortex. *Nature* **566**, 212–217 (2019).
12. Li, N., Daie, K., Svoboda, K. & Druckmann, S. Robust neuronal dynamics in premotor cortex during motor planning. *Nature* **532**, 459–464 (2016).
13. Remington, E. D., Narain, D., Hosseini, E. A. & Jazayeri, M. Flexible Sensorimotor Computations through Rapid Reconfiguration of Cortical Dynamics. *Neuron* **98**,

1005-1019.e5 (2018).

14. Remington, E. D., Egger, S. W., Narain, D., Wang, J. & Jazayeri, M. A Dynamical Systems Perspective on Flexible Motor Timing. *Trends Cogn. Sci.* **22**, 938–952 (2018).
15. Wang, J., Narain, D., Hosseini, E. A. & Jazayeri, M. Flexible timing by temporal scaling of cortical responses. *Nat. Neurosci.* **21**, 102–110 (2018).
16. Hintiryan, H. *et al.* The mouse cortico-striatal projectome. *Nat. Neurosci.* **19**, 1100–1114 (2016).
17. Hunnicutt, B. J. *et al.* A comprehensive excitatory input map of the striatum reveals novel functional organization. *eLife* **5**, e19103.
18. Brown, J., Pan, W.-X. & Dudman, J. T. The inhibitory microcircuit of the substantia nigra provides feedback gain control of the basal ganglia output. *eLife* **3**, e02397 (2014).

Dear Reviewers,

We thank the reviewers for their supportive and constructive comments. Please see below for a point-by-point response.

Referee #1 (Remarks to the Author):

The authors have provided a comprehensive response to my review, including new, convincing experiments. I support publication of the current version.

Nevertheless, I don't understand why the authors decided to bury these new results/analyses in the supplemental materials. In particular, the differential behavioral effects of perturbing ALM and striatum at different times are arguably the most convincing and direct evidence that the two areas are performing distinct functions. Yet, the result is buried in Extended Data Figure 19. More generally, my main complaint about this wonderful paper is that much of the important information is mixed within an enormous amount of supplementary materials. Indeed, most of the figures cited in the discussion are extended data figures. Most readers will not go through the Supplementals in detail and end up viewing the results with unjustified skepticism. I would encourage the authors to revisit their decisions about what to include in the primary figures.

We thank the reviewer for the comments. Following the suggestions, We have moved EDF19 to Fig. 6, and some of the single-session traces have been moved to Figs. 4 and 5.

Referee #2 (Remarks to the Author):

The authors have addressed my major concerns. I realized that my previous review had its intro paragraph cut off so, for the record, I'll add a couple sentences from that: "The experimental work is very beautiful and an exemplar for the type of work needed to dissect challenging, fundamental computations that involve multi-brain region interactions. Furthermore, the model-based framework for interpreting the data is very elegant and, together with the experiments, provides a beautiful exposition on how to tease apart the source of a neural integrator from its readout and its inputs." [The criticisms that followed this were in the previous Major comments and were generally addressed in this excellent revision, which has made the connections between the conclusions and the data much stronger and more rigorous.]. A few minor comments and suggestions are below, but overall I applaud the authors on a tour de force paper that truly sets the standard for the field.

We thank the reviewer for encouraging comments.

Minor comments:

1) The overall response to review points out that quantities like state dependence and hazard rate offer more robust means of distinguishing pause/slowdown from rewind. Yet, the experiments

doing inactivations at two delays and considering the hazard rate only appear in Supplementary Figure 19 (which I thought was a great figure). Should these therefore be in the main? (There were a couple other things in this spirit that may have been unaddressed or buried, such as reviewer 1's comment that, if one wants to group trials, then a more logical way is to group by level of activity in the ramp mode at the time of laser onset, rather than sorting by lick time to make comparisons; instead, it seems that single trial analysis was put in the supplement but grouping was kept by lick time in the main text).

I will leave to the authors and editor the ultimate decision on such matters, but I found the reply to review to be really nice in addressing the collective reviewer requests for clearer, more robust connections between the conceptual picture/storyline and the data/results, so it seems like a shame to have these more conclusive (I think?) analyses heavily buried in Extended Figures that many readers won't see. Not sure what the space constraints will be, but I could imagine Fig. 4 getting combined with another figure, and then one could consider which figures/figure panels are the best characterizations for a given point being made, i.e. which main text figures should stay or be cut if freeing up space is required. One possible example: do the peaks of the correlations in Figs. 5 and 6 provide the most rigorous and clear assessment, relative to many of the analyses in the supplements, given they are tracking only the peak of a very broad yellow swath of correlation that has very broad error bars?.

Following the suggestion, we combined the original Fig. 3 and 4 and then moved EDF19 to Fig. 6, and some of the single-session traces have been moved to Figs. 4 and 5.

2) As multiple reviewers pointed out, a naïve read of the manuscript would suggest that there could be positive feedback between ALM and STR since each is driving the other. Page 12 (points 1 and 2 paragraphs) and figures 5 and 6 do a nice job of explaining the on vs. off manifold aspects of ALM input to striatum and their consequences on activity, and of the on-manifold optogenetic perturbation of striatal D1 activity. However, I think it's very much buried that the reason ALM can follow striatal input without this causing a positive feedback loop between striatum and ALM, is because striatum is driving the component of ALM firing that projects onto the off-manifold aspect of STR activity, thus avoiding the feedback loop that would occur if STR drove the component of ALM firing that projects back to the 'on-manifold/integrating mode' of the striatum. One really needs to understand Extended Data Fig. 1f (which hardly is explained) to see this. It seems like this STR-to-ALM aspect of the picture needs to be clarified, not only in the supplemental/extended data material but also in the main text.

Also, related to this, I think the critical thing is not (as suggested by the reply to review and Extended data fig. 5m) that the ALM ramp mode and trial-history mode are orthogonal. Indeed, one could imagine that these are orthogonal and each is at 45 degree angles to the (left eigenvector of) the striatal integrating mode, so that both would get integrated. I think the implicit assumption is that the trial-history mode is perfectly aligned with the (left eigenvector of the) striatal integrating mode, so that then the ALM ramp and trial-history modes being orthogonal make the ALM ramp mode not get integrated (and therefore, if the striatum projects to the ALM ramp mode, then there is no positive feedback loop). However, I think the key condition is that the striatum projects to a mode of ALM dynamics that is orthogonal to the (left eigenvector of) the striatal integrating mode.

To make this point clearer in the main text, we added the text below:

Line. 408:

“ALM ramping is orthogonal to the subcortical integrator, preventing runaway excitation”

4) In the PCA analysis of Extended Data Fig. 9, PC2 is explained as looking like the trial history mode. However, unless I'm missing something, it seems very much like the middle mode and not nearly so much like the trial history mode -- in fact, as shown in panel d, while it has a “significant” deviation from orthogonality, the peak of the shown distribution is still very close to $\pi/2$, i.e. close to orthogonal. Clarification is needed here on what to take away from this analysis, and what not to take away from it.

We agree that the activity profile of PC2 following the cue looks similar to Middle mode, but the more important point is that its ITI activity anticipates lick times in a manner similar to the trial-history model.

In high-dimensional spaces, small perturbations can induce large angular deviations between vectors, typically driving them toward orthogonality. Thus, the key comparison is between shuffled and actual data, which shows significantly lower values in the data. Therefore, instead of using the term ‘aligned’, we now refer to them as ‘non-orthogonal’ to more accurately describe the relationship between the modes and PCAs. Since this result is not the most important point in the manuscript, we have moved this figure and discussion to the Supplementary Information.

5) In Extended Data Fig. 8, why is decoded time to lick negative on the y-axis? If negative means “longer time preceding lick”, then why is the loud cue above the faint cue, which seems like that would mean the loud cue has more time from cue to lick in panel j (which would contradict panel k, which makes intuitive sense in showing there is typically less time from cue to lick for the loud cue). Are the loud and faint lines in this panel switched, or can you clarify the meaning of the y-axis (or maybe I just missed something obvious).

Thank you for pointing this out. It is because “decoded time to lick” was baseline-subtracted relative to the value at cue onset (time 0), we did so because the mean lick-time in control trials varies across sessions. This subtraction causes the decoded time to lick to start at 0 and take on negative values as it approaches the actual lick. We have now clarified this process in the corresponding figure legend.

6) In the Extended Data Fig. 6a3-6e row, it would be worth making clear what a value of 1 means for the square sum of spiking activity explained axis.

Thanks for pointing out this confusion. We have updated the corresponding axis label for clarification as follows:

“Proportion of square sum of activity explained”

Referee #3 (Remarks to the Author):

The authors have added new data and modeling that support the overall conclusions and interpretations. My concerns have been adequately addressed. I continue to feel that the results are somewhat mixed (or difficult to interpret) between rewinding and pausing the timer. The text makes the distinction between areas to sound black and white, including in the abstract. As I mentioned in my previous review and as other reviewers mentioned as well, some of the results appear more intermediate. I feel it would be appropriate for the authors to scale back any language that makes the result seem black and white and/or add in some caveats or more nuanced interpretation throughout. This will both be more accurate to the data, in my opinion, and likely help the paper to have longevity in case future studies reveal that there are somewhat mixed functions of these areas.

We thank the reviewer for their supportive comments. Following suggestions, we have avoided definitive statements across the manuscript, such as adding “appeared to”, “consistent with” instead of using definitive terms.

On behalf of the authors,
Hidehiko Inagaki

Max Planck Florida Institute for Neuroscience
1 Max Planck Way
Jupiter FL 33458